# PertEval-scFM: Benchmarking Single-Cell Foundation Models for Perturbation Effect Prediction

**Aaron Wenteler** [* † 1]  **Martina Occhetta** [* † 1]  **Nikhil Branson** [† 1 2]  **Victor Curean** [3]  **Magdalena Huebner** [1]
**William Dee** [1]  **William Connell** [4]  **Siu Pui Chung** [1]  **Alex Hawkins-Hooker** [5]  **Yasha Ektefaie** [6]
**César Miguel Valdez Córdova** [‡ 7 8]  **Amaya Gallagher-Syed** [‡ † 1]

## Abstract

*In silico* modeling of transcriptional responses to perturbations is crucial for advancing our understanding of cellular processes and disease mechanisms. We present PertEval-scFM, a standardized framework designed to evaluate models for perturbation effect prediction. We apply PertEval-scFM to benchmark zero-shot single-cell foundation model (scFM) embeddings against baseline models to assess whether these contextualized representations enhance perturbation effect prediction. Our results show that scFM embeddings offer limited improvement over simple baseline models in the zero-shot setting, particularly under distribution shift. Overall, this study provides a systematic evaluation of zero-shot scFM embeddings for perturbation effect prediction, highlighting the challenges of this task and the limitations of current-generation scFMs. Our findings underscore the need for specialized models and high-quality datasets that capture a broader range of cellular states. Source code and documentation can be found at: `https://github.com/aaronwtr/PertEval`.

## 1. Introduction

Inspired by the success of foundation models in fields such as natural language processing (Devlin et al., 2019; Brown et al., 2020; OpenAI, 2024) and computer vision (Dosovitskiy et al., 2021), there has been an increase in the development of biological foundation models. Among these, single-cell foundation models (scFMs) leverage vast amounts of unlabeled transcriptomic single-cell RNA sequencing (scRNA-seq) data to learn contextualized representations through self-supervised pre-training (Ericsson et al., 2022). Fine-tuning the resulting model on labeled data enhances the performance on downstream applications, such as cell-type classification, gene regulatory network inference, and the prediction of cellular responses to perturbations (Yang et al., 2022; Kedzierska et al., 2023; Theodoris et al., 2023; Rosen et al., 2023; Cui et al., 2024; Wen et al., 2023; Hao et al., 2023).

A perturbation refers to any intervention or event leading to the phenotypic alteration of a cell. Perturbation response prediction can provide invaluable insights into cellular mechanisms and disease progression, facilitating the mapping of genotype to phenotype and the identification of potential drug targets (Lotfollahi et al., 2019). Numerous models, here referred to as *narrow perturbation prediction models* (NPPMs), have been developed specifically for this task (Gavriilidis et al., 2024). However, perturbation response prediction is a challenging task, as demonstrated by the difficulty of models to improve consistently over simpler baseline methods (Wu et al., 2024; Branson et al., 2024; Ahlmann-Eltze et al., 2024).

Recently, there has been a concerted effort to evaluate biological foundation models. The Therapeutic Data Commons, an open science initiative, has curated some datasets, models and benchmarks for single-cell analysis (Velez-Arce et al., 2024). Additionally, Wu et al. (2024) and Ahlmann-Eltze et al. (2024) show that simple baseline models perform comparably to scFMs in predicting transcriptomic response to perturbations. However, their analysis does not account for distribution shift and focuses only on predictions for highly variable genes, many of which show little to no effect in response to a perturbation (Nadig et al., 2024).

Yet, distribution shift is a well-documented issue with scRNA-seq data (Boiarsky et al., 2023; Marklund et al., 2020), which often hinders the deployment of models that appear to perform well during evaluation. Distribution shift

---

[*]Equal contribution [†]Core contributor [‡]Senior author [1]Queen Mary University of London [2]University of Oxford [3]University of Medicine and Pharmacy of Cluj-Napoca [4]University of California, San Francisco [5]University College London [6]Harvard University [7]Mila [8]McGill University. Correspondence to: Aaron Wenteler <a.wenteler@qmul.ac.uk>, Martina Occhetta <m.occhetta@qmul.ac.uk >.

*Proceedings of the 42nd International Conference on Machine Learning*, Vancouver, Canada. PMLR 267, 2025. Copyright 2025 by the author(s).

can occur as a consequence of inherent technical and biological noise, abundant in scRNA-seq data, and, while scFMs have been proposed to mitigate such problems, there have been conflicting reports on their ability to do so (Theodoris et al., 2023; Cui et al., 2024; Wu et al., 2024). This highlights the need for a comprehensive benchmark to evaluate their limitations and failure modes, specifically for distribution shift.

## 1.1. Contributions

Here, we present PertEval-scFM to address these research gaps by providing:

- A standardized framework for evaluating biologically meaningful perturbation effect prediction in a zero-shot setting. The source code and documentation can be found on our GitHub.

- Integration of a spectral graph theory method – SPECTRA (Ektefaie et al., 2024) – that allows us to assess model generalizability under distribution shift, a crucial consideration for real-world applications of scFMs.

- A toolbox of comprehensive metrics, providing a detailed analysis of model performance, focusing on assessing robustness and sensitivity to distribution shifts.

## 2. PertEval-scFM

PertEval-scFM is designed to assess the zero-shot information content of scFM embeddings for perturbation effect prediction. To achieve this goal, we obtain zero-shot embeddings from five pre-trained scFMs across four datasets, then train a multi-layer perceptron (MLP) probe for each scFM. This method mirrors established probing techniques to assess the semantic content of embeddings (Alain & Bengio, 2018; Jin et al., 2019). This approach enables fair evaluation of the base information content of embeddings across models, as it evaluates representation quality while removing confounding effects introduced by task-specific prediction heads (Tenney et al., 2019; Radford et al., 2021). In Figure 1 we present an overview of the pipeline, composed of three parts: data pre-processing, training, and evaluation.

## 2.1. Data Pre-Processing

To interrogate cellular response to perturbations, we use high-dimensional Perturb-seq screens, which combine single-cell RNA sequencing with CRISPR-mediated genetic perturbations, enabling systematic profiling of transcriptional landscapes at single-cell resolution (Dixit et al., 2016). Perturb-seq data consists of transcriptomic data for unperturbed control cells $C \in \mathbb{R}^{n_c \times g}$ and perturbed cells $P \in \mathbb{R}^{n_p \times g}$, where $n_c$ and $n_p$ correspond to the number of control and perturbed cells being measured respectively,

and $g$ corresponds to the number of genes in the dataset (see Appendix A.1).

### 2.1.1. DATA PREPARATION

Briefly, during our pre-processing we normalize and log-transform the raw expression count matrix $C$. We then select the top 2,000 highly variable genes (HVGs), $v$, obtaining a reduced control matrix $C \in \mathbb{R}^{n_c \times v}$. Additionally, we identify the top 20 differentially expressed genes (DEGs) for each perturbation to ensure that our evaluations capture biologically relevant gene expression changes (see Appendix A.2).

### 2.1.2. DATA FEATURIZATION

To generate the input features for our baselines, we randomly select 500 cells from $C$ to form a pseudo-bulk sample $\widetilde{C}$. To combat noise and sparsity issues, we calculate the average expression across $\widetilde{C}$ and repeat this process $n_p$ times. The resulting basal gene expression vectors can then be paired with perturbed cells, resulting in control expression feature matrix $X_c \in \mathbb{R}^{n_p \times v}$ (see Appendix C.1).

**Single-cell foundation model embeddings.** To construct the control cell embeddings, we then feed our input matrix $X_c$ into the scFM:

$$f_{\text{scFM}}(X_c) = Z_c, \qquad Z_c \in \mathbb{R}^{n_p \times e} \qquad (1)$$

where $e$ is the embedding dimension. To simulate genetic perturbations *in silico*, we adopt a universal strategy of nullifying the expression of a targeted gene. This approach is motivated by prior work demonstrating that discrete manipulations of rank-order vectors yield biologically meaningful shifts in cell embeddings (Theodoris et al., 2023) and by our own experiments showing that alternative representations – such as modeling a CRISPRa perturbation by expression doubling – do not improve performance (see Appendix D.1). Furthermore, these representations ensure consistent testing conditions across all models, given the current lack of standardized methods for generating comparable *in silico* representations of perturbations. Perturbed cell embeddings $Z_p \in \mathbb{R}^{n_p \times e}$ are therefore generated by setting the expression counts of perturbed genes to zero in cells exposed to that perturbation. The final input for the MLP probe is then formed by concatenating the control and perturbation embeddings (see Appendix C.2):

$$Z_{\text{scFM}} = Z_c \oplus Z_p \qquad (2)$$

**Raw expression data.** To serve as a baseline against which to compare the performance of the scFM embeddings, we use our input matrix $X_c$. Here, we model single-gene perturbations by calculating the gene co-expression matrix $G_c \in \mathbb{R}^{n_p \times v}$ between the perturbed genes and the highly

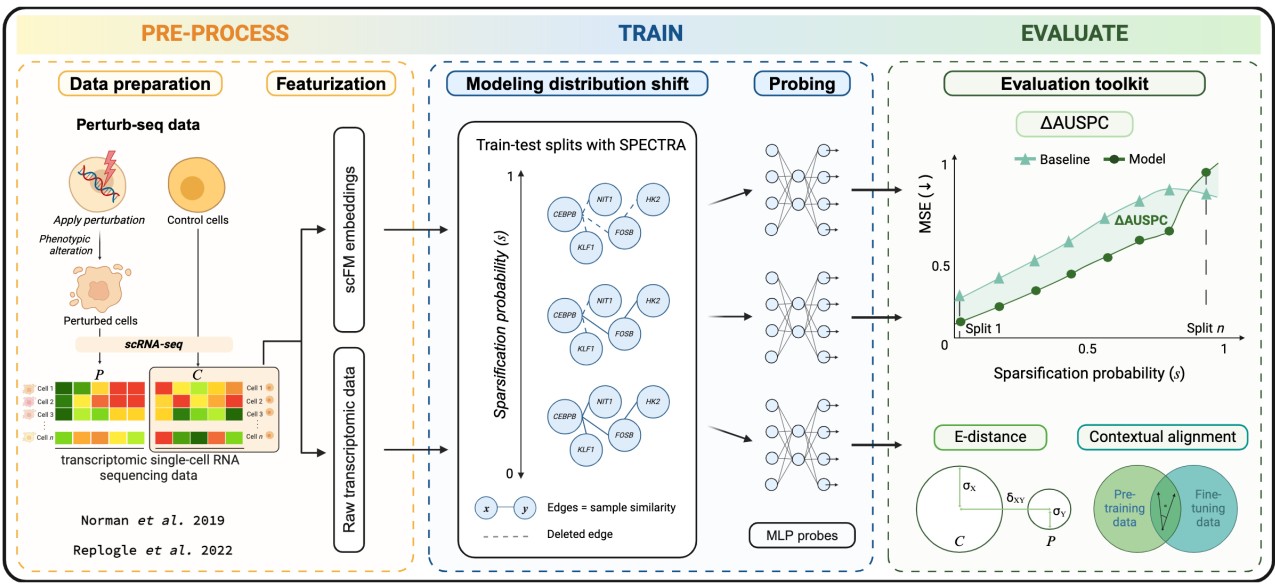

*Figure 1.* PertEval-scFM framework (left to right) – data pre-processing, training of MLP probes under different sparsification conditions; evaluation of trained models with AUSPC, E-distance and contextual alignment metrics.

variable genes in $X_c$. Similarly, for double-gene perturbations, we calculate the co-expression matrices for the individual perturbations, and then average them to obtain $G_c$. We then concatenate the control and perturbation embeddings to form the final input for the MLP probe (see Appendix C.1).

$$Z_{\text{GE}} = X_c \oplus G_c \qquad (3)$$

## 2.2. Baseline Models

We establish baseline models against which to compare the performance of the MLP probes trained with scFM embeddings.

**MLP baseline.** The MLP baseline uses log-normalized raw gene expression data directly as input. This allows us to ensure that any performance differences can be traced back to the semantic information introduced into the embeddings by the scFM. The perturbation effect $\hat{\delta}$ is predicted as follows:

$$\hat{\delta}^{\eta}(Z_{\text{GE}}) = \text{ReLU}(Z_{\text{GE}}W_1^{\top} + \mathbf{b}_1)W_2^{\top} + \mathbf{b}_2, \qquad (4)$$

where dimensions of parameters $\eta$ correspond to $W_1 \in \mathbb{R}^{h \times 2v}$, $W_2 \in \mathbb{R}^{v \times h}$, $\mathbf{b}_1 \in \mathbb{R}^h$ and $\mathbf{b}_2 \in \mathbb{R}^v$.

**GEARS baseline.** To benchmark zero-shot scFMs against existing task-specific models, we implement GEARS, a state-of-the-art model that integrates raw gene expression data with known biological priors via a graph-based architecture (Roohani et al., 2023). We reproduce the original implementation, modifying only the train–test splits to align

with the SPECTRA framework. All other training configurations, hyperparameters, and preprocessing steps remain at their default values. To ensure consistency across our benchmark, we train GEARS from scratch without using pretrained weights. While GEARS is a task-specific model, it shares the same supervised training setup as the other methods we evaluate. The key distinction lies in how representations are handled: GEARS learns representations end-to-end, while our probes evaluate the information content of fixed inputs (transcriptomic counts or zero-shot scFM embeddings) under controlled conditions.

**Mean baseline.** The mean baseline assumes that a perturbation has little effect on the perturbed cell's gene expression. This reflects the biological reality that most perturbations result in small changes in gene expression, providing a simple biologically plausible null model highlighting the challenge inherent in distinguishing meaningful perturbation effects from background variability in single-cell data. The predicted perturbation effect, $\hat{\delta}$, is then simply computed as the deviation of the cell's gene expression, $X_c$, from the mean gene expression of all cells in the same context, $\overline{X}_c$, as defined by $\hat{\delta} = \overline{X}_c - X_c$.

## 2.3. Training

### 2.3.1. MLP PROBE FOR PERTURBATION EFFECT PREDICTION

**MLP probe design.** A 1-hidden layer MLP was selected as a probe for its flexibility and simplicity in handling various types of data representations. For each perturbation,

the MLP learns the log fold change perturbation effect $\delta$, defined as:

$$\delta := P - X_c \quad (5)$$

where $P \in \mathbb{R}^{n_p \times v}$ represents the perturbed gene expression matrix. The MLP probe predicts the perturbation effect, denoted by $\hat{\delta}$, described by the following equation:

$$\hat{\delta}^\theta(Z_{\text{scFM}}) = \text{ReLU}(Z_{\text{scFM}} W_1^\top + \mathbf{b}_1) W_2^\top + \mathbf{b}_2 \quad (6)$$

The model parameters $\theta$ include the weight matrices $W_1 \in \mathbb{R}^{h \times 2e}$ and $W_2 \in \mathbb{R}^{e \times h}$, where $h$ corresponds to the dimension of the hidden layer, and the bias vectors $\mathbf{b}_1 \in \mathbb{R}^h$ and $\mathbf{b}_2 \in \mathbb{R}^e$.

**MLP probe parameters.** To assess whether parameter count impacts model performance, we analyze the sensitivity of MLP probes to varying model capacities. We train a series of MLP probes with increasing parameter counts on both raw gene expression data and scFM embeddings. The results, summarized in Table E2, show no meaningful relationship between probe capacity and prediction performance. Additional details are provided in Appendix E.2.

### 2.3.2. MODELING DISTRIBUTION SHIFT

To assess the robustness of the MLP probes when using either gene expression data or scFM embeddings, we implement SPECTRA (Ektefaie et al., 2024), a graph-based method that partitions data into increasingly challenging train-test splits while controlling for *cross-split overlap* between the train and test data.

In SPECTRA, edges within the graph represent sample-to-sample similarity. The connectivity of the similarity graph is controlled by the *sparsification probability* ($s$). For each split, this connectivity is adjusted by stochastically removing edges with probability $s$. We introduce the constraint $s < s_{\max}$, where $s_{\max}$ is empirically chosen to ensure a sufficient number of samples in both the train and test sets. After sparsification, the train and test sets are sampled from distinct subgraphs. As the sparsification probability increases, the degree of similarity between the train and test sets decreases, making it harder for the model to generalize to unseen perturbations effectively (see Appendix F).

### 2.4. Evaluation

Following the empirical findings from Ji et al. (2023), we adopt *Mean Squared Error (MSE)* as our primary evaluation metric. In addition, to address the current lack of standardized evaluation metrics for perturbation effect prediction, we propose using three complementary metrics: (i) *Area Under the SPECTRA Performance Curve (AUSPC)* to quantify model generalization across distribution shifts, (ii) *E-distance* to quantitatively measure perturbation effect

magnitude, and (iii) *contextual alignment* to measure how the overlap between pre-training and fine-tuning datasets influences model performance. Together, these metrics provide a robust basis for comparative scFM evaluation while capturing distinct aspects of predictive performance in transcriptomic perturbation response modeling.

#### 2.4.1. AUSPC

To evaluate robustness under distribution shift, we adapt the approach introduced by Ektefaie et al. (2024) and define the AUSPC as:

$$\text{AUSPC} = \int_0^{s_{\max}} \phi(s)\, ds \quad (7)$$

where $\phi(s)$ is the MSE as a function of the sparsification probability $s$ used to define each train-test split. Integrating the MSE across $s$ yields a single performance metric that reflects a model's ability to generalize under increasing distribution shift. The integral is approximated with the trapezoidal rule (see Appendix F.2).

Motivated by the observation that simple baselines often perform surprisingly well in perturbation prediction, we introduce the $\Delta$AUSPC metric. This metric anchors a model's robustness to the mean baseline. The $\Delta$AUSPC is defined as:

$$\Delta\text{AUSPC} = \int_0^{s_{\max}} [\phi_b(s) - \phi_m(s)]\, ds \quad (8)$$

Here, $\phi_b$ represents the MSE of the mean baseline, and $\phi_m$ is the MSE of the model being evaluated. A positive $\Delta$AUSPC indicates that the model outperforms the baseline, while a negative value suggests the opposite. This metric provides a clear measure of a model's generalizability improvement over simply predicting the mean perturbation effect.

#### 2.4.2. E-DISTANCE

We use the E-distance, introduced by Peidli et al. (2024), to quantify the difference between perturbed and control cell gene expression profiles (see Appendix G.1). This metric captures both within-group variability and distributional differences, offering a robust measure of perturbation effect strength. With E-distance, we can better analyze the characteristics of perturbations that models handle well versus those they struggle with, providing context for model performance—particularly in cases where outlier perturbations may not be immediately apparent using traditional metrics.

#### 2.4.3. CONTEXTUAL ALIGNMENT

While pre-training dataset size is often linked to improved downstream model performance, recent research emphasizes the critical role of data quality over dataset size (El-Nouby et al., 2021; Fournier et al., 2024). We therefore suggest the inclusion of a contextual alignment metric, which

quantifies the similarity between the pre-training and fine-tuning datasets, and its effect on model performance. We calculate the cross-split overlap between the pre-train and fine-tune datasets using cosine similarity to determine how representative the pre-training data is of the fine-tuning data (see Appendix H.1).

### 2.5. Models and Datasets

#### 2.5.1. SINGLE-CELL FOUNDATION MODELS

PertEval-scFM currently evaluates the performance of the following five scFMs: scBERT (Yang et al., 2022), Geneformer (Theodoris et al., 2023), scGPT (Cui et al., 2024), scFoundation (Hao et al., 2023) and UCE (Rosen et al., 2023). See Appendix B.1 for details on their architecture and pre-training data.

#### 2.5.2. DATASETS

**Norman**. PertEval-scFM is applied to the 105 single-gene and 91 double-gene perturbation datasets derived from a Perturb-seq screen in K562 cells from Norman et al. (2019). These datasets contain strong CRISPRa perturbation signals, as well as baseline expression for unperturbed cells, which allows for the systematic evaluation of model performance in predicting the effects of genetic perturbations at single-cell resolution.

**Replogle**. Additionally, we apply our framework to the two single-gene perturbation datasets from Replogle et al. (2022), which profile transcriptomic responses to CRISPRi-mediated genetic perturbations in both K562 (2,058 perturbations) and RPE1 (2,394 perturbations) cells. Compared to the Norman dataset, the overall perturbation effect signal in Replogle et al. (2022) is less pronounced, despite the considerably larger number of perturbations (Peidli et al., 2024). For additional details on the datasets, see Appendix A.

## 3. Results

### 3.1. Evaluation across 2,000 HVGs

In our initial evaluation, we assess the models' ability to predict the effect of perturbations on the top 2,000 HVGs. We present our evaluation results in Table 1, Table 2 and Figure 2 and discuss the results for each dataset below.

**Norman single-gene.** For single-gene perturbations, GEARS achieved the highest performance with an AUSPC of 0.00815, significantly outperforming all scFM models. This is followed by the MLP baseline at 0.0448, which is slightly better than the scFM models. Performance differences between the scFMs were minimal, suggesting similar capabilities in predicting perturbation effects. As the sparsification probability $s$ increased from 0.1 to 0.7, MSE values

rose across all models. However, scFMs showed a steeper performance drop at higher sparsification levels compared to the MLP baseline, suggesting that the zero-shot embeddings are less robust to distribution shifts.

**Norman double-gene.** The ranking pattern observed for single-gene perturbations persisted for double-gene perturbations, with GEARS and the MLP baseline performing better than the scFMs. Most scFM models displayed similar performance, with a notable exception of scGPT, which experienced a significant drop in performance, moving from rank 3 to rank 8. This increased the performance spread, with a larger $\Delta$AUSPC difference ($\Delta$AUSPC range of 0.00980 for double-gene perturbations compared to 0.00126 for single-gene), highlighting greater variability in model performance for the more complex perturbations. In line with our expectations, increasing the sparsification probability led to higher MSE across all models.

**Replogle K562.** For the K562 cell line, GEARS maintained its superior performance with an AUSPC of 0.0082, followed by scGPT and UCE, both with an AUSPC of 0.1384. The MLP baseline only ranked fifth with an AUSPC of 0.1420. Apart from GEARS, none of these differences were statistically significant, as indicated by the overlapping error bars in Figure 2b.

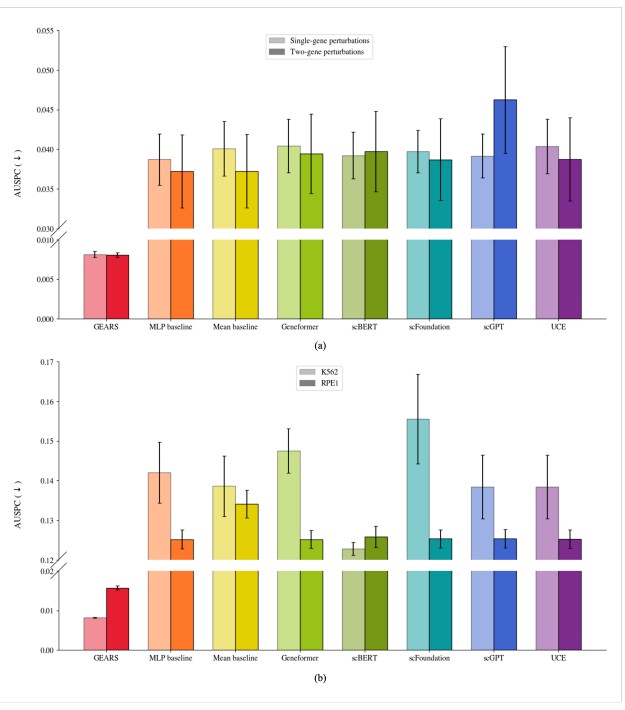

*Figure 2.* Average AUSPC (↓) across sparsification probabilities for each model with standard error bars. (a) Norman single-gene (left) and double-gene (right) perturbation (b) Replogle K562 (left) and RPE1 (right)

Table 1. Perturbation effect prediction evaluation across 2,000 HVGs. Models are listed in the specified order.

| Dataset | Model | ↓ MSE ($10^{-2}$) | | | | | | | ↓ AUSPC ($10^{-2}$) | ↑ ΔAUSPC ($10^{-2}$) | Rank |
|---|---|---|---|---|---|---|---|---|---|---|---|
| | | S 0.1 | S 0.2 | S 0.3 | S 0.4 | S 0.5 | S 0.6 | S 0.7 | | | |
| Norman single-gene | GEARS | 0.887 ± 0.202 | 0.937 ± 0.177 | 1.120 ± 0.167 | 1.693 ± 0.328 | 1.750 ± 0.401 | 1.0067 ± 0.427 | 1.000 ± 0.257 | 0.815 ± 0.039 | **3.7968** | **1** |
| | MLP baseline | 6.288 ± 0.282 | 6.410 ± 0.289 | 6.699 ± 0.705 | 6.453 ± 0.584 | 5.984 ± 0.458 | 6.502 ± 1.277 | 7.065 ± 1.022 | 4.484 ± 0.299 | **0.1280** | **2** |
| | Mean baseline | 6.177 ± 0.204 | 5.980 ± 0.621 | 6.497 ± 0.513 | 6.219 ± 0.308 | 6.659 ± 0.154 | 7.413 ± 1.038 | 8.430 ± 0.540 | 4.612 ± 0.317 | - | 6 |
| | Geneformer | 6.257 ± 0.049 | 6.132 ± 0.622 | 6.565 ± 0.520 | 6.395 ± 0.300 | 6.550 ± 0.140 | 7.382 ± 1.155 | 8.525 ± 0.494 | 4.651 ± 0.309 | -0.0396 | 8 |
| | scBERT | 6.301 ± 0.316 | 6.341 ± 0.356 | 6.761 ± 0.765 | 6.363 ± 0.544 | 5.924 ± 0.418 | 6.451 ± 1.200 | 8.488 ± 0.558 | 4.537 ± 0.268 | 0.0748 | 4 |
| | scFoundation | 6.421 ± 0.317 | 6.366 ± 0.356 | 6.793 ± 0.764 | 6.440 ± 0.538 | 5.919 ± 0.417 | 6.705 ± 1.183 | 8.601 ± 0.537 | 4.594 ± 0.246 | 0.0179 | 5 |
| | scGPT | 6.237 ± 0.218 | 6.340 ± 0.608 | 6.765 ± 0.428 | 6.363 ± 0.345 | 5.926 ± 0.174 | 6.400 ± 1.144 | 8.506 ± 1.020 | 4.525 ± 0.255 | **0.0863** | **3** |
| | UCE | 6.258 ± 0.311 | 6.132 ± 0.620 | 6.565 ± 0.514 | 6.387 ± 0.307 | 6.551 ± 0.155 | 7.370 ± 1.065 | 8.479 ± 0.601 | 4.647 ± 0.312 | -0.0355 | 7 |
| Norman double-gene | GEARS | 0.783 ± 0.044 | 0.960 ± 0.050 | 1.153 ± 0.049 | 1.230 ± 0.289 | 1.467 ± 0.351 | 1.223 ± 0.147 | 1.810 ± 0.287 | 0.808 ± 0.028 | **4.254** | **1** |
| | MLP baseline | 5.261 ± 0.100 | 5.913 ± 0.255 | 5.728 ± 0.402 | 6.635 ± 0.161 | 7.675 ± 0.953 | 6.050 ± 0.763 | 5.198 ± 0.593 | 4.253 ± 0.073 | **0.002** | **2** |
| | Mean baseline | 5.257 ± 0.102 | 5.910 ± 0.255 | 5.722 ± 0.401 | 6.644 ± 0.167 | 7.674 ± 0.962 | 6.071 ± 0.772 | 5.201 ± 0.594 | 4.255 ± 0.073 | - | 3 |
| | Geneformer | 5.514 ± 0.067 | 6.145 ± 0.182 | 6.029 ± 0.458 | 6.742 ± 0.287 | 7.937 ± 1.187 | 7.246 ± 0.707 | 5.179 ± 0.628 | 4.503 ± 0.081 | -0.248 | 6 |
| | scBERT | 5.515 ± 0.067 | 6.159 ± 0.196 | 6.022 ± 0.465 | 6.757 ± 0.281 | 7.999 ± 1.240 | 6.493 ± 0.736 | 7.110 ± 0.579 | 4.533 ± 0.081 | -0.278 | 7 |
| | scFoundation | 5.564 ± 0.051 | 6.173 ± 0.196 | 6.050 ± 0.462 | 6.755 ± 0.279 | 7.944 ± 1.186 | 6.382 ± 0.876 | 5.238 ± 0.578 | 4.432 ± 0.467 | -0.177 | 4 |
| | scGPT | 5.515 ± 0.067 | 6.153 ± 0.189 | 6.023 ± 0.464 | 6.766 ± 0.287 | 8.272 ± 1.377 | 8.826 ± 0.182 | 14.906 ± 2.154 | 5.184 ± 0.132 | -0.929 | 8 |
| | UCE | 5.514 ± 0.066 | 6.145 ± 0.183 | 6.029 ± 0.460 | 6.736 ± 0.289 | 7.939 ± 1.184 | 6.352 ± 0.831 | 5.612 ± 0.665 | 4.435 ± 0.085 | -0.180 | 5 |

Table 2. Perturbation effect prediction evaluation across 2,000 HVGs. Models are listed in the specified order.

| Dataset | Model | ↓ MSE | | | | | | | ↓ AUSPC | ↑ ΔAUSPC ($10^{-2}$) | Rank |
|---|---|---|---|---|---|---|---|---|---|---|---|
| | | S 0.1 | S 0.2 | S 0.3 | S 0.4 | S 0.5 | S 0.6 | S 0.7 | | | |
| Replogle K562 | GEARS | 0.0096 ± 0.0005 | 0.0102 ± 0.0006 | 0.0129 ± 0.0003 | 0.0133 ± 0.0005 | 0.0150 ± 0.0003 | 0.0169 ± 0.0016 | 0.0175 ± 0.0009 | 0.0082 ± 0.0001 | **13.044** | **1** |
| | MLP baseline | 0.2125 ± 0.0007 | 0.2245 ± 0.0011 | 0.2343 ± 0.0055 | 0.2269 ± 0.0084 | 0.2437 ± 0.0151 | 0.2445 ± 0.0348 | 0.2799 ± 0.0279 | 0.1420 ± 0.0077 | -0.3385 | 5 |
| | Mean baseline | 0.2129 ± 0.0007 | 0.2272 ± 0.0017 | 0.2319 ± 0.0093 | 0.2252 ± 0.0089 | 0.2356 ± 0.0231 | 0.2435 ± 0.0267 | 0.2703 ± 0.0347 | 0.1386 ± 0.0076 | - | 4 |
| | Geneformer | 0.2274 ± 0.0028 | 0.2374 ± 0.0014 | 0.2515 ± 0.0008 | 0.2412 ± 0.0058 | 0.2506 ± 0.0047 | 0.2526 ± 0.0202 | 0.2639 ± 0.0330 | 0.1475 ± 0.0056 | -0.8887 | 6 |
| | scBERT | 0.2071 ± 0.0014 | 0.2073 ± 0.0019 | 0.2050 ± 0.0041 | 0.2032 ± 0.0028 | 0.2064 ± 0.0035 | 0.2022 ± 0.0055 | 0.2012 ± 0.0004 | 0.1228 ± 0.0016 | * | * |
| | scFoundation | 0.2396 ± 0.0033 | 0.2576 ± 0.0109 | 0.2682 ± 0.0102 | 0.2644 ± 0.0153 | 0.2788 ± 0.0141 | 0.2506 ± 0.0278 | 0.2639 ± 0.0331 | 0.1556 ± 0.0113 | -1.6977 | 7 |
| | scGPT | 0.2107 ± 0.0 | 0.2245 ± 0.0016 | 0.2287 ± 0.0086 | 0.2223 ± 0.0081 | 0.2321 ± 0.0213 | 0.2390 ± 0.0245 | 0.2635 ± 0.0327 | 0.1384 ± 0.0080 | **0.0221** | **2** |
| | UCE | 0.2114 ± 0.0007 | 0.2245 ± 0.0016 | 0.2287 ± 0.0086 | 0.2223 ± 0.0081 | 0.2321 ± 0.0212 | 0.2391 ± 0.0245 | 0.2635 ± 0.0326 | 0.1384 ± 0.0080 | **0.0220** | **3** |
| Replogle RPE1 | GEARS | 0.0154 ± 0.0004 | 0.0183 ± 0.0001 | 0.0217 ± 0.0006 | 0.0254 ± 0.0018 | 0.0285 ± 0.0026 | 0.0326 ± 0.0048 | 0.0454 ± 0.0077 | 0.0157 ± 0.0005 | **11.838** | **1** |
| | MLP baseline | 0.2068 ± 0.0023 | 0.2057 ± 0.0027 | 0.2045 ± 0.0011 | 0.2058 ± 0.0058 | 0.2101 ± 0.0053 | 0.2167 ± 0.0073 | 0.2112 ± 0.0042 | 0.1252 ± 0.0024 | **0.8892** | **3** |
| | Mean baseline | 0.2167 ± 0.0015 | 0.2215 ± 0.0027 | 0.2190 ± 0.0013 | 0.224 ± 0.0037 | 0.2203 ± 0.0122 | 0.2353 ± 0.0106 | 0.2246 ± 0.0101 | 0.1341 ± 0.0035 | - | 7 |
| | Geneformer | 0.2053 ± 0.0021 | 0.2056 ± 0.0023 | 0.2048 ± 0.0019 | 0.2062 ± 0.0055 | 0.2113 ± 0.0064 | 0.2170 ± 0.0066 | 0.2103 ± 0.0037 | 0.1251 ± 0.0023 | **0.8922** | **2** |
| | scBERT | 0.2050 ± 0.0029 | 0.2056 ± 0.0023 | 0.2047 ± 0.0019 | 0.2062 ± 0.0054 | 0.2113 ± 0.0059 | 0.2185 ± 0.0070 | 0.2178 ± 0.0068 | 0.1258 ± 0.0027 | * | * |
| | scFoundation | 0.2054 ± 0.0021 | 0.2056 ± 0.0024 | 0.2048 ± 0.0020 | 0.2063 ± 0.0056 | 0.2113 ± 0.0064 | 0.2173 ± 0.0068 | 0.2101 ± 0.0035 | 0.1253 ± 0.0023 | 0.8764 | 6 |
| | scGPT | 0.2053 ± 0.0021 | 0.2056 ± 0.0023 | 0.2047 ± 0.0019 | 0.2062 ± 0.0055 | 0.2112 ± 0.0065 | 0.2171 ± 0.0067 | 0.2103 ± 0.0038 | 0.1253 ± 0.0023 | 0.8768 | 5 |
| | UCE | 0.2053 ± 0.0021 | 0.2056 ± 0.0023 | 0.2048 ± 0.0019 | 0.2062 ± 0.0054 | 0.2113 ± 0.0064 | 0.2170 ± 0.0066 | 0.2104 ± 0.0037 | 0.1253 ± 0.0024 | 0.8791 | 4 |

**Replogle RPE1.** In the RPE1 cell line, GEARS again outperformed all other models with an AUSPC of 0.0157, followed by the Mean baseline (0.1251) and the MLP baseline (0.1252). The scFMs exhibited very similar performance, with AUSPC values tightly clustered around 0.1255 and no significant differences observed.

Overall, GEARS outperforms all zero-shot scFMs and baseline models by an order of magnitude, suggesting that its architecture and training paradigm allow it to better capture underlying biological processes and generalize more effectively across a variety of perturbation types. This underscores the potential value of incorporating stronger inductive biases into the perturbation effect prediction task and suggests that current scFMs, which rely solely on a masked pre-training objective, may primarily capture average perturbation effects in the absence of fine-tuning.

### 3.2. Additional Experiments

We conduct additional experiments on the Norman single- and double-gene perturbation datasets, which exhibit stronger perturbation effects and thus provide a clearer signal for evaluating more challenging prediction tasks. Specifically, we assess the models' performance when restricting the evaluation to the top 20 DEGs, which typically capture the majority of the transcriptional response to a genetic perturbation. Poor performance in this setting would indicate that the semantic information encoded in zero-shot scFM embeddings does not meaningfully contribute to predicting perturbation effects. Lastly, we evaluate the impact of contextual alignment on performance for two of the scFMs.

#### 3.2.1. EVALUATION ACROSS DEGs

We evaluate the models on predicting the effect of perturbations on the top 20 DEGs per perturbation, providing a more stringent test of model performance than the full set of 2,000 HVGs. Indeed, genetic perturbations typically alter the expression of a limited subset of genes within the transcriptome (Figure A4), hence models predicting mean gene expression can still achieve low MSE values across 2,000 HVGs. The results of the evaluation across DEGs are displayed in Table 3.

**Norman single-gene.** For single-gene perturbations, GEARS achieved the highest performance (AUSPC 0.266), significantly outperforming all scFMs and baseline models. The MLP baseline ranked second (0.342), followed closely by UCE (0.334), which was the best-performing scFM. As the sparsification probability increased, MSE values worsened across all models, with Geneformer and scGPT exhibiting the steepest decline in performance. scBERT performed best among scFMs across most sparsity levels

*Table 3.* Perturbation effect prediction evaluation across the top 20 DEGs per perturbation. Note that for double-gene perturbations split 0.5, there were not enough perturbations that passed our quality control to define multiple replicates.

| Dataset | Model | ↓ MSE | | | | | | | ↓ AUSPC | ↑ ΔAUSPC ($10^{-2}$) | Rank |
|---|---|---|---|---|---|---|---|---|---|---|---|
| | | S 0.1 | S 0.2 | S 0.3 | S 0.4 | S 0.5 | S 0.6 | S 0.7 | | | |
| Norman single-gene | GEARS | 0.284 ± 0.024 | 0.215 ± 0.037 | 0.314 ± 0.071 | 0.256 ± 0.035 | 0.341 ± 0.014 | 0.682 ± 0.194 | 0.888 ± 0.285 | 0.266 ± 0.018 | **7.967** | **1** |
| | MLP gene expression | 0.466 ± 0.051 | 0.468 ± 0.074 | 0.456 ± 0.039 | 0.497 ± 0.042 | 0.521 ± 0.071 | 0.513 ± 0.123 | 0.622 ± 0.172 | 0.342 ± 0.013 | 0.312 | 4 |
| | Mean baseline | 0.479 ± 0.050 | 0.474 ± 0.078 | 0.489 ± 0.053 | 0.492 ± 0.047 | 0.492 ± 0.047 | 0.525 ± 0.126 | 0.604 ± 0.144 | 0.345 ± 0.011 | - | 5 |
| | Geneformer | 0.464 ± 0.048 | 0.464 ± 0.077 | 0.481 ± 0.052 | 0.475 ± 0.042 | 0.483 ± 0.046 | 0.488 ± 0.106 | 0.902 ± 0.220 | 0.351 ± 0.014 | -0.564 | 8 |
| | scBERT | 0.469 ± 0.050 | 0.464 ± 0.077 | 0.481 ± 0.053 | 0.475 ± 0.042 | 0.482 ± 0.045 | 0.499 ± 0.117 | 0.608 ± 0.149 | 0.336 ± 0.011 | 0.878 | 3 |
| | scFoundation | 0.502 ± 0.052 | 0.466 ± 0.077 | 0.489 ± 0.056 | 0.469 ± 0.040 | 0.486 ± 0.046 | 0.567 ± 0.090 | 0.638 ± 0.166 | 0.350 ± 0.011 | -0.486 | 7 |
| | scGPT | 0.463 ± 0.048 | 0.464 ± 0.077 | 0.482 ± 0.053 | 0.475 ± 0.042 | 0.484 ± 0.047 | 0.485 ± 0.105 | 0.828 ± 0.249 | 0.347 ± 0.015 | -0.168 | 6 |
| | UCE | 0.463 ± 0.048 | 0.464 ± 0.077 | 0.482 ± 0.053 | 0.476 ± 0.042 | 0.485 ± 0.047 | 0.484 ± 0.104 | 0.624 ± 0.162 | 0.334 ± 0.012 | 1.078 | 2 |
| Norman double-gene | GEARS | 0.211 ± 0.032 | 0.200 ± 0.013 | 0.296 ± 0.052 | 0.425 ± 0.041 | 0.335 ± 0.0* | 0.473 ± 0.109 | 0.422 ± 0.077 | 0.223 ± 0.010 | **29.9** | **1** |
| | MLP gene expression | 0.484 ± 0.046 | 0.538 ± 0.082 | 0.585 ± 0.061 | 0.618 ± 0.048 | 0.690 ± 0.0* | 0.552 ± 0.049 | 0.500 ± 0.056 | 0.371 ± 0.009 | 15.1 | 2 |
| | Mean baseline | 0.549 ± 0.055 | 0.580 ± 0.075 | 0.615 ± 0.074 | 0.653 ± 0.037 | 0.757 ± 0.0* | 0.659 ± 0.047 | 0.497 ± 0.056 | 0.522 ± 0.053 | - | 8 |
| | Geneformer | 0.527 ± 0.055 | 0.550 ± 0.069 | 0.603 ± 0.076 | 0.661 ± 0.045 | 0.706 ± 0.0* | 0.623 ± 0.054 | 0.487 ± 0.048 | 0.409 ± 0.008 | 11.3 | 3 |
| | scBERT | 0.528 ± 0.056 | 0.550 ± 0.069 | 0.596 ± 0.071 | 0.661 ± 0.041 | 0.740 ± 0.0* | 0.622 ± 0.049 | 0.681 ± 0.086 | 0.418 ± 0.008 | 10.4 | 6 |
| | scFoundation | 0.534 ± 0.057 | 0.554 ± 0.070 | 0.606 ± 0.073 | 0.656 ± 0.045 | 0.683 ± 0.0* | 0.621 ± 0.060 | 0.497 ± 0.051 | 0.410 ± 0.008 | 11.2 | 5 |
| | scGPT | 0.527 ± 0.056 | 0.550 ± 0.069 | 0.597 ± 0.072 | 0.673 ± 0.044 | 0.724 ± 0.0* | 0.724 ± 0.028 | 1.941 ± 0.329 | 0.500 ± 0.018 | 2.2 | 7 |
| | UCE | 0.527 ± 0.055 | 0.550 ± 0.069 | 0.601 ± 0.072 | 0.656 ± 0.043 | 0.726 ± 0.0* | 0.624 ± 0.053 | 0.506 ± 0.048 | 0.410 ± 0.007 | 11.2 | 4 |

(ΔAUSPC 0.00878), while UCE provided the most stable results throughout (ΔAUSPC 0.0108). These results suggest that, while some scFMs exhibit marginal advantages over the mean baseline, the performance gaps remained minimal, with UCE (best) only outperforming Geneformer (worst) by 4.8%.

**Norman double-gene.** A similar ranking pattern was observed for double-gene perturbations, where models outperformed the mean baseline but showed no clear advantage over the MLP. GEARS again demonstrated superior performance (AUSPC 0.223, ΔAUSPC 0.299), surpassing the MLP baseline (AUSPC 0.371, ΔAUSPC 0.151) and all scFMs. The performance gap widened for more complex perturbations, suggesting that GEARS' architecture and training paradigm allow it to better model gene interactions and their impact on the effect of perturbations.

Overall, this evaluation proves more challenging, evidenced by the order of magnitude increase in MSE (see Appendix I.4). However, the results again suggest that zero-shot scFM embeddings provide limited advantage over simple baseline approaches in this setting. Consistent with previous findings, scFM models struggled to generalize to perturbation-specific expression shifts, further reinforcing their limitations in predicting biologically relevant perturbation effects.

### 3.2.2. E-DISTANCE

We analyzed the relationship between perturbation strength and model performance using E-distance to measure perturbation effect magnitude. The results, shown in Figure 3a, confirm that models generally perform worse when predicting perturbations with higher E-distance, which corresponds to stronger perturbation effects. This pattern was consistent across all models, for both single-gene and double-gene perturbations, supporting the hypothesis that training data biased toward moderate perturbation effects limits a model's

ability to generalize to more extreme cases.

Figure 3b further illustrates how perturbation strength varies across train-test splits for both single- and double-gene perturbations. At higher sparsification probabilities, low E-distance perturbations become less frequent, while stronger perturbations appear more often. This aligns with previous observations that model performance declines as sparsity increases, as models are increasingly challenged to predict relatively rare, and strong perturbation effects. Two examples illustrate this trend: *AHR* (Figure 4a), a perturbation with low E-distance, exhibited a narrow effect range (–0.1 to 0.25) and was predicted with relatively high accuracy. In contrast, *CEBPE* (Figure 4b), which produced a broader and more pronounced perturbation effect (–0.5 to 1), was predicted with significantly lower accuracy.

However, deviations from this trend suggest that the magnitude of the perturbation alone does not fully determine prediction difficulty. *CEBPA* (Figure 4d), despite being a strong perturbation, was predicted with relatively high accuracy. This could be attributed to its localized effect on a small subset of genes, with a long tail of mildly or

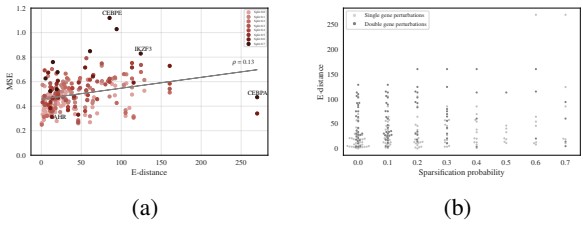

(a)  (b)

*Figure 3.* (a) MSEs for all test perturbations as a function of the E-distance. The predictions displayed are the averaged across all scFMs. (b) The E-distance of all test perturbations stratified per split as a function of the sparsification probability. The mean of the E-distance per split is included in red.

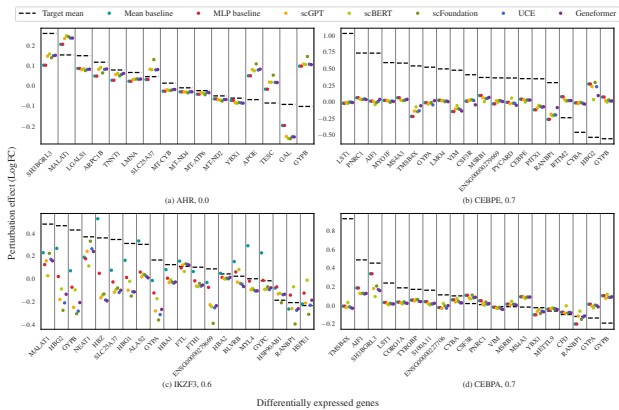

(a) AHR, 0.0

(b) CEBPE, 0.7

(c) IKZF3, 0.6

(d) CEBPA, 0.7

Differentially expressed genes

*Figure 4.* Predictions of models across the top 20 DEGs for 4 perturbations from different splits. Subcaptions indicate perturbation name, sparsification probability. The predictions are included as colored dots, and the target perturbation effect is displayed as a dashed line.

non-affected genes. Conversely, *IKZF3* (Figure 4c), which elicited a weaker overall perturbation, was predicted with lower accuracy—likely due to its atypical effect distribution (Appendix I.7). These results suggest that a model's capability to predict perturbation effects depends not only on the magnitude of the perturbation, but also on its distribution.

Taken together, these findings highlight the importance of a more balanced representation of perturbation effects during training. Ensuring that training data covers a diverse range of perturbation magnitudes and distributions could improve model generalization. Zero-shot scFM embeddings alone do not address this challenge, reinforcing the need for targeted strategies to enhance model robustness for perturbation effect prediction.

### 3.2.3. CONTEXTUAL ALIGNMENT

Previous work by Cui et al. (2024) demonstrated that the performance of zero-shot scFM models in cell-type annotation tasks is highly dependent on the overlap between their pre-training datasets and the downstream task data. To investigate whether this reliance on contextual alignment extends to perturbation effect prediction, we analyzed the similarity between pre-training corpus and fine-tuning dataset, as well as its impact on model performance.

In Figure 5, we compute the contextual alignment between the pre-training datasets of scGPT and scBERT and the Norman dataset. scBERT exhibits a higher alignment score (0.718) compared to scGPT (0.606), indicating that its pre-training corpus is approximately 19% more similar to the Norman dataset. Despite both models displaying comparable MSE across splits, scBERT demonstrates greater robustness, suggesting that contextual alignment plays a more

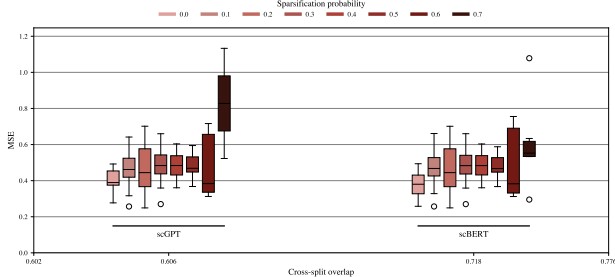

*Figure 5.* MSE as a function of the pre-train and fine-tune data cross-split overlap for scGPT and scBERT.

significant role than dataset scale alone. Notably, scGPT's pre-training corpus is an order of magnitude larger than scBERT's, reinforcing the idea that increasing dataset size does not necessarily compensate for a lack of alignment with the downstream task. For this analysis we focus on scGPT and scBERT, however assessing this phenomenon across a broader range of models and perturbation datasets is necessary to fully understand its impact on perturbation effect prediction. Future studies should explore how curating pre-training datasets to better reflect perturbation-specific distributions influences model performance, particularly in cases where strong or rare perturbations are under-represented in the training corpus.

### 3.3. Limitations

A key limitation of our study is that perturbations are modeled exclusively as knockouts, regardless of their original experimental context. This constraint aligns with prior work and reflects an inherent limitation of current foundation models, as not all architectures support the modeling of gene activation. More broadly, the lack of a standardized approach for representing perturbations in scFMs limits the interpretability and generalizability of findings across studies. To advance the field, there is a pressing need for consensus on how to represent perturbations, ensuring they can robustly capture a diverse range of modalities.

## 4. Conclusion

PertEval-scFM addresses the current lack of consensus in benchmarking models for perturbation effect prediction by introducing a modular evaluation toolkit with diverse metrics, designed to assess and interpret model performance. Notably, our framework accounts for distribution shift, a factor often overlooked in previous studies. We applied PertEval-scFM to evaluate whether zero-shot scFM embeddings provide an advantage over raw gene expression data for perturbation effect prediction. Our results show that current-generation scFM embeddings, when used zero-shot,

do not consistently outperform simple baselines when predicting the effects of perturbations on 2,000 HVGs or the top 20 DEGs. The E-distance and contextual alignment metrics allow us to further contextualize our results and identify the failure modes of our models. We plan to maintain PertEval-scFM as an open and extensible benchmarking suite, designed to support diverse use cases and facilitate the evaluation of perturbation models.

**Future work.** Although our results highlight the limitations of using zero-shot scFM embeddings for perturbation effect prediction, we remain optimistic about the wider potential of scFMs. Key open questions include how to best represent perturbations *in silico* and how to fully leverage large-scale pre-training data to improve prediction accuracy. Existing cell atlases capture only a small fraction of the human phenoscape—the full range of states a cell can occupy (Fleck et al., 2023)—and often exclude perturbation-induced states. Moreover, specialized models must be designed to fully leverage large-scale datasets for predicting transcriptomic responses to perturbations. The superior performance of GEARS, which incorporates inductive biases tailored to perturbation prediction, exemplifies the importance of model architectures that explicitly encode relevant biological priors. Future work may benefit from exploring how such priors can be integrated into the scFM architecture to improve the utility of zero-shot representations of single-cell RNA-seq data for perturbation effect prediction.

## Computational Requirements

A single MLP probe requires 1 NVIDIA A100-PCIE-40GB GPU (using 12 cores) for training. Runtime depends on the hidden dimension of the probe, which is around 5 to 30 minutes for the smallest to biggest probes, respectively.

## Acknowledgments

The authors would like to thank Martino Mansoldo, Ionelia Buzatu, Pascal Notin, Andreas Bender, Kristof Szalay, Yuge Ji, Charlotte Bunne, Patrick Schwab, Djorde Miladinovic, Francesco Piatti, Michael Barnes, Claudia Cabrera, and Wei Wei for their helpful feedback and discussions.

A.W., M.O., W.T.D. and S.P.C. receive funding from the UKRI/BBSRC Collaborative Training Partnership in AI for Drug Discovery and Queen Mary University of London. N.B. and A.G.S. receive funding from the Wellcome Trust [218584/Z/19/Z].

This research utilised Queen Mary's Apocrita HPC facility, supported by QMUL Research-IT (King et al., 2017).

## Impact Statement

This paper aims to advance the field of machine learning by providing a standardized framework for benchmarking perturbation effect prediction models. Our work has the potential to improve the evaluation of computational models used in drug discovery and precision medicine by ensuring rigorous and reproducible assessment of their predictive capabilities. By highlighting the limitations of current zero-shot single-cell foundation models (scFMs) in perturbation effect prediction, we encourage future research to develop more robust and biologically informed architectures.

While our findings do not have immediate ethical concerns, the broader application of machine learning in biological and clinical settings requires careful consideration of biases in training data and the potential misinterpretation of model predictions. Ensuring that predictive models generalize across diverse biological contexts is crucial to avoiding misleading conclusions that could impact downstream biomedical applications. Future efforts should focus on increasing the diversity and representativeness of training datasets, as well as fostering transparency in model evaluation, to maximize the real-world utility of these approaches.

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

# Appendix

## A. Single-Cell Transcriptomics Data

The advent of single-cell RNA sequencing technology (scRNA-seq) has revolutionized our understanding of cellular heterogeneity and dynamic biological processes (Chen et al., 2019). Unlike traditional bulk sequencing methods, which average signals across large populations of cells, scRNA-seq technologies enable the study of gene expression at single-cell resolution. This granularity provides unprecedented insights into complex mechanisms of development, differentiation, and disease progression (Trapnell, 2015; Svensson et al., 2018; Fleck et al., 2023). The broad-scale application potential of scRNA-seq technology has led to the generation of large-scale datasets, such as the Human Cell Atlas (Regev et al., 2017) and the CellxGene Census (Program et al., 2023), which collectively span millions of cells and most sources of primary tissue.

### A.1. Perturb-Seq Data

Perturb-seq integrates scRNA-seq with CRISPR-based perturbations to profile gene expression changes in response to specific genetic modifications at the single-cell resolution (Dixit et al., 2016). By systematically perturbing genes and measuring the resulting transcriptomic changes, Perturb-seq data provides a detailed map of cellular responses to specific genetic modifications. These datasets, such as those generated by Norman et al. (2019) and Replogle et al. (2022), allow researchers to explore the relationships between gene perturbations and cellular phenotypes in a high-dimensional space, providing invaluable insights into gene regulatory networks and cellular behavior and allowing the identification of potential drug targets (Wenteler et al., 2024).

#### A.1.1. THE NORMAN DATASET

The dataset from Norman et al. (2019) represents one of the most comprehensive Perturb-seq resources available. It profiles transcriptional responses to 200 single-gene and double-gene perturbations in the human K562 leukemia cell line, using pooled CRISPRa screening and scRNA-seq. This dataset captures gene expression data from thousands of individual cells, each subjected to either a control or a perturbation, providing an ideal testing ground for models designed to predict perturbation effects. The Norman dataset includes both perturbed and unperturbed cells, allowing for systematic evaluation of model performance in predicting the effects of genetic perturbations at single-cell resolution.

*Table A1.* Overview of the Norman dataset

| Characteristic | Description |
| --- | --- |
| Cell type | K562 (human leukemia cells) |
| Total number of perturbations | 196 |
| Number of single-gene perturbations | 105 |
| Number of double-gene perturbations | 91 |
| Perturbation method | CRISPRa |
| Number of control cells | ∼12,000 |
| Number of cells | ∼110,000 |
| Sequencing platform | 10x Genomics Chromium |
| Gene expression data | Single-cell RNA-seq |
| Number of genes measured | 20,000+ |
| Reference | Norman et al. (2019) |

#### A.1.2. THE REPLOGLE DATASET

The datasets from Replogle et al. (2022) provide another comprehensive Perturb-seq resource, offering a large-scale map of transcriptional responses to pooled CRISPR-mediated perturbations. This datasets profile over 4000 genetic perturbations across multiple cell lines, including K562 (human leukemia) and RPE1 (human retinal pigment epithelial) cells, using scRNA-seq. Its expansive coverage of individual perturbations enables the evaluation of models designed to predict

transcriptional responses in diverse cellular contexts. The Replogle dataset captures single-cell gene expression data under a wide range of experimental conditions, providing an excellent benchmark for assessing the robustness and generalizability of predictive models. Furthermore, its inclusion of both perturbed and control cells facilitates systematic performance comparisons.

*Table A2.* Overview of the Replogle dataset

| Characteristic | Description |
|---|---|
| Cell types | K562 (human leukemia cells), |
| | RPE1 (human retinal pigment epithelial cells) |
| Total number of perturbations | 4,452 |
| Number of single-gene perturbations | 2,058 (K562), 2,394 (RPE1) |
| Perturbation method | CRISPRi |
| Number of cells | ∼310,000 (K562), ∼250,000 (RPE1) |
| Sequencing platform | NovaSeq 6000, Ultima Genomics |
| Gene expression data | Single-cell RNA-seq |
| Number of genes measured | 20,000+ |
| Reference | Replogle et al. (2022) |

### A.2. Single-Cell Data Pre-Processing and Quality Control

The datasets were downloaded and pre-processed using `ScPerturb` (Peidli et al., 2024), `PertPy` (Heumos et al., 2024), and `ScanPy` (Wolf et al., 2018). As scFMs utilize raw gene expression counts, two versions of the datasets are stored internally: an `AnnData` object containing raw expression counts, used to generate embeddings with scFMs, and an `AnnData` object with pre-processed gene expression values, used to train the baseline models.

Pre-processing involves normalizing the raw gene expression counts by the total number of counts for each gene to account for differences in sequencing depth and ensure comparability across samples. This was performed using the `scanpy.pp.normalize_total(adata)` method with default settings. Next, the normalized counts were log-transformed with `scanpy.pp.log1p(adata)` to stabilize variance and make the data more amenable to downstream analysis. Finally, the top 2,000 highly variable genes were selected for training, using the `scanpy.pp.highly_variable_genes(pert_adata, n_top_genes=2000)` function.

### A.3. Quality Control Plots

To ensure the robustness and reliability of analyses derived from the Norman and Replogle datasets used in this study, quality control (QC) steps were applied to evaluate the integrity and consistency of the data. These QC plots provide a visual assessment of key metrics such as the number of cells per gene and the number of genes detected per cell, both critical indicators of dataset quality.

The number of cells per gene (Figures A1a, A2a, A3a) reveals how consistently individual genes are captured across cells, identifying genes widely expressed (potentially housekeeping or essential genes) and filtering out low-quality genes only detected in a few cells. The number of genes detected per cell (Figures A1b, A2b, A3b) highlights the overall quality of cell-level data, with higher numbers indicating more comprehensive transcriptome coverage.

These distribution of perturbation effect plots (Figure A4) illustrate the range and variability of transcriptional changes induced by the perturbations, offering insights into the sensitivity of gene expression to these manipulations in each dataset.

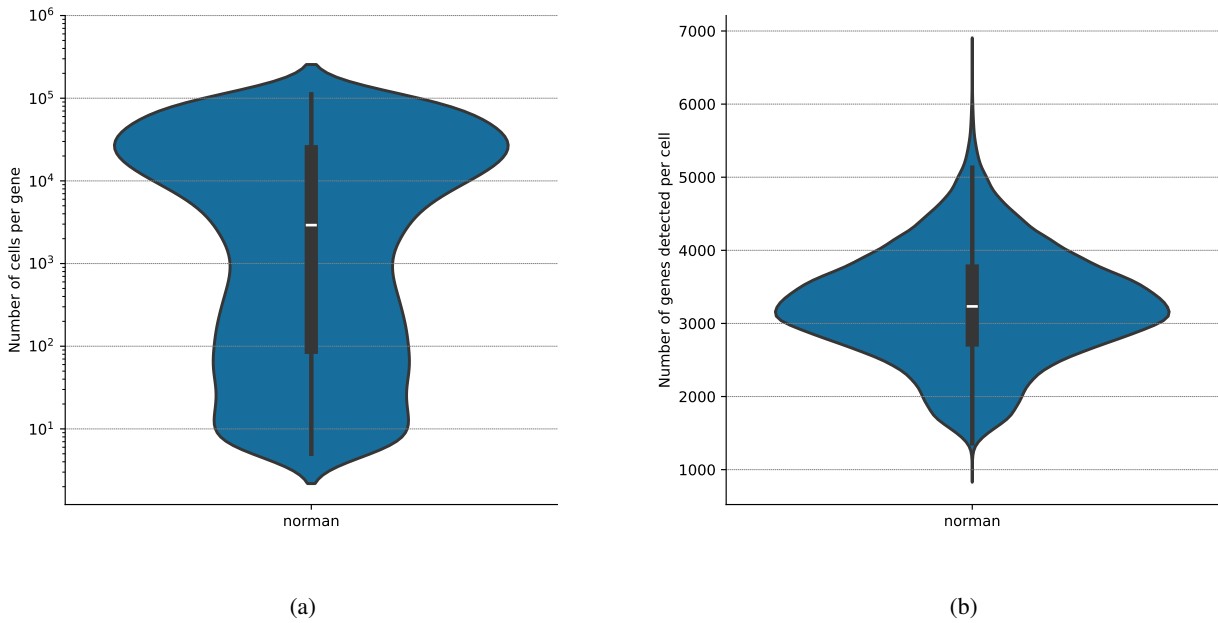

(a)                                                    (b)

*Figure A1.* Quality control plots for the Norman dataset. (a) The number of cells per gene. This indicates how often an individual gene is measured across cells. Genes that are present in many cells might be housekeeping genes or essential genes. Because many genes were present in only a few cells, only genes present in minimum 5 cells were considered. (b) The number of genes detected per cell across all datasets. This offers insights into the distribution of genes among cells and indicates how representative the measurements are of single-cell transcriptomes.

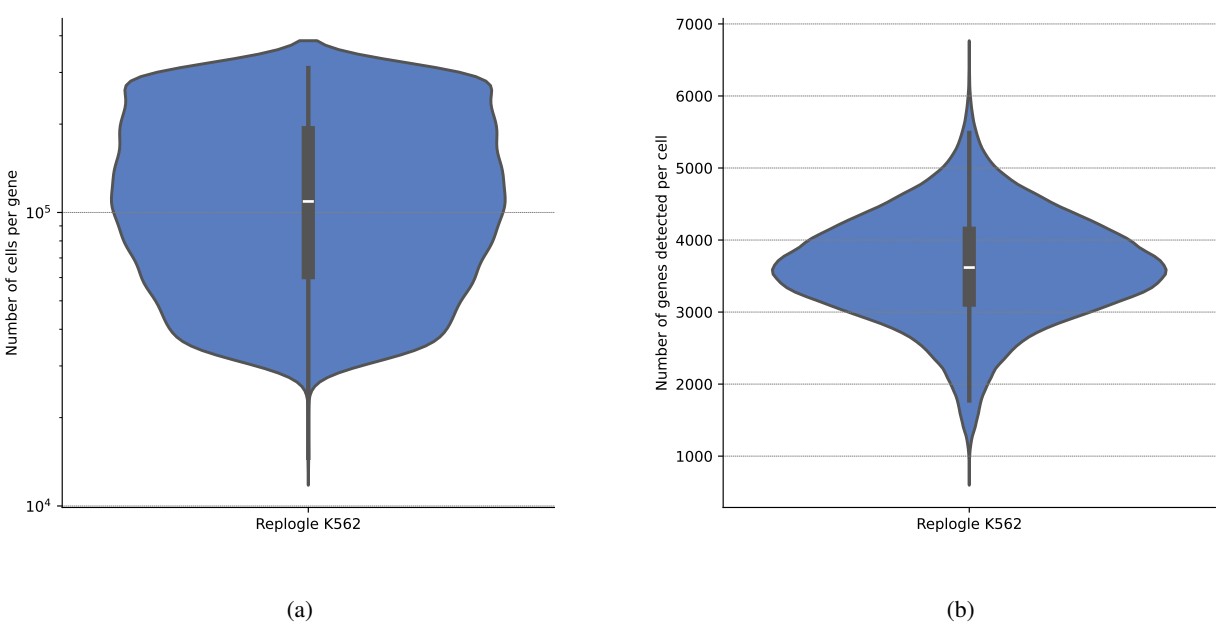

(a)                                                    (b)

*Figure A2.* Quality control plots for the Replogle K562 dataset. (a) The number of cells per gene. (b) The number of genes detected per cell across all datasets.

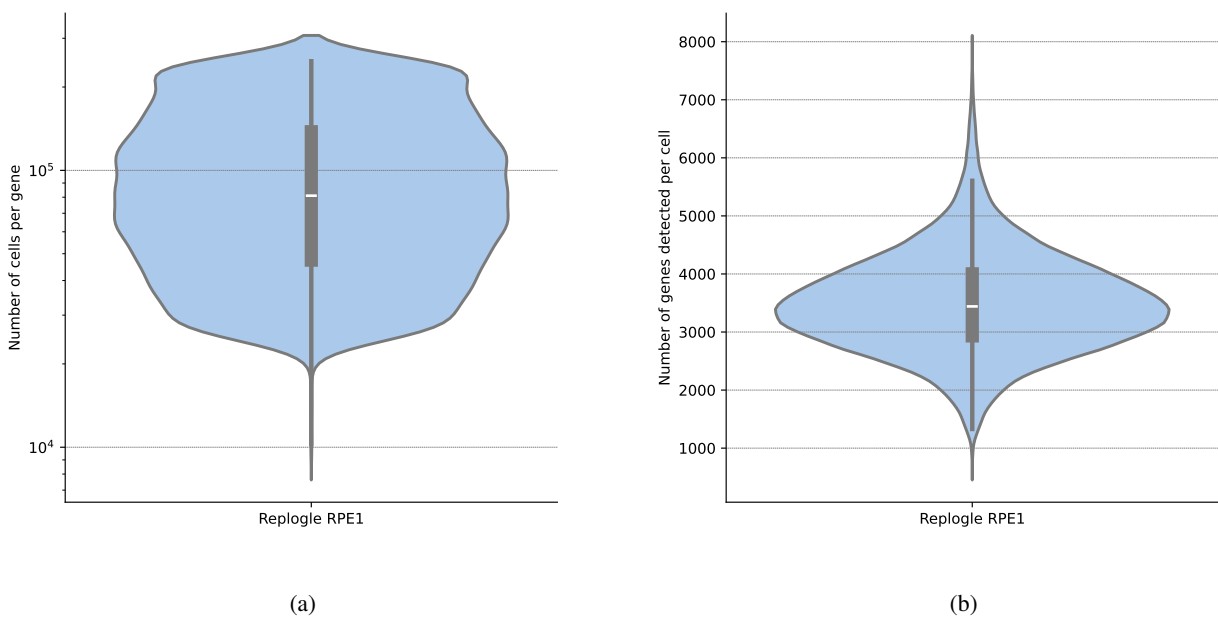

(a)  (b)

*Figure A3.* Quality control plots for the Replogle RPE1 dataset. (a) The number of cells per gene. (b) The number of genes detected per cell across all datasets.

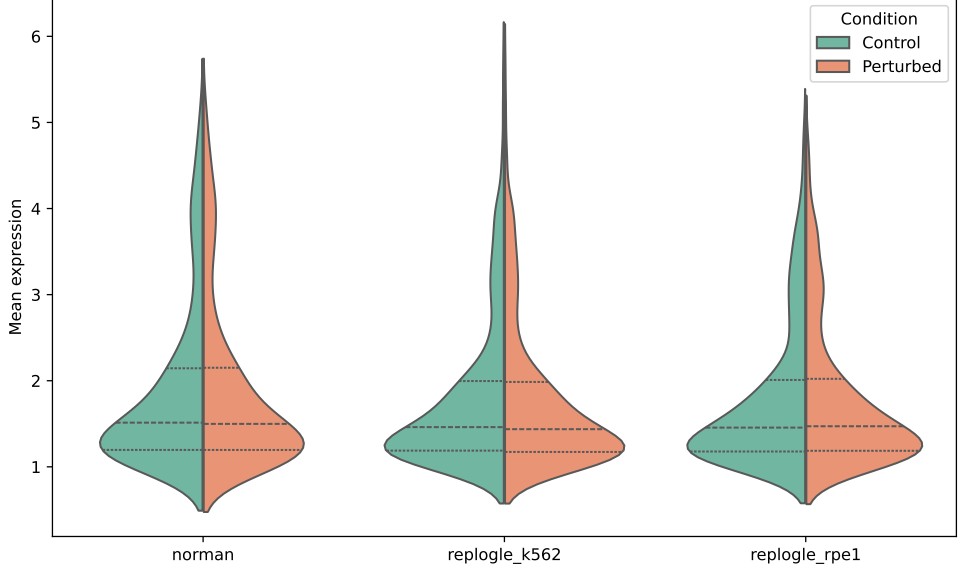

*Figure A4.* Overall distribution of mean expression in control and perturbed cells. The distributions are relatively similar for both conditions, indicating that most perturbations have a limited effect on the expression of most genes.

# B. Models

## B.1. Single-Cell Foundation Models (scFMs)

Single-cell foundation models (scFMs) are trained on broad single-cell data using large-scale self-supervision, allowing them to be adapted (i.e., fine-tuned) for a wide range of downstream tasks. Most scFMs use variants of the Transformer (Vaswani et al., 2017) architecture to process embedded representations of input gene expression data. However, they differ in input data representation, model architecture, and training procedures. Here, we provide a brief overview of the scFMs included in PertEval-scFM.

*Table B1.* Overview of the scFMs included in PertEval-scFM..

| MODEL NAME | ARCHITECTURE | PRE-TRAINING OBJECTIVE | # OF CELLS | ORGANISM | EMB. DIM. |
|---|---|---|---|---|---|
| SCBERT | TRANSFORMER | MASKED LANGUAGE MODELING (MLM) | ∼5 MILLION | HUMAN & MOUSE | 200 |
| GENEFORMER | TRANSFORMER | MASKED LANGUAGE MODELING (MLM) | ∼30 MILLION | HUMAN | 256 |
| SCGPT | TRANSFORMER | SPECIALIZED ATTENTION-MASKING MECHANISM | ∼33 MILLION | HUMAN | 512 |
| UCE | TRANSFORMER | MASKED LANGUAGE MODELING (MLM) | ∼36 MILLION | 8 SPECIES | 1,280 |
| SCFOUNDATION | TRANSFORMER | READ-DEPTH-AWARE (RDA) MODELING | ∼50 MILLION | HUMAN | 3,072 |

**Geneformer.** Geneformer (Theodoris et al., 2023) employs six transformer units, each consisting of a self-attention layer and an MLP layer. The model is pre-trained on Genecorpus-30M, which comprises 29.9 million human single-cell transcriptomes from a broad range of tissues obtained from publicly available data. Before feeding the data into the model, gene expression values are converted into rank value encodings. This method provides a non-parametric representation of each single-cell transcriptome by ranking genes based on their expression levels in each cell and normalizing these ranks within the entire dataset. Consequently, housekeeping genes, which are ubiquitously highly expressed, are normalized to lower ranks, reducing their influence. Rank value encodings for each single-cell transcriptome are then tokenized, allowing genes to be stored as ranked tokens instead of their exact transcript values. Only genes detected within each cell are stored, thus reducing the sparsity of the data. When input into the model, genes from each single-cell transcriptome are embedded into a 256-dimensional space. Cell embeddings can also be generated by averaging the embeddings of each detected gene in the cell, resulting in a 256-dimensional embedding for each cell. The model is pre-trained using a masked learning objective, masking a portion of the genes and predicting the masked genes, which is intended to allow the model to learn gene network dynamics.

**scBERT.** scBERT (Yang et al., 2022) adapts the BERT architecture (Devlin et al., 2019) for single-cell data analysis. A transformer is used as the model's backbone. The input data is represented as a sequence of gene expression values for each cell, where cells are constructed from gene expression value tokens. Gene embeddings are generated from the sum of two embeddings, where the first represents the gene's binned log-scale expression level, and the second is generated with gene2vec (Du et al., 2019) and specifies the gene's identity. The model is pre-trained via imputation on 5 million cells using a masked learning objective – masked gene expression values are predicted as a function of the other gene embeddings in the cell. In the paper, scBERT is fine-tuned for cell type annotation.

**scFoundation.** scFoundation (Hao et al., 2023) employs xTrimogene as a backbone model, a scalable transformer-based architecture that includes an embedding module and an asymmetric encoder-decoder. The embedding module converts continuous gene expression scalars into high-dimensional vectors, allowing the model to fully retain the information from raw expression values, rather than discretizing them like other methods. The encoder is designed to only process nonzero and nonmasked gene expression embeddings, reducing computational load and thus enabling the application of *"vanilla transformer blocks to capture gene dependency without any kernel of low-rank approximation"*. These encoded embeddings are then recombined with the zero-expressed gene embeddings at the decoder stage to establish transcriptome-wide embedded representations. This backbone approach can then be built upon additional architectures which are specialized for specific tasks - i.e., GEARS (Roohani et al., 2023) for perturbation response prediction. scFoundation is pre-trained using read-depth-aware (RDA) modeling, an extension of masked language modeling developed to take the high variance in read depth of the data into account. The raw gene expression values are pre-processed using hierarchical Bayesian downsampling in order to generate the input vectors, which can either be the unchanged gene expression profile or where downsampling has resulted in a variant of the data with lower total gene expression counts. After gene expression has been normalized, raw

and input gene expression count indicators are represented as tokens which are concatenated with the model input, allowing the model to learn relationships between cells with different read depths. Pre-training used data from over 50 million single cells sourced from a wide range of organs and tissues originating from both healthy and donors with a variety of diseases and cancer types.

**scGPT.** scGPT (Cui et al., 2024) follows a similar architectural and pre-training paradigm to scBERT. However, scGPT bins genes according to their expression, ensuring an even distribution across each bin. It uses random gene identity embeddings and incorporates an additional "condition embedding" to store meta-information and differentiate each gene. Along with gene embeddings, scGPT trains a cell token to summarize each cell. Instead of the long-range Performer architecture, scGPT processes embeddings via Flash-Attention (Dao et al., 2022) blocks. The model implements a generative masked pre-training using a causal masking strategy inspired by OpenAI's GPT series (Radford et al., 2018). scGPT is pre-trained on 33 million human cells and fine-tuned on a wide suite of downstream tasks, including cell type annotation, genetic perturbation response prediction, batch correction, and multi-omic integration.

**Universal Cell Embeddings (UCE).** Universal Cell Embeddings (UCE) (Rosen et al., 2023) is trained on a large compendium of single-cell RNA-seq datasets from multiple species, including human, mouse, mouse lemur, zebrafish, pig, rhesus macaque, crab-eating macaque, and western clawed frog, to create a universal embedding space for cells. The model converts the transcriptome of a single cell into an expression-weighted sample of its corresponding genes and then represents these genes by their protein products using a large protein language model. This representation is then fed into a transformer model. UCE is pre-trained in a self-supervised manner with a contrastive learning objective, where similar cells are mapped to nearby points in the embedding space, and dissimilar cells are mapped to distant points. This training paradigm enables UCE to provide high-quality embeddings that facilitate various downstream analyses. Benchmarks carried out by Rosen et al. (2023) in a zero-shot framework shown that UCE outperforms Geneformer (Theodoris et al., 2023) and scGPT (Cui et al., 2024), as well as cell annotation models such as scVI and scArches, in cell representation tasks.

### B.2. scFM Embedding Generation

In this section, we detail the process of generating embeddings for each foundation model in a zero-shot context using pre-trained models with frozen weights. For some models, pre-trained checkpoints are available and can be directly utilized, while others require initial pre-training. By freezing model weights, we ensure that the embeddings represent the learned features from the initial training phase, without further adaptation to the specific perturbation prediction task. This approach allows us to evaluate the inherent quality and utility of the pre-trained representations for downstream applications in biological research.

**Geneformer.** To generate embeddings for Geneformer (Theodoris et al., 2023), we downloaded the repository, including pre-trained model checkpoints, from Hugging Face. For control cells, we pre-processed the raw expression files to ensure the correct naming of columns and then fed them into the Geneformer tokenizer (`TranscriptomeTokenizer`). Once the dataset had been tokenized, we extracted embeddings using the pre-trained checkpoint (6-layer model) with the `EmbExtractor` method. For the perturbation data, we loaded the data and iterated through it in order to remove perturbed genes, simulating their deletion. The perturbed cells were then tokenized, and embeddings were extracted for each perturbed cell using the same functions.

**scBERT.** To generate emeddings for scBERT (Yang et al., 2022), we first downloaded the checkpoint and data shared in the scBERT GitHub repository. The environment was set up using the scBERT-reusability GitHub repository. For the raw expression counts, the genes were aligned using Ensembl *Homo sapiens* gene information. Log-normalization was performed and cells with less than 200 expressed genes were filtered out. For the perturbation data, the gene expression value was set to 0 to simulate perturbation, and embeddings were generated using the `predict.py` script.

**scFoundation.** To generate scFoundation embeddings (Hao et al., 2023), we initialized the scFoundation class shared at the official scFoundation GitHub repository. The `01B-resolution` pre-trained model checkpoint was loaded and the embeddings were generated while setting the `input_type = singlecell` and `tgthighres = f1` to indicate no read depth differences between unperturbed and perturbed cells. The embeddings were then generated using the `get_embeddings` function.

**scGPT.** To generate embeddings for scGPT (Cui et al., 2024) we installed the scGPT python package. We downloaded and used the whole-human scGPT model for embedding. For control cells, we used the scGPT `embed_data` function to generate the embeddings from the raw expression values. This function tokenises the data before feeding it through the model. For the perturbation data, we removed the perturbed genes, to simulate their deletion. The embeddings for the perturbed cells were then generated using the scGPT `embed_data` function.

**Universal Cell Embeddings (UCE).** To generate cell embeddings for UCE (Rosen et al., 2023), we ran the `eval_single_anndata.py` script provided in the UCE GitHub repository. Model weights for the 33-layer model and the pre-computed protein embeddings were downloaded separately from figshare. The script takes as input an h5ad raw expression file with variable names set as gene_symbols. The script was run with default parameters, except for the filter argument which was set to `False`, in order to skip an additional gene and cell filtering step. No further pre-processing was required to generate embeddings for control cells. For *in vitro* perturbed cells, the raw count value of the perturbed gene was explicitly set to zero for each condition prior to model inference, and saved as a h5ad file. The output of the script was an identical h5ad file with the input, except for cell-level embeddings that are stored in the `Anndata.obsm['X_uce']` slot.

# C. Featurization

## C.1. Single-Cell Expression Data Featurization

To generate the input features for raw single-cell expression data, we begin with the control matrix $C \in \mathbb{R}^{n_c \times v}$, consisting of $n_c$ unperturbed single-cell transcriptomes across $v$ highly variable genes (see Appendix A.2). From this matrix, we form a pseudo-bulk sample $\widetilde{C}$, which aggregates expression values from groups of cells within the same sample, in order to reduce sparsity and noise. Formally, let $\widetilde{C} = \{\mathbf{c}_i\}_{i=1}^{500}$ denote the set of randomly sampled cells from $C$. The average expression value $\overline{C}_j$ for each cell $j$ is then calculated by averaging the expression across the pseudo-bulked cells:

$$\overline{C}_j = \frac{1}{|\widetilde{C}|} \sum_{\mathbf{c}_i \in \widetilde{C}} c_{i,j} \quad \forall\, j \in \{1, \ldots, n_p\} \tag{C1}$$

Using this basal expression, we construct the input matrix $X_c \in \mathbb{R}^{n_p \times v}$, which has the same dimensions of the perturbed transcriptomic matrix $P \in \mathbb{R}^{n_p \times v}$ (i.e. what we want to predict), where $n_p$ is the number of perturbed cells. The input matrix $X_c$ is generated by sub-sampling from $\overline{C}_j$, ensuring that the dimensions are consistent between the input and the target output.

This approach ensures that input-target pairs are consistently defined for all training examples, as the dimensions of $X_c \in \mathbb{R}^{n_p \times v}$ align with the target matrix $P$. Representing input expression at pseudo-bulked basal levels helps mitigate sparsity issues caused by limited gene coverage in individual single-cell measurements from the original dataset. However, this method introduces a trade-off by reducing the heterogeneity of the input gene expression. As a result, some salient single-cell signals, such as those related to its initial state, may be diminished. However, inferring cellular states based solely on gene expression data is inherently challenging, given the many confounding factors and technical noise present in single-cell datasets (Fleming et al., 2023). Therefore, conventional machine learning models should not be expected to perform this task with high fidelity to begin with.

### C.1.1. MLP BASELINE

To generate the full set of input features for the MLP, we must encode the identity of each perturbation alongside capturing basal gene expression. Let $\mathcal{P} = \{p_1, \ldots, p_k\}$ denote the set of *perturbable* genes, and let $\mathcal{D} = \{d_1, \ldots, d_v\}$ represent all highly variable genes.

To evaluate the models' ability to generalize to unseen perturbations, it is important to incorporate information about gene interactions within a specific cell type. This allows the models to learn gene interaction networks, helping to extrapolate effects from known perturbations to novel ones.

To achieve this, we construct a $v$-dimensional correlation vector for each perturbable gene by calculating the Pearson correlation between its basal expression and that of all other genes, including itself. By including the auto-correlation of the perturbable gene, we explicitly encode the identity of the gene to be perturbed. The resulting feature vector for each perturbable gene, $\mathbf{g}_c \in \mathbb{R}^v$, captures the correlations between its basal expression and the basal expression of all highly variable genes. Aggregating these correlation vectors for all perturbable genes produces the matrix $G_c \in \mathbb{R}^{n_p \times v}$, where the perturbation in each row corresponds to the transcriptomic state observed in $T$.

Finally, the control gene expression matrix $X_c$ is concatenated with the perturbation correlation matrix $G_c$ to construct the complete input feature matrix:

$$Z_{\text{GE}} = X_c \oplus G_c \tag{C2}$$

Here, $Z_{\text{GE}} \in \mathbb{R}^{n \times 2v}$ represents the input feature matrix, where each row $\mathbf{g}_i$ combines the log-normalized basal expression values of a cell with the corresponding perturbation correlation features. This procedure is applied to both the training and testing sets, to generate $Z_{\text{GE}_{\text{train}}}$ and $Z_{\text{GE}_{\text{test}}}$.

## C.2. Single-Cell Foundation Model Embedding Featurization

To generate embeddings from a pre-trained single-cell foundation model (scFM) with frozen weights, we begin by mapping raw gene expression counts to transcriptomic embeddings. Let $f_{\text{scFM}} : \mathbb{R}^l \to \mathbb{R}^{e_c}$ represent the function that transforms raw

expression data into an embedding for each cell.

To construct the control cell embedding, we feed the raw expression vector $\mathbf{x}_i^c$ for each of the $n_c$ control cells into the scFM:

$$f_{\text{scFM}}(X_c) = Z_c \tag{C3}$$

The embedding vectors are then subsampled to create $\overline{Z}_c \in \mathbb{R}^{n_p \times e_c}$, where $n_p$ matches the number of perturbed cells and the dimension of $\overline{Z}_c$ aligns with the target output matrix.

An *in silico* perturbation embedding is then generated by nullifying the expression of the perturbed genes across all control cells in which it is expressed, up to a maximum of 500 cells. The nullification process, denoted by $N(\mathbf{x}_i^c, p_i)$, adjusts the gene expression vector according to the requirements of the scFM model in use. The nullification function can be defined as $N : \mathbb{R}^v \times \mathbb{N}_v \to \mathbb{R}^l$, where $\mathbb{R}^v$ represents the space of the gene expression vector, and $\mathbb{N}_v$ denotes the set of natural numbers from 1 to $v$, corresponding to the indices of genes in $\mathbf{x}_i^c$. If the scFM requires setting the perturbed gene's expression to zero, $l = v$. However, some scFMs filter out non-expressed genes during tokenization (scGPT), or train on ranked gene token representations instead of expression values (Geneformer). In these cases, the perturbed gene must be removed from the control gene expression vector, resulting in $l = v - 1$. Nonetheless, the perturbation embedding $\mathbf{x}_i^p$ is constructed as follows:

$$f_{\text{scFM}}(N(\mathbf{x}_i^c, p_i)) = \mathbf{z}_i^p \tag{C4}$$

The perturbation embeddings for all cells form the matrix $Z_p \in \mathbb{R}^{n_p \times e_c}$. It is trivial to extend the above framework to combinatorial perturbations, where the nullification function accepts multiple perturbations and nullifies the associated gene expression values.

The final cell embedding is then obtained by concatenating the control embedding $\overline{Z}_c$ with the perturbation embedding $Z_p$:

$$Z_{\text{scFM}} = \overline{Z}_c \oplus Z_p \tag{C5}$$

This approach differs from raw expression featurization, where co-expression patterns are explicitly encoded to model perturbations. In the scFM embedding featurization, *in silico* perturbation simulates the changes caused by gene perturbation. We hypothesize that the embeddings generated by scFMs inherently encode co-expression relationships, aligning with their pre-training objective based on masked language modeling.

In this study, zero-shot embeddings are generated using five different scFMs (Table B1). Inference for each scFM is tailored to the specific idiosyncrasies of the model in question. Detailed information on all the scFMs used can be found in Appendix B.1.

# D. Perturbation Representation Experiment

## D.1. Doubling Gene Expression Perturbation Representation Results

Here we assess whether simulating perturbations via an *in silico* knockout (scBERT−) versus an upregulation strategy (scBERT+) leads to appreciable differences in embedding quality on the Norman CRISPRa dataset. By applying each perturbation scheme for embedding generation using scBERT, and computing the mean squared error (MSE) of the predicted post-perturbation expression across a range of sparsification probabilities ($S = 0.1$–$0.7$), we directly compare knockout against a gain-of-function representation obtained by doubling the perturbed gene's expression. As shown in Table D1 and Figure D1, both approaches yield nearly identical performance, indicating that our knockout-based perturbation protocol provides a fair and robust benchmark for scFM comparison while avoiding the arbitrary magnitude choices and re-ranking challenges inherent to upregulation simulations.

*Table D1.* MSE $\pm$ standard deviation for Norman single-gene embeddings generated with scBERT using an *in silico* knockout (scBERT−) vs. an upregulation strategy (scBERT+) across sparsification probabilities.

| MODEL | $S$ 0.1 | $S$ 0.2 | $S$ 0.3 | $S$ 0.4 | $S$ 0.5 | $S$ 0.6 | $S$ 0.7 |
|---|---|---|---|---|---|---|---|
| scBERT− | $0.0630 \pm 0.0031$ | $0.0634 \pm 0.0062$ | $0.0676 \pm 0.0051$ | $0.0636 \pm 0.0031$ | $0.0592 \pm 0.0015$ | $0.0645 \pm 0.0107$ | $0.0849 \pm 0.0060$ |
| scBERT+ | $0.0640 \pm 0.0038$ | $0.0620 \pm 0.0038$ | $0.0658 \pm 0.0077$ | $0.0610 \pm 0.0009$ | $0.0659 \pm 0.0046$ | $0.0744 \pm 0.0122$ | $0.0853 \pm 0.0064$ |

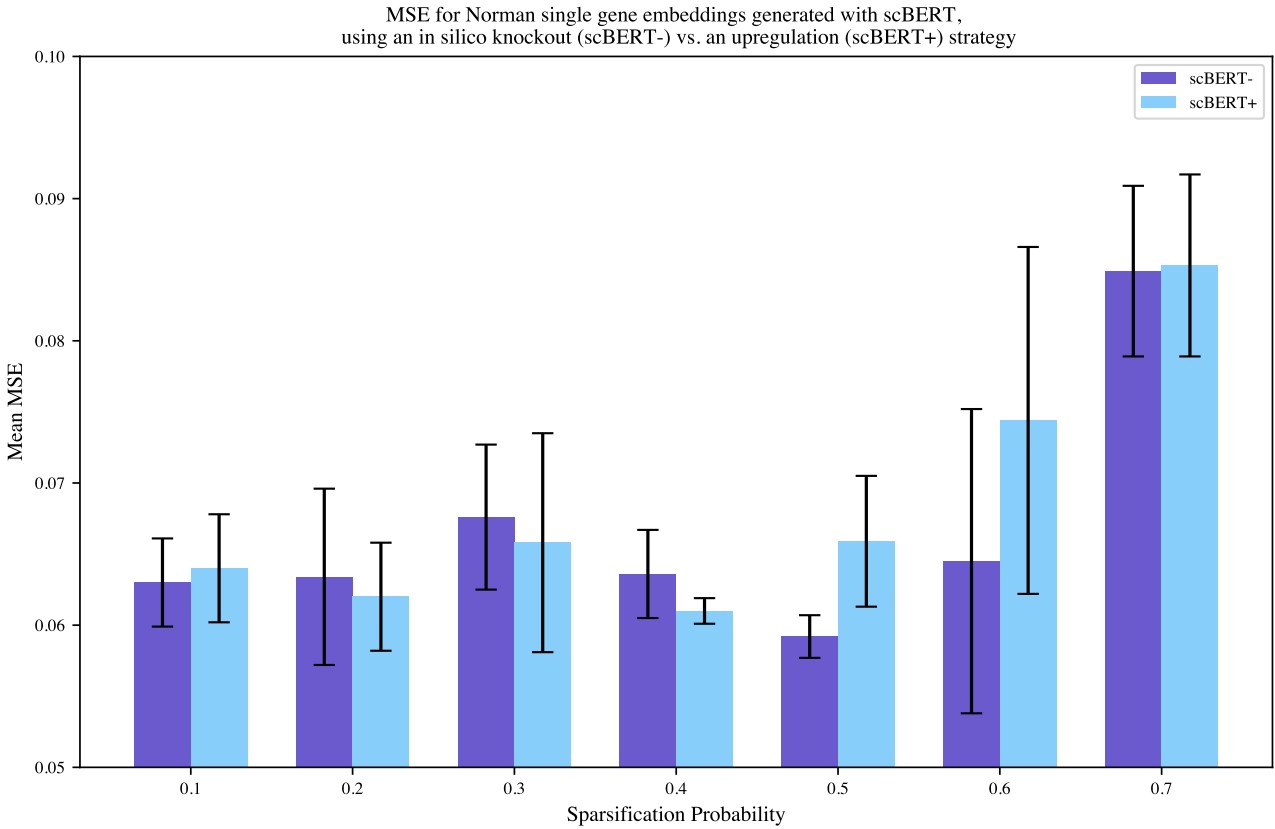

*Figure D1.* MSE for Norman single-gene embeddings generated with scBERT using an *in silico* knockout (scBERT−, purple) vs. an upregulation strategy (scBERT+, light blue) across sparsification probabilities. Error bars represent standard deviations.

# E. MLP

## E.1. MLP Parameter Count

The architectural capacity of neural networks, as reflected by the number of trainable parameters, can significantly impact their performance. This is particularly relevant when probing embeddings with variable dimensions, such as those derived from single-cell foundation models (scFMs). To explore the relationship between parameter count and model performance, we trained MLPs of varying sizes on raw gene expression data. This allowed us to evaluate whether increasing the expressiveness of the MLP affects its ability to capture biologically relevant patterns. The results, summarized in Table E2, demonstrate that parameter count has negligible influence on the mean squared error (MSE) for both training and validation datasets. Similar experiments conducted using scFoundation and scBERT embeddings reinforce these findings. Notably, despite increasing the parameter count, the scFM embeddings do not provide additional biologically relevant information for enhanced performance. These results suggest that embedding sizes do not inherently constrain the MLP probes and that the observed performance differences across models are unlikely to be influenced by disparities in architectural capacity.

*Table E2.* Train and test set results with MLPs of increasing parameter count

| Trainable parameters (million) | Training data | train/MSE | val/MSE |
|---|---|---|---|
| 1.6 | Raw gene expression | 0.057067 | 0.057642 |
| 3.2 | Raw gene expression | 0.058670 | 0.057493 |
| 6.3 | Raw gene expression | 0.056748 | 0.057424 |
| 12.7 | Raw gene expression | 0.056724 | 0.057428 |
| 1.6 | scFoundation embeddings | 0.060780 | 0.060260 |
| 3.2 | scFoundation embeddings | 0.060440 | 0.059910 |
| 12.6 | scFoundation embeddings | 0.059570 | 0.059050 |
| 0.2 | scBERT embeddings | 0.061040 | 0.061426 |
| 1.0 | scBERT embeddings | 0.061046 | 0.061428 |
| 8.0 | scBERT embeddings | 0.061040 | 0.061421 |

## E.2. Hyperparameter Optimization

To train the MLP probes, we used root mean square error (RMSE) as the objective function and the Adam optimizer (Kingma & Ba, 2017). Model performance was evaluated on an independent test set comprising unseen perturbations. The objective function to be minimized is:

$$\mathcal{L}(\theta) = \sqrt{\frac{1}{n_b} \sum_{j=1}^{n_b} \left( (T - X_c)_j - \hat{\delta}^\theta(X)_j \right)^2} \tag{E1}$$

where $j$ indexes each cell and $n_b$ denotes the batch size.

Hyperparameters were selected using the tree-structured Parzen estimator (TPE) tuning algorithm (Bergstra et al., 2011). This optimization was performed on the first train-test split, which contains the largest training set. Given the computational demands of exhaustive parameter sweeps, we focused on optimizing the hyperparameters using the gene expression data as a reference.

An initial search across different numbers of hidden layers revealed that this parameter had no substantial effect on model performance. Therefore, a single hidden layer was used throughout the experiments to maintain model simplicity. The learning rate, however, was found to significantly influence performance and was thus adjusted for the models trained using the scFM embeddings. Following the manifold hypothesis, we set the hidden dimension to half of the input dimension (Bengio et al., 2013).

# F. SPECTRA

The SPECTRA framework addresses a critical gap in the evaluation of machine learning models for biological tasks, particularly in the context of single-cell datasets like Perturb-seq, where distribution shifts are common. Unlike random splits, which may either overestimate or underestimate model performance due to unquantified train-test similarity, SPECTRA systematically controls and quantifies train-test dissimilarity. This enables a nuanced assessment of model robustness across a spectrum of difficulty levels, including challenging scenarios that better simulate real-world variability.

By generating train-test splits based on a defined sparsification probability, SPECTRA not only evaluates models under diverse conditions but also provides a framework to contextualize performance results and compare robustness across datasets and tasks. Importantly, SPECTRA's methodology does not artificially inflate task difficulty; instead, it highlights inherent redundancies and patterns within the dataset. This approach ensures that performance assessments reflect meaningful generalization.

Moreover, SPECTRA is not just an evaluation tool but a valuable guide for experimental design. For example, in Perturb-seq, SPECTRA can reveal if a model generalises well within pathways but struggles across pathways, guiding researchers on which genes to test next, optimizing experimental efforts. Random splits, by contrast, obscure such insights due to high train-test similarity.

SPECTRA's ability to expose weaknesses in generalizability, simulate biologically realistic distribution shifts, and guide experimental design makes it an high-impact addition to evaluation pipelines. This is particularly crucial for tasks like perturbation effect prediction, where navigating complex, real-world variability is a fundamental challenge.

## F.1. Evaluating Model Robustness under Distribution Shift in Single-Cell Data with SPECTRA

To construct the spectral graph for single-cell data, sample-to-sample similarity between distributions is calculated using the L2 norm, denoted by $\| \cdot \|$, of the $\log 1p$-fold change between the mean perturbation expression vector, $\overline{\mathbf{p}}_i$, and the mean control gene expression vector, $\overline{\mathbf{c}}$:

$$S(\overline{\mathbf{p}}_i, \overline{\mathbf{c}}) = \| \log(\overline{\mathbf{p}}_i + 1) - \log(\overline{\mathbf{c}} + 1) \| \tag{F1}$$

If two samples are sufficiently similar, an edge will be inserted in the spectral graph. A series of train-test splits are then generated by sparsifying the initial graph. Train and test instances are sampled from distinct subgraphs for each split, with decreasing mean pairwise similarity between the two sets. The sparsification of the initial graph is attenuated by a *sparsification probability* ($s$), which is the probability that an edge between two samples will be be dropped. Mathematically, SPECTRA employs a graph sparsification technique similar to what is described in Spielman & Teng (2010). A practical limitation of the current implementation of SPECTRA lies in its tendency to unevenly distribute perturbations of similar magnitudes across the training and test splits while minimizing cross-split overlap. This uneven distribution engenders class imbalances that become increasingly pronounced at higher sparsification probabilities. Consequently, this imposes a trade-off between induced class imbalance and simulated distribution shift. Empirical observations on our datasets indicate that the sparsification probability threshold at which the class imbalance remains manageable is approximately 0.7. Beyond this threshold, the deleterious effects of class imbalance as well as low sample numbers begin to outweigh the benefits of reduced cross-split overlap.

Indeed, for the Norman dataset, Figure F1a illustrates a rapid decrease in the number of training and testing samples as the sparsification probability increases. This is expected, as a higher sparsification probability leads to increasingly disconnected subgraphs to draw samples from. Furthermore, Figure F1b confirms that SPECTRA can simulate distribution shift by showing a corresponding decrease in similarity between the samples as sparsification probability rises. Subsequently, we train and test models on each SPECTRA split and plot the MSE as a function of the decreasing cross-split overlap. The area under this curve is defined as the AUSPC, which serves as a measure of model generalizability under distribution shift.

Similarly to the within-dataset case outlined above, the cross-split overlap can be used to measure the similarity between-datasets, in this case between the scFM pre-train and our fine-tune datasets for scBERT and scGPT. This approach allows us to investigate the impact of pre-training data on the quality of scFM embeddings. Further details are provided in Section H.1.

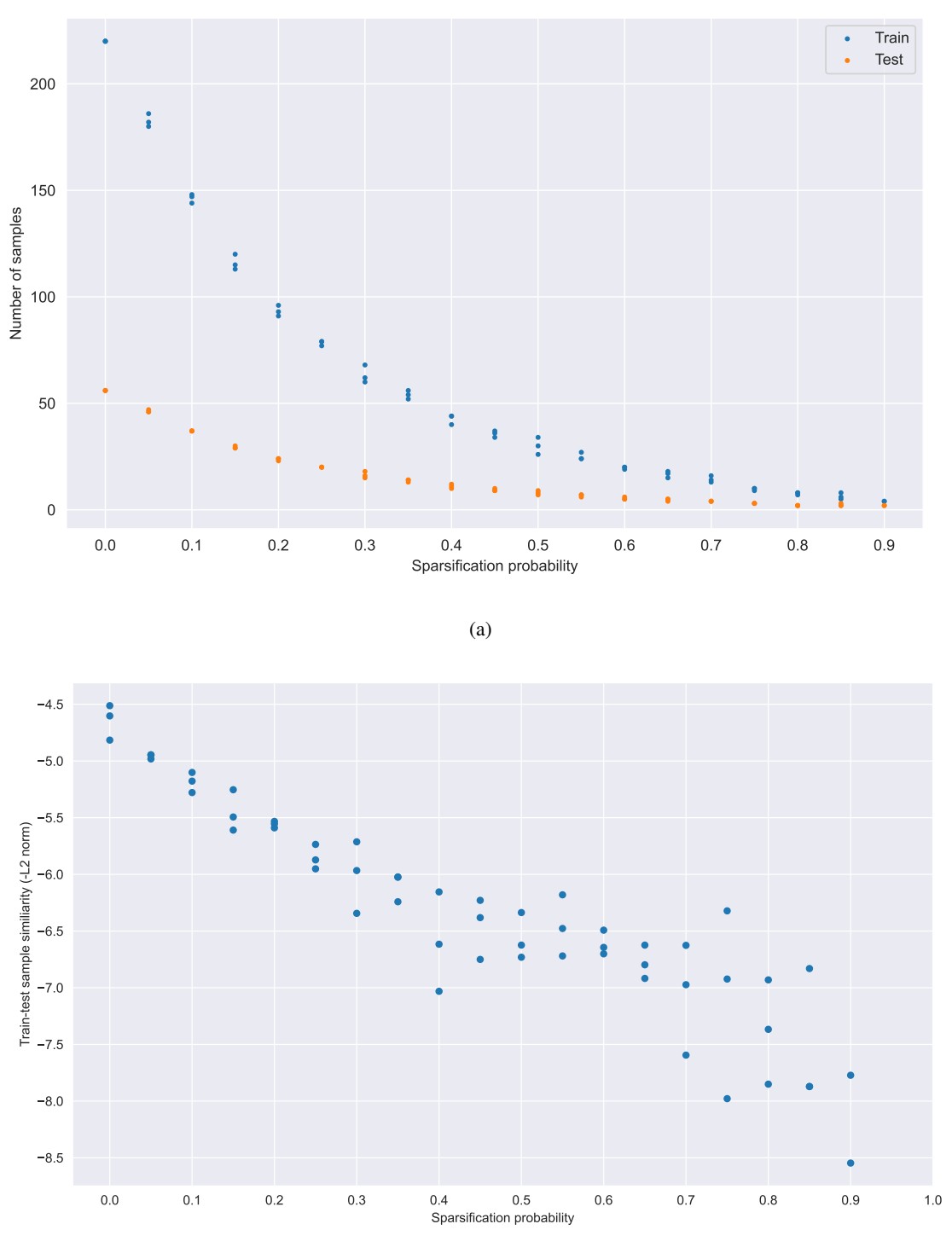

(a)

(b)

*Figure F1.* (a) Number of samples in train and test as a function of the sparsification probability. (b) Cross-split overlap as a function of the sparsification probability.

## F.2. Implementation Details of the AUSPC

The AUSPC is defined by Equation 7. For numerical evaluation, the integral is approximated using the trapezoidal rule with sparsification probabilities $s_i \in \{0.1, 0.2, ..., 0.7\}$:

$$
\begin{aligned}
\text{AUSPC} = f(\phi) &= \int_0^{s_{\max}} \phi(s) \, ds \\
&\approx \frac{d}{2} \sum_{i=0}^{n-1} [\phi(s_i) + \phi(s_{i+1})]
\end{aligned}
\tag{F2}
$$

where $d$ denotes the step size of the sparsification probability (0.1 in this case) and $\phi$ represents the metric of interest, (MSE). The $\Delta\text{AUSPC}$ is subsequently derived by calculating this value for both the baseline and the model independently, and then subtracting the AUSPC of the model from that of the baseline. For simplicity, we use the notation $\phi_i = \phi(s_i)$.

To quantify the uncertainty associated with the AUSPC, uncertainty propagation is utilised, wherein the AUSPC is assumed to be a non-linear function of the metric of interest, $\phi(s)$. For uncertainty propagation in this context, the following equation is employed:

$$
\sigma^2 = \sum_{i=1}^{n-1} \left( \frac{\partial f}{\partial \phi_i} \sigma_{\phi_i} \right)^2
\tag{F3}
$$

where $\sigma$ represents the total error associated with the AUSPC and $\sigma_{\phi_i}$ denotes the error associated with the MSE for split $i$.

The partial derivative $\frac{\partial f}{\partial \phi_i}$ is calculated using the definition of $f$ given in Equation F2:

$$
\begin{aligned}
\frac{\partial f}{\partial \phi_i} &= \frac{d}{2} \sum_i \left( \frac{\partial}{\partial \phi_i} \phi_i + \frac{\partial}{\partial \phi_i} \phi_{i+1} \right) \\
\frac{\partial f}{\partial \phi_i} &= \frac{d}{2}
\end{aligned}
\tag{F4}
$$

Substituting this result into Equation F3 yields:

$$
\begin{aligned}
\sigma^2 &= \sum_i \left( \frac{d}{2} \sigma_{\phi_i} \right)^2 \\
\sigma &= \sqrt{\sum_i \left( \frac{d}{2} \sigma_{\phi_i} \right)^2}
\end{aligned}
\tag{F5}
$$

The algorithmic implementation is given in Algorithm 1.

---

**Algorithm 1** Calculate AUSPC and its associated error

---

1: **function** TRAPEZOIDALAUSPC($\phi$, $s$)
2:      AUSPC $\leftarrow$ `np.trapz`($\phi$, $s$)
3:      **return** AUSPC
4: **end function**
5: **function** CALCULATEDELTAAUSPC($\phi_b$, $\phi_m$, $\sigma_b$, $\sigma_m$, $s$)
6:      AUSPC$_b$ $\leftarrow$ TRAPEZOIDALAUSPC($\phi_b$, $s$)
7:      AUSPC$_m$ $\leftarrow$ TRAPEZOIDALAUSPC($\phi_m$, $s$)
8:      $d \leftarrow s[1] - s[0]$                  $\triangleright$ Assuming uniform step size
9:      $\sigma_b \leftarrow \sqrt{\sum_i (\frac{d}{2} \sigma_{\phi_b, i})^2}$
10:     $\sigma_m \leftarrow \sqrt{\sum_i (\frac{d}{2} \sigma_{\phi_m, i})^2}$
11:     $\Delta$AUSPC $\leftarrow$ AUSPC$_b$ $-$ AUSPC$_m$
12:     **return** $\Delta$AUSPC, $\sigma_b$, $\sigma_m$
13: **end function**

---

# G. E-Statistics

## G.1. Using E-Distance and Differential Gene Expression Analysis to Evaluate Significant Perturbations

While examining transcriptome-wide, aggregated perturbation effects provides valuable insights, it lacks the granularity needed to assess a model's ability to reconstruct perturbation effects at the gene level. To address this limitation, energy statistics (E-statistics) are employed to evaluate and select significant perturbations in single-cell expression profiles. Subsequently, differential gene expression analysis is carried out to identify the top 20 differentially expressed genes which are then used to evaluate individual perturbations.

Perturbation effects are quantified using the E-distance, which compares mean pairwise distances between perturbed and control cells. Let $\mathcal{X} \in \{\mathbf{x_1}, \ldots, \mathbf{x}_{n_a}\}$ and $\mathcal{Y} \in \{\mathbf{y_1}, \ldots, \mathbf{y}_{n_b}\}$ be two distributions of cells in different conditions with $n_a$ and $n_b$ cells respectively, where $\mathbf{x}_i, \mathbf{y}_i \in \mathbb{R}^m$ refer to the transcriptomes for cell $i$. Now the between-distribution distance $\delta_{\mathcal{X}\mathcal{Y}}$ and the within-distribution distances $\sigma_{\mathcal{X}}$ and $\sigma_{\mathcal{Y}}$ can be defined as:

$$
\delta_{\mathcal{X}\mathcal{Y}} = \frac{1}{n_a \cdot n_b} \sum_{i=1}^{n_a} \sum_{j=1}^{n_b} d(\mathbf{x}_i, \mathbf{y}_j)
$$
$$
\sigma_{\mathcal{X}} = \frac{1}{n_a^2} \sum_{i=1}^{n_a} \sum_{j=1}^{n_a} d(\mathbf{x}_i, \mathbf{x}_j) \tag{G1}
$$
$$
\sigma_{\mathcal{Y}} = \frac{1}{n_b^2} \sum_{i=1}^{n_b} \sum_{j=1}^{n_b} d(\mathbf{y}_i, \mathbf{y}_j)
$$

where $d(\cdot, \cdot)$ is the squared Euclidean distance. The E-distance, $E$, is then defined as:

$$
E(\mathcal{X}, \mathcal{Y}) := 2\delta_{\mathcal{X}\mathcal{Y}} - \sigma_{\mathcal{X}} - \sigma_{\mathcal{Y}} \tag{G2}
$$

The E-test, a Monte Carlo permutation test, is used to assess the statistical significance of observed E-distances. This test generates a null distribution by randomly permuting perturbation labels 10,000 times, comparing the observed E-distance against this distribution to yield an adjusted p-value, calculated using the Holm-Sidak method. This p-value can then be used to select which perturbations result in a perturbation effect that is significantly different from the control.

Before E-statistics are calculated, the data is pre-processed. The number of cells per perturbation is subsampled to 300, following the 200-500 range proposed by Peidli et al. (2024). Perturbations with fewer than 300 cells are excluded from downstream analysis. For the Norman dataset, this threshold excludes 20 perturbations, leaving 186 perturbations. One additional perturbation (*BCL2L11*) is excluded by the E-test as not significant.

For significant perturbations, the top 20 differentially expressed genes between perturbation and control are selected for evaluation. This approach is based on the observation that genetic perturbations tend to significantly affect only a fraction of the full transcriptome, while the remainder remains close to control expression (Nadig et al., 2024). This allows us to evaluate whether the predicted perturbation effect aligns with the experimental observations specifically for individual perturbations. The data is pre-processed for differential gene expression testing as described in Appendix A.2. Differential gene expression calculation is performed using the Wilcoxon rank sum test implemented in `scanpy.tl.rank_gene_groups`.

# H. Contextual Alignment

## H.1. Calculating Contextual Alignment between Pre-Train and Fine-Tune Datasets

To evaluate the influence of pre-training on the efficacy of scFM embeddings, we estimate the contextual alignment between the datasets used for pre-training and those used for fine-tuning. We expect that enhanced model performance correlates with a greater overlap between these datasets. Following the instructions outlined on the scGPT GitHub, we obtained the complete pre-training cell corpus for scGPT from the CellXGene Census. As for scBERT, the pre-training dataset is derived from PanglaoDB and provided by the authors. The scBERT and scGPT datasets contain 1.4 million and 33 million cells, and 16,906 and 60,664 features respectively.

To carry out the contextual alignment experiment, we first ensure alignment between the paired datasets based on common genes. We normalize the fine-tuning dataset to a total read count of 10,000 over all genes and apply log1p-transformation. Additionally, we filter the data to include the same set of 2,061 highly variable genes that are used in the fine-tuning process (see Appendix A.2). Following these steps, we obtain two pre-training/fine-tuning common gene sets, 1,408 for scBERT + Norman and 2,044 for scGPT + Norman.

To quantify the alignment, we compare gene expression profiles between the fine-tuning and pre-training datasets by computing cosine similarity scores, which are advantageous due to their insensitivity to expression magnitude. This comparison generates a dense score matrix of dimensions $N_{\text{finetune}} \times N_{\text{pre-train}}$. For a subset of $N_{\text{pre-train}}$, used in at least one train-test split, an aggregate cross-split overlap is calculated to evaluate the impact of different pre-training/fine-tuning dataset configurations on model performance.

Initially, a matrix $S \in \mathbb{R}^{N_{\text{finetune}} \times N_{\text{pre-train}}}$ is constructed, where each element $s_{ij}$ represents the cosine similarity between the $i$-th cell in the fine-tuning dataset and the $j$-th cell in the pre-training dataset. From this, we derive a binary similarity matrix $B$ of the same dimensions with entries $b_{ij}$. The matrix is constructed as follows:

$$b_{ij} = \begin{cases} 1 & \text{if } s_{ij} \geq \mu + 2\sigma, \\ 0 & \text{otherwise,} \end{cases} \tag{H1}$$

where $\mu$ and $\sigma$ are the mean and standard deviation of the cosine similarities computed across 100,000 randomly sampled cell pairs. Based on this established threshold, $B$ represents whether each fine-tuning cell significantly overlaps with each pre-training cell.

To quantify the alignment for each fine-tuning cell, we aggregate over the pre-training dimension of matrix $B$ for each fine-tuning cell, resulting in a vector $\mathbf{f}$ where each component $f_i$ is given by:

$$f_i = \frac{1}{N_{\text{pre-train}}} \sum_{j=1}^{N_{\text{pre-train}}} B_{ij} \tag{H2}$$

Here, $f_i \in \mathbb{R}^{N_{\text{finetune}}}$ represents the fraction of the pre-training dataset that is similar to the $i$-th fine-tuning cell.

To conduct the sensitivity analysis, we define a threshold $\tau$, which represents the minimum fraction of the pre-training dataset that a fine-tuning cell must be similar to in order to be considered significantly aligned. $\tau$ is varied within the range of 0 to 0.1% of $N_{\text{pre-train}}$. For each value of $\tau$, we calculate the proportion of fine-tuning cells that meet or exceed this threshold, thus generating a series of values:

$$p(\tau) = \frac{1}{N_{\text{finetune}}} \sum_{i=1}^{N_{\text{finetune}}} \mathbf{1}_{\{f_i > \tau\}} \tag{H3}$$

where $\mathbf{1}$ is the indicator function that evaluates to 1 if the condition is true and 0 otherwise.

The sensitivity curve is then plotted as $p(\tau)$ versus $\tau$. The area under this curve reflects the overall cross-split overlap of the fine-tuning dataset relative to the pre-training dataset, as visualized in Figure H1.

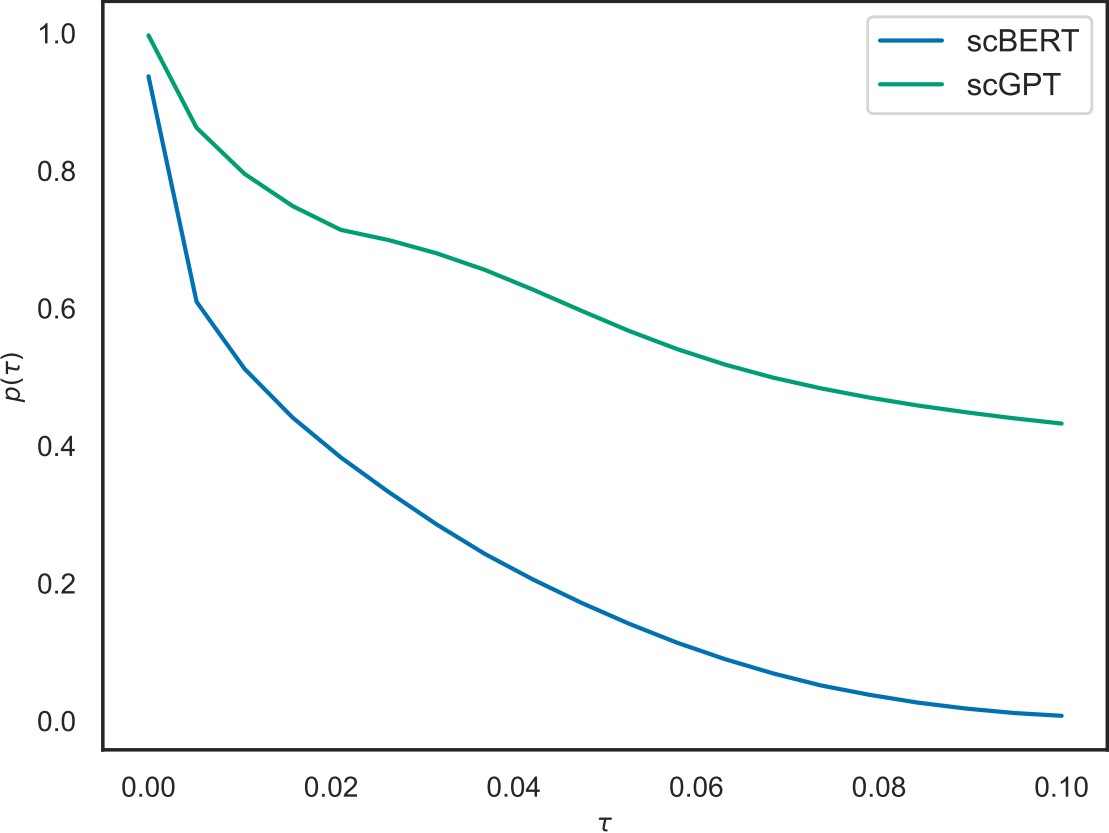

*Figure H1.* Plot of the probability that a cell from the pre-train dataset is similar to a cell from the fine-tune dataset as a function of $\tau$, the similarity threshold at which two cells are considered similar based on their cosine similarity.

# I. Supplementary Figures

## I.1. Perturbation Effect Prediction Evaluated across 2000 HVGs

### I.1.1. NORMAN SINGLE-GENE

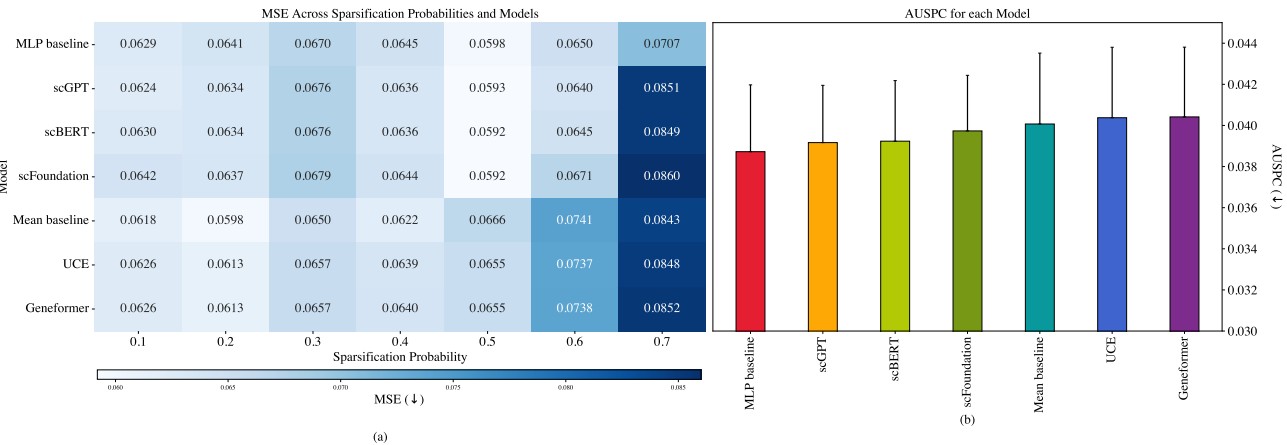

*Figure I1.* Predictions of single-gene perturbation effect for the Norman dataset evaluated across 2,000 highly variable genes for 8 train-test splits of increasing difficulty. (a) MSE for all prediction models. Experiments were carried out in triplicate for each model. The heatmap shows the mean MSE values (↓). (b) Average AUSPC (↓) across sparsification probabilities for each model with standard error bars.

### I.1.2. NORMAN DOUBLE-GENE

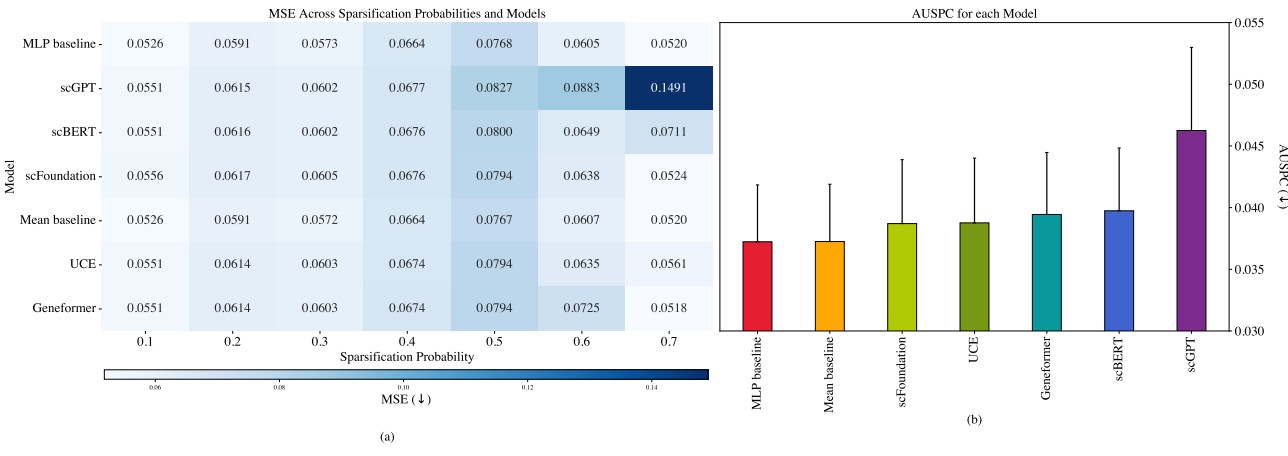

*Figure I2.* Predictions of double-gene perturbation effect for the Norman dataset evaluated across 2,000 highly variable genes for 8 train-test splits of increasing difficulty. (a) MSE for all prediction models. Experiments were carried out in triplicate for each model. The heatmap shows the mean MSE values (↓). (b) Average AUSPC (↓) across sparsification probabilities for each model with standard error bars.

## I.1.3. REPLOGLE K562

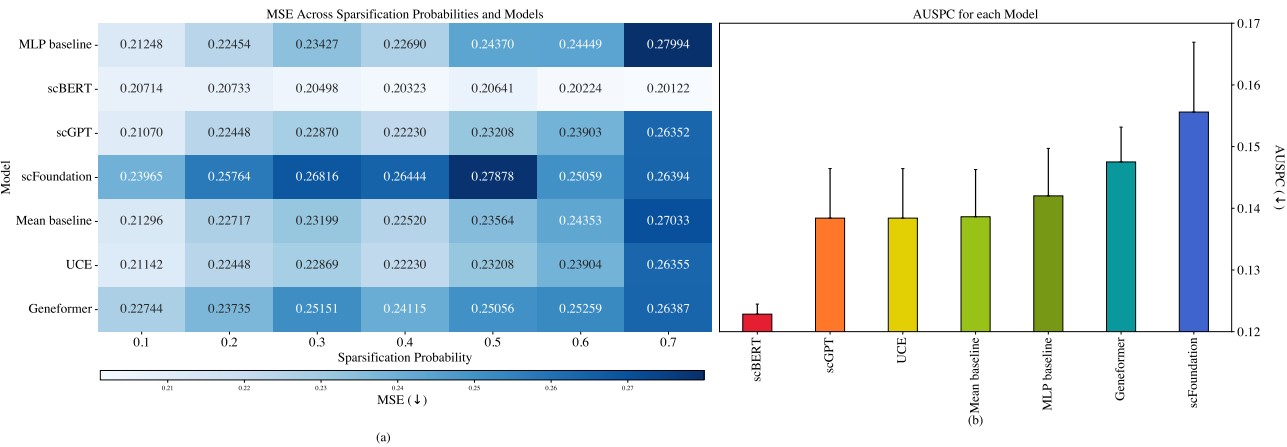

(a)

**Figure I3.** Predictions of single-gene perturbation effect for the Replogle K562 dataset evaluated across 2,000 highly variable genes for 8 train-test splits of increasing difficulty. (a) MSE for all prediction models. Experiments were carried out in triplicate for each model. The heatmap shows the mean MSE values (↓). (b) Average AUSPC (↓) across sparsification probabilities for each model with standard error bars.

## I.1.4. REPLOGLE RPE1

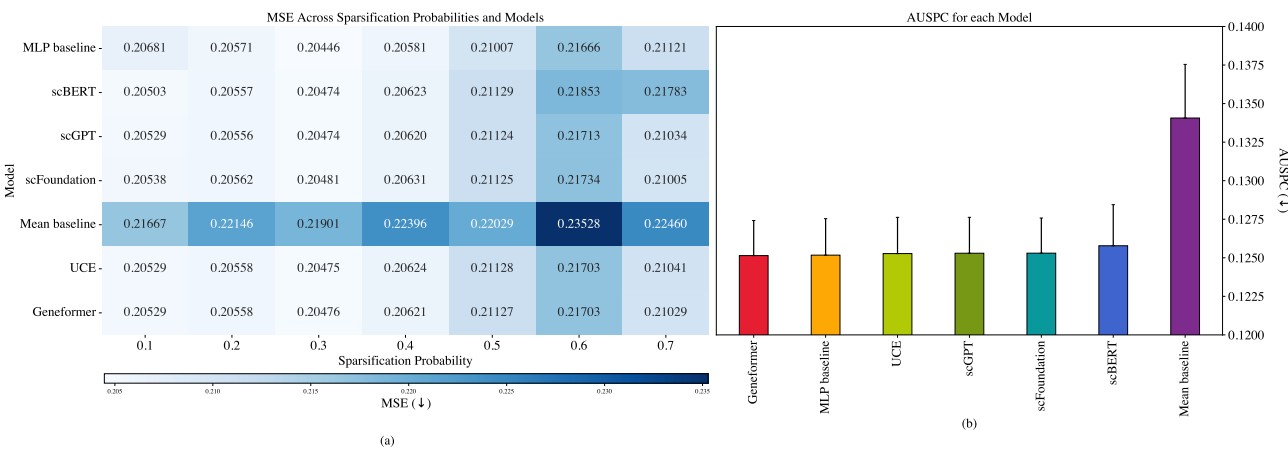

(a)

**Figure I4.** Predictions of single-gene perturbation effect for the Replogle RPE1 dataset evaluated across 2,000 highly variable genes for 8 train-test splits of increasing difficulty. (a) MSE for all prediction models. Experiments were carried out in triplicate for each model. The heatmap shows the mean MSE values (↓). (b) Average AUSPC (↓) across sparsification probabilities for each model with standard error bars

## I.2. SPECTRA Performance Curves

### I.2.1. NORMAN SINGLE-GENE

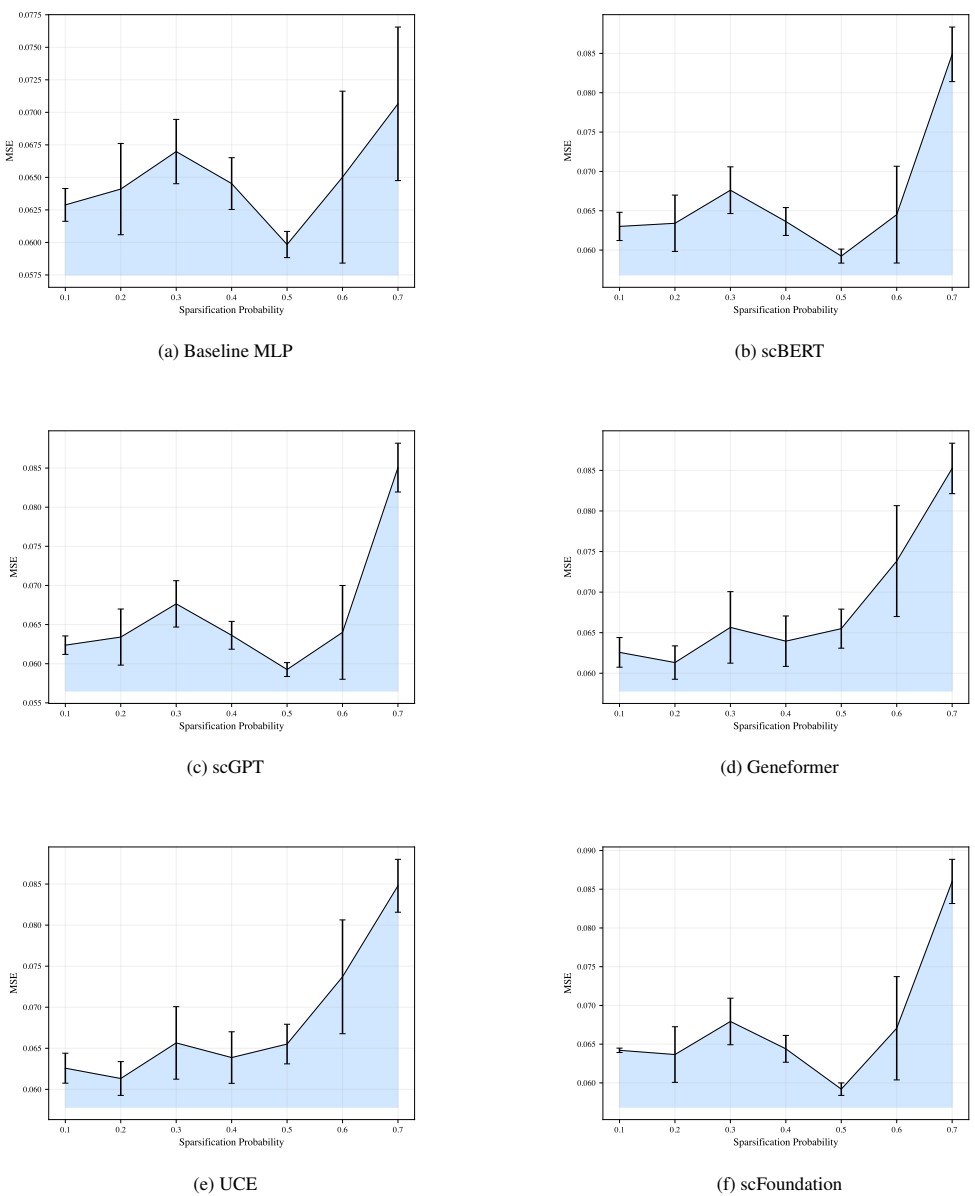

(a) Baseline MLP

(b) scBERT

(c) scGPT

(d) Geneformer

(e) UCE

(f) scFoundation

*Figure I5.* MSE as a function of the sparsification probability for the different models, tested on Norman single-gene effect perturbation prediction. These functions are used to calculate to calculate the AUSPC, which is here shaded in blue.

### I.2.2. NORMAN DOUBLE-GENE

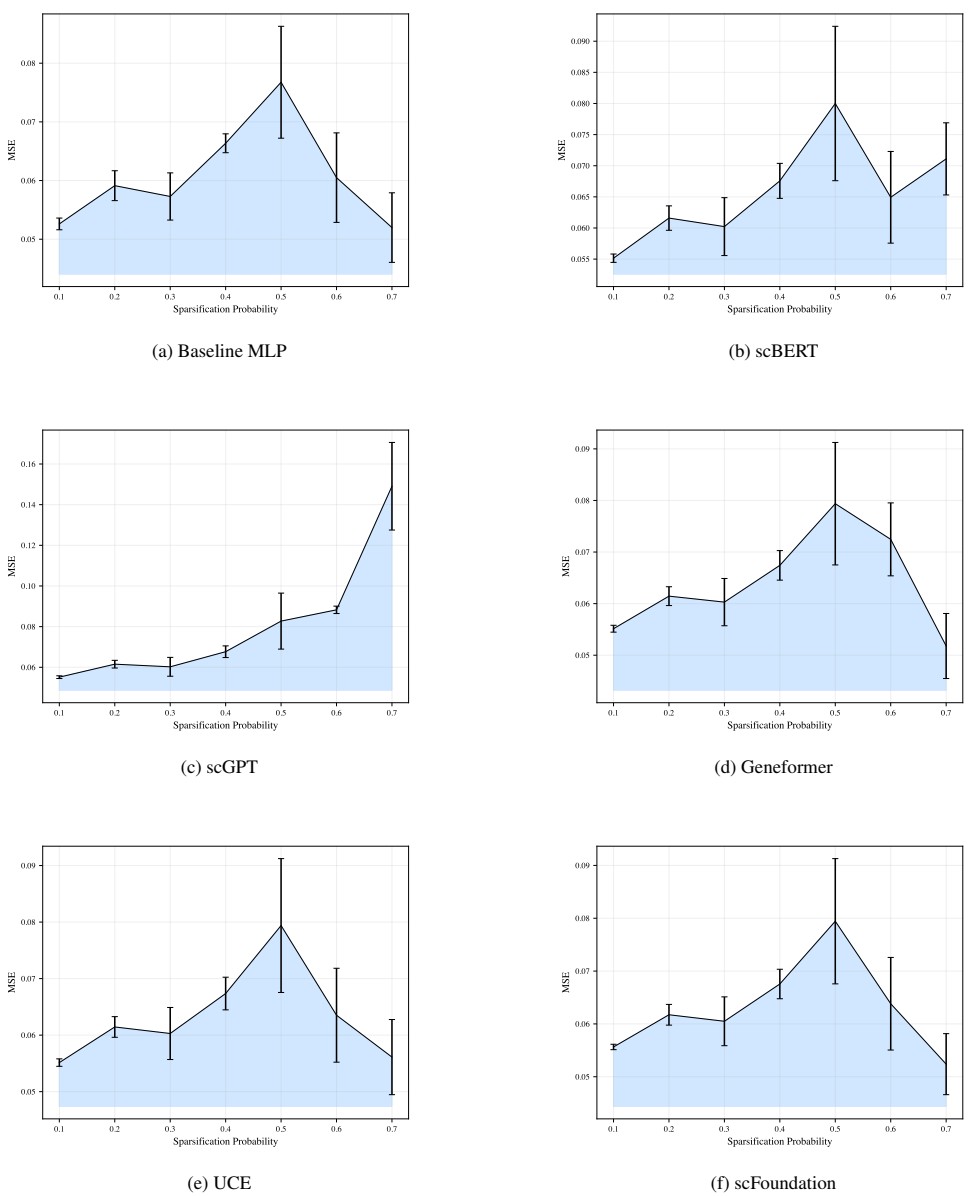

Figure I6. MSE as a function of the sparsification probability for the different models, tested on Norman double-gene perturbation effect prediction. These functions are used to calculate to calculate the AUSPC, which is here shaded in blue.

## I.2.3. REPLOGLE K562

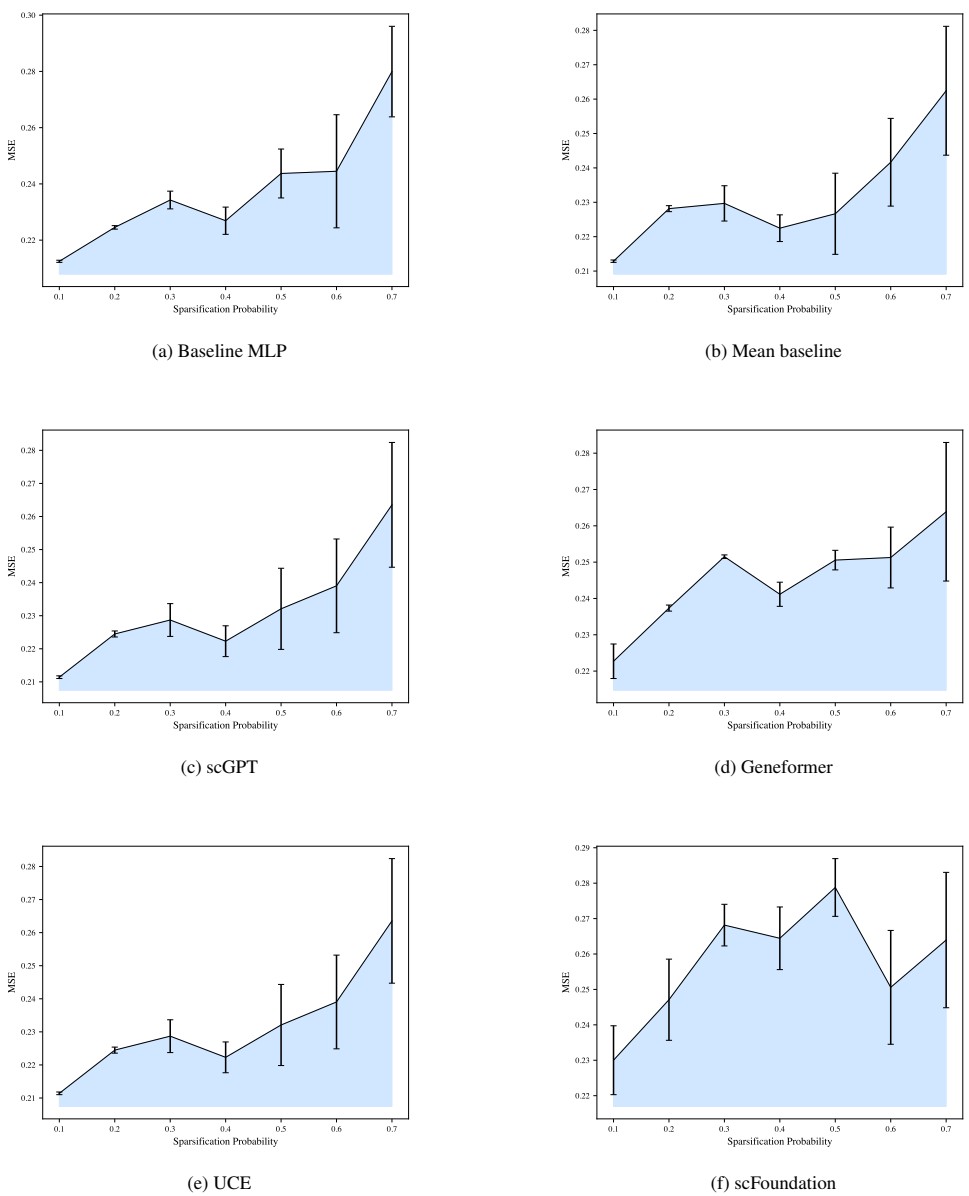

Figure I7. MSE as a function of the sparsification probability for the different models, tested on the Replogle K562 dataset. These functions are used to calculate to calculate the AUSPC, which is here shaded in blue.

## I.2.4. REPLOGLE RPE1

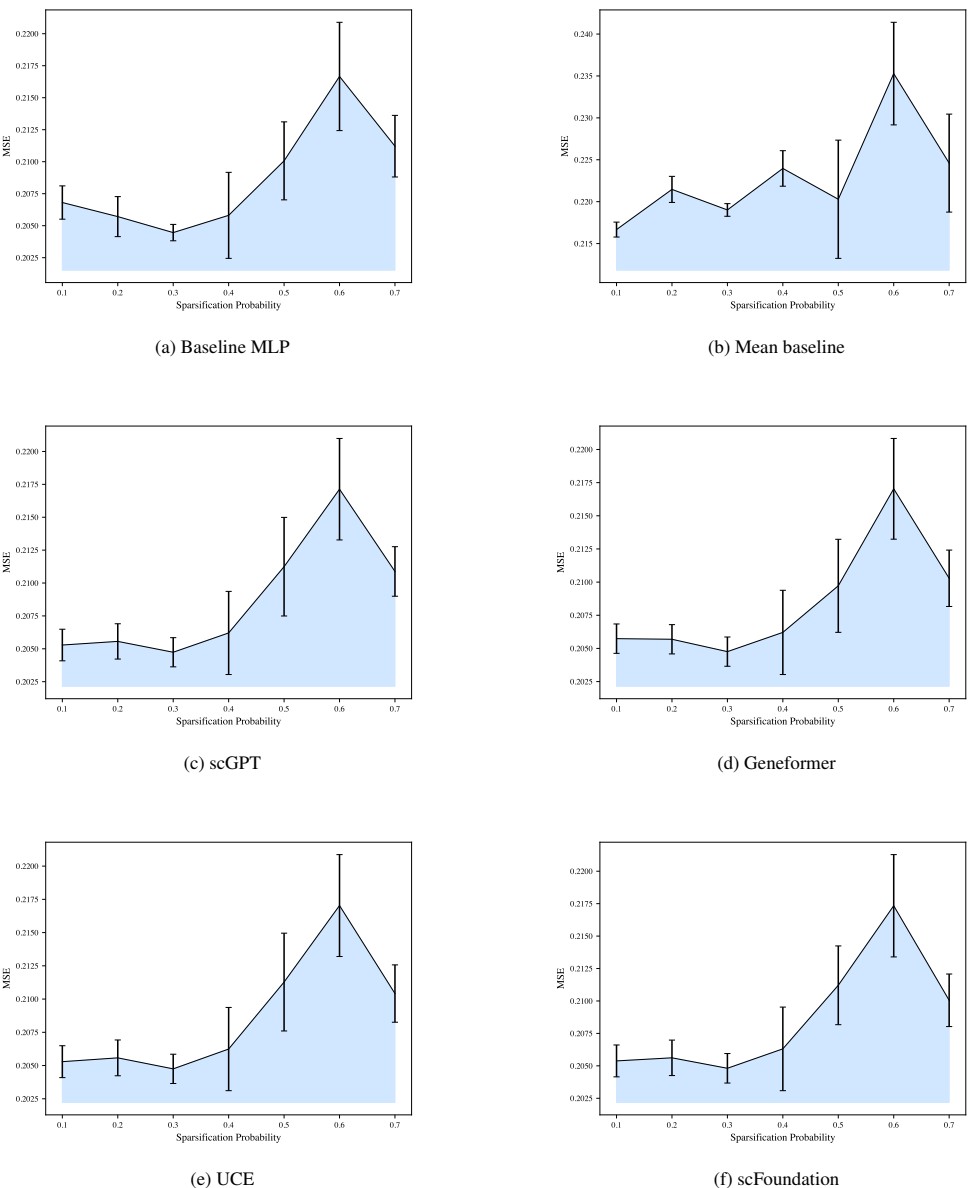

*Figure I8.* MSE as a function of the sparsification probability for the different models, tested on the Replogle RPE1 dataset. These functions are used to calculate to calculate the AUSPC, which is here shaded in blue.

## I.3. MSE for all Models Compared to Mean Baseline across 2000 HVGs

### I.3.1. NORMAN SINGLE-GENE

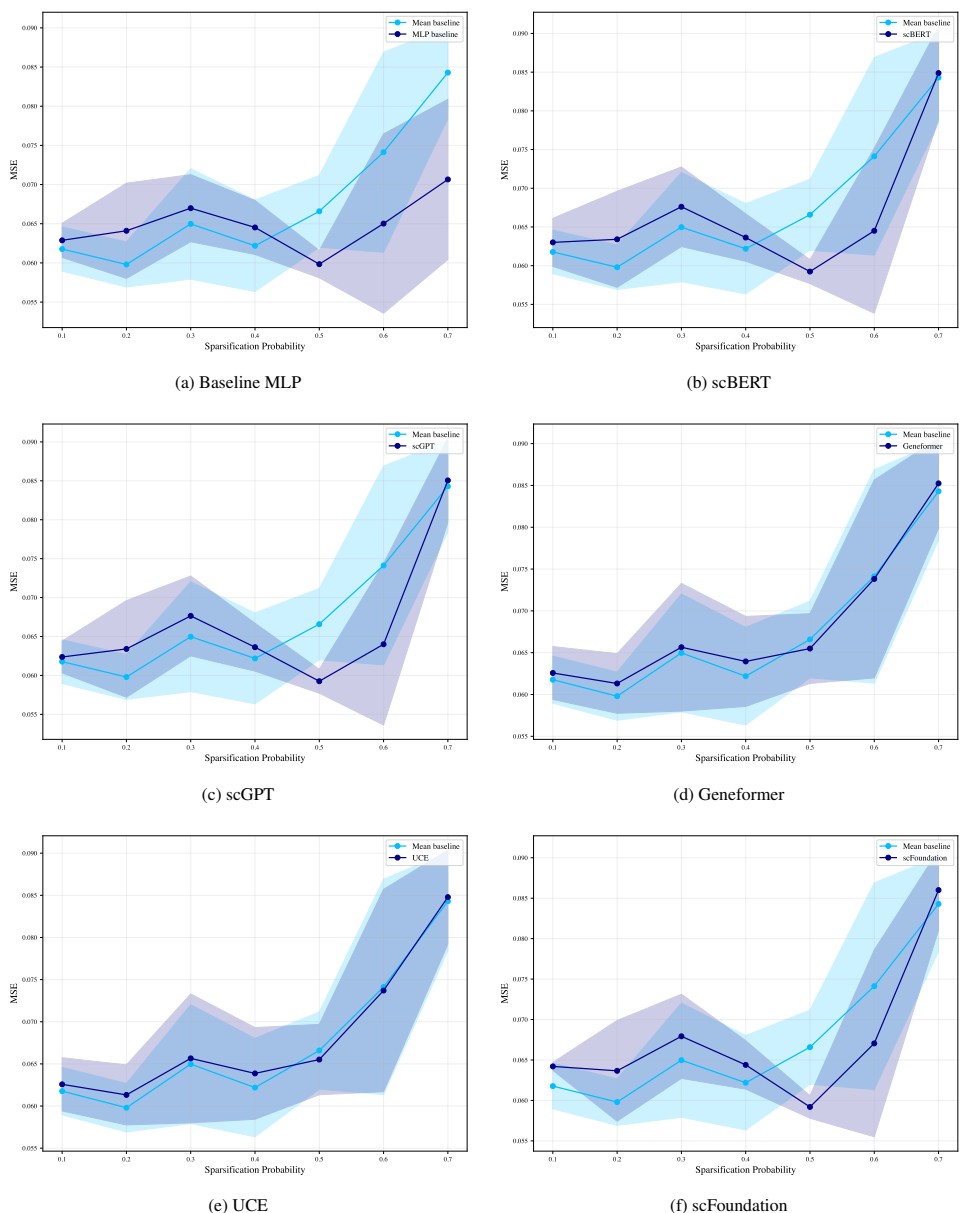

(a) Baseline MLP

(b) scBERT

(c) scGPT

(d) Geneformer

(e) UCE

(f) scFoundation

*Figure I9.* MSE as a function of the sparsification probability for the different models. This is a depiction of the curves that are used to calculate the $\Delta$AUSPC.

I.3.2. NORMAN DOUBLE-GENE

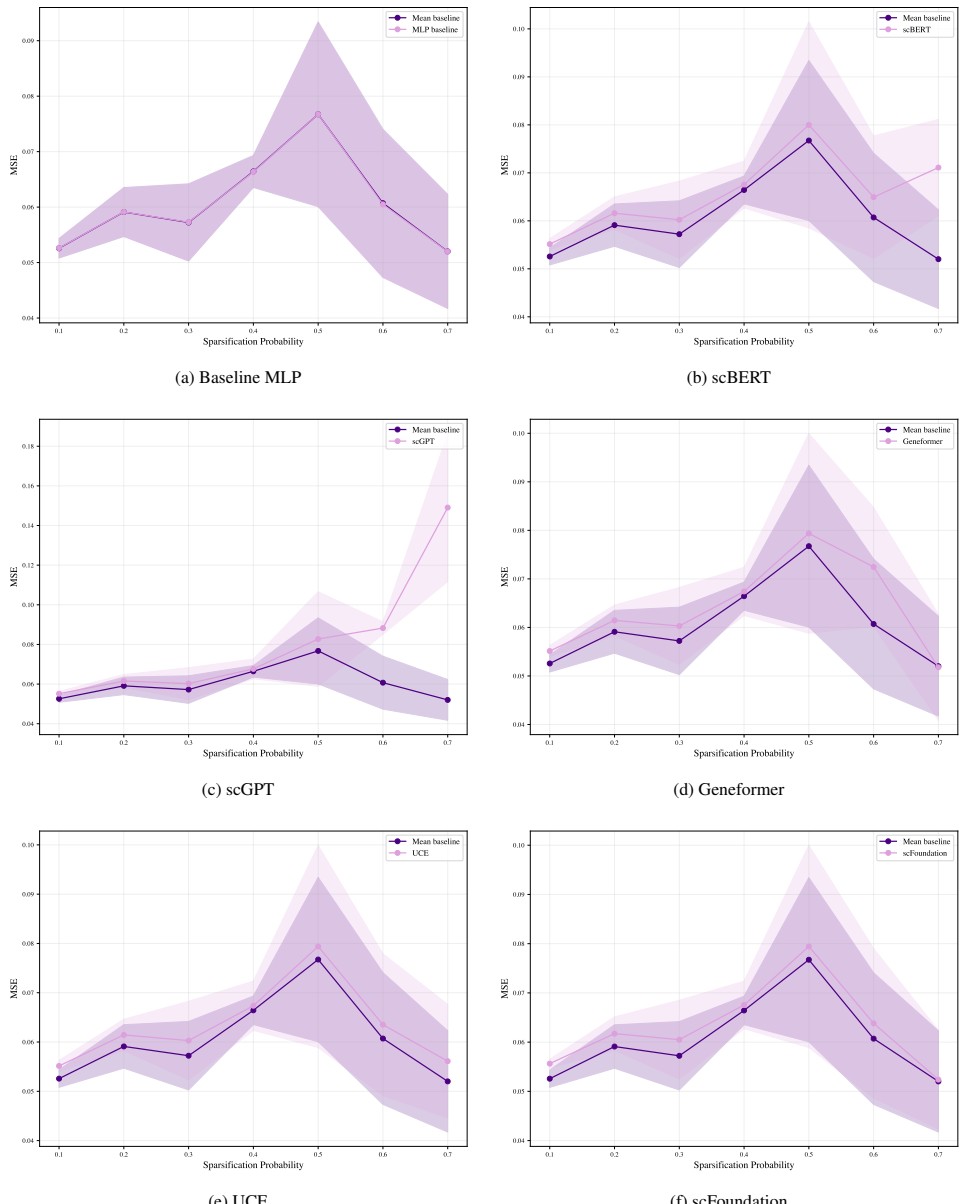

(a) Baseline MLP

(b) scBERT

(c) scGPT

(d) Geneformer

(e) UCE

(f) scFoundation

*Figure I10.* MSE as a function of the sparsification probability for the different models. This is a depiction of the curves that are used to calculate the ΔAUSPC.

## I.3.3. REPLOGLE K562

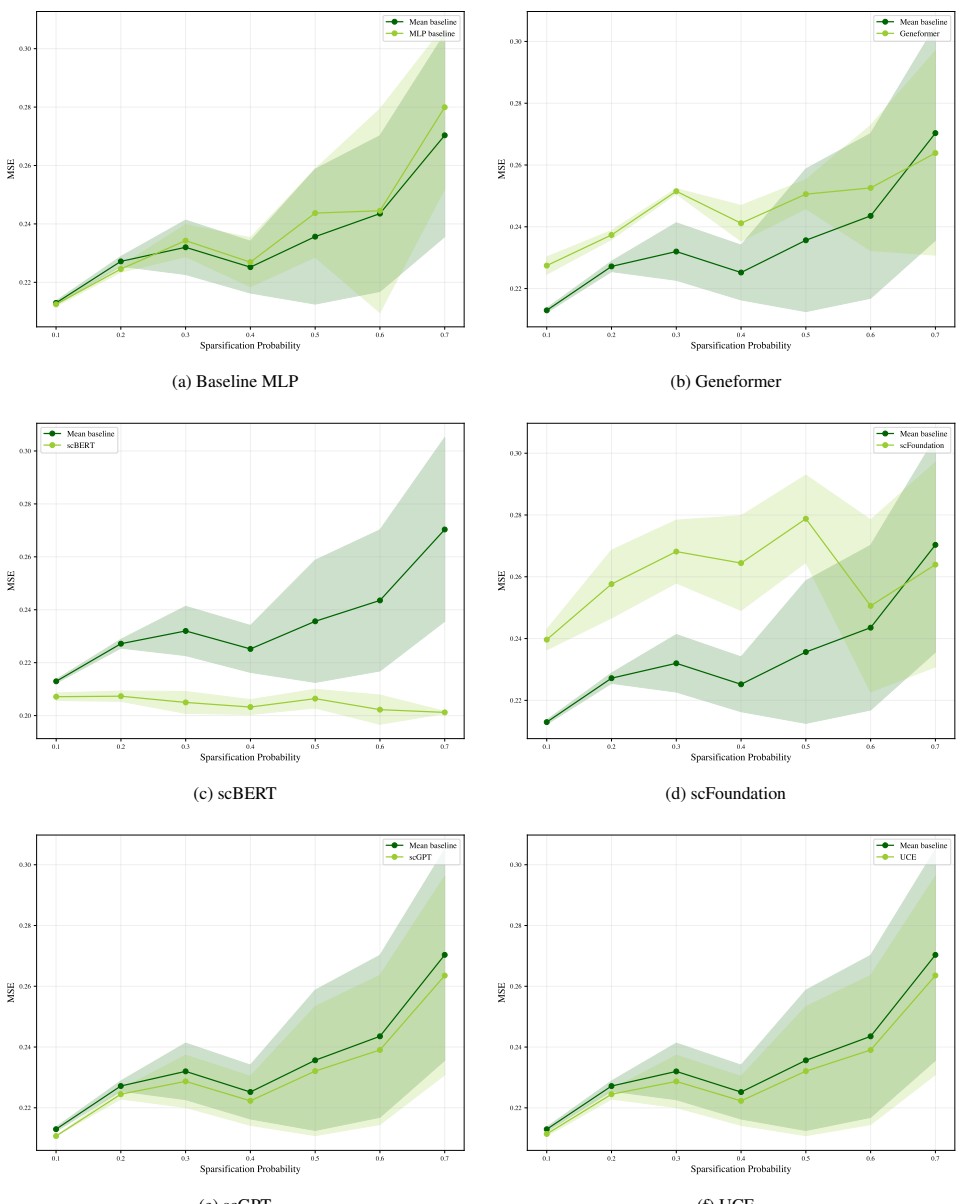

*Figure I11.* MSE as a function of the sparsification probability for the different models. This is a depiction of the curves that are used to calculate the $\Delta$AUSPC.

## I.3.4. REPLOGLE RPE1

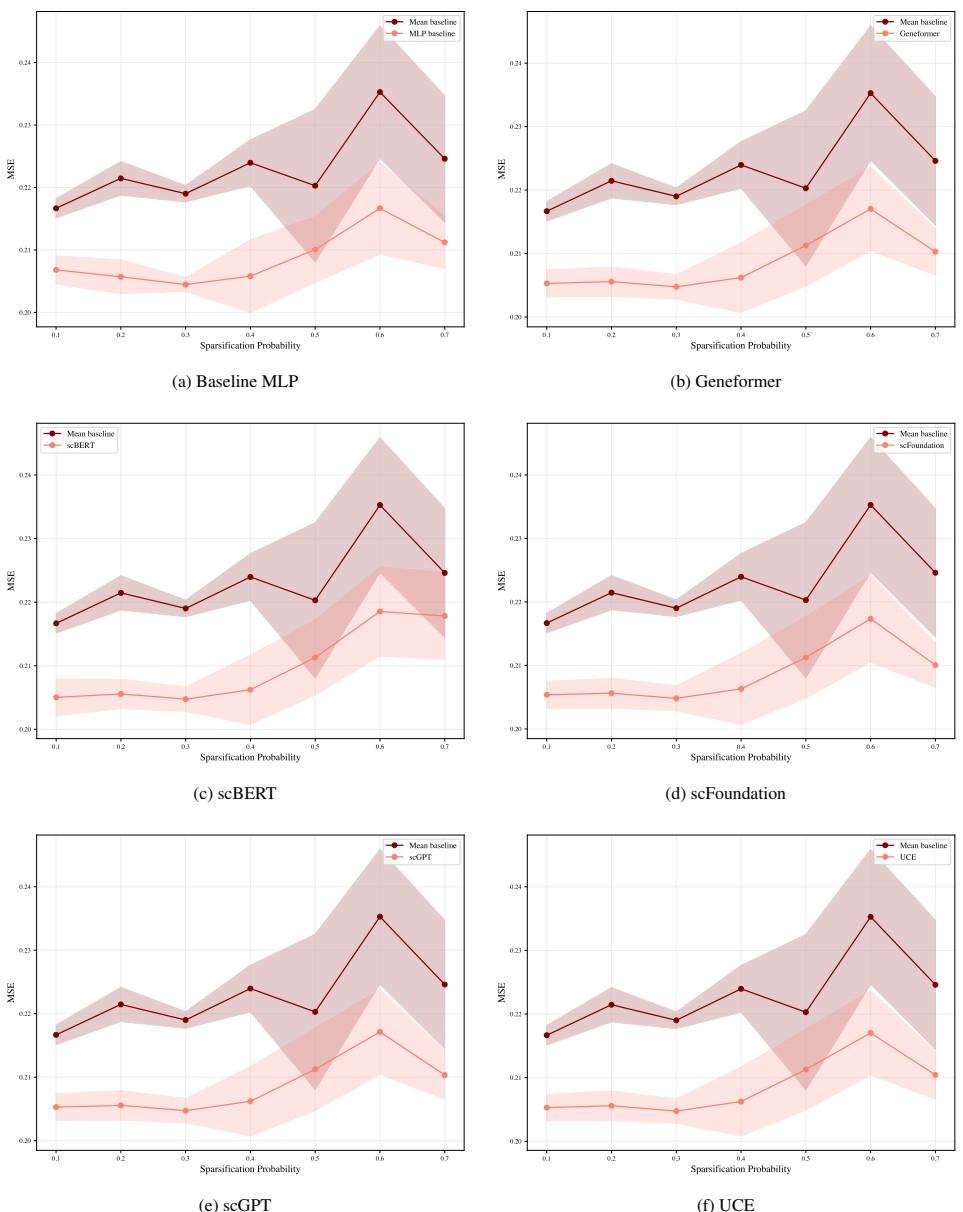

*Figure I12.* MSE as a function of the sparsification probability for the different models. This is a depiction of the curves that are used to calculate the ∆AUSPC.

## I.4. Comparison between Mean Baseline Performance on Predicting Perturbation Effect on the top 2000 HVGs and the top 20 DEGs

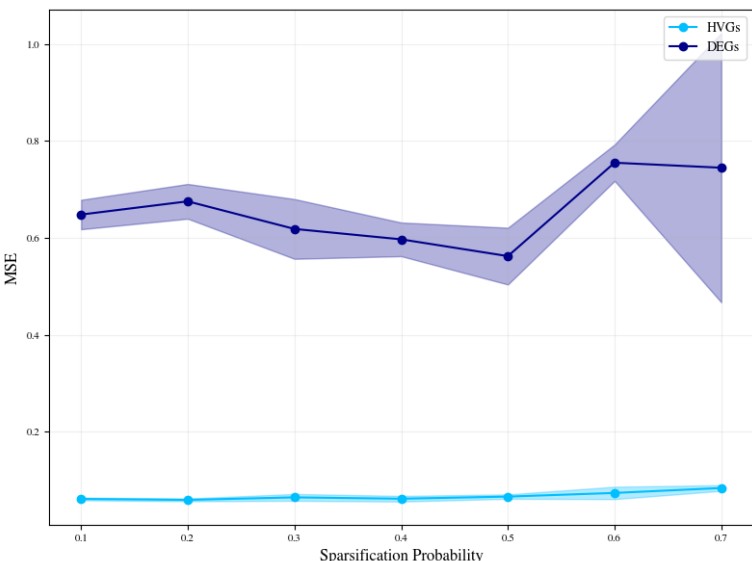

*Figure I13.* Comparison of the performance of the mean baseline on the highly variable genes (HVGs) vs. differentially expressed genes (DEGs) task across different sparsification probability train-test splits for Norman single-gene perturbation effect prediction.

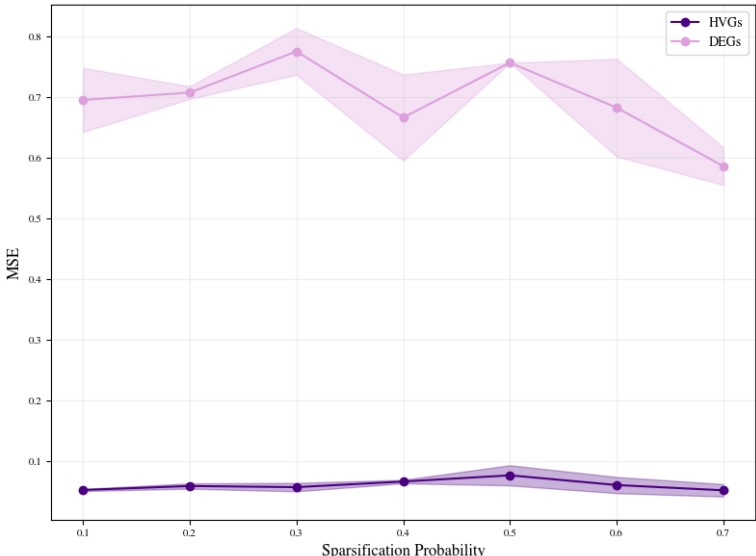

*Figure I14.* Comparison of the performance of the mean baseline on the highly variable genes (HVGs) vs. differentially expressed genes (DEGs) task across different sparsification probability train-test splits for Norman double-gene perturbation effect prediction.

## I.5. Perturbation Effect Prediction Results across the top 20 DEGs

### I.5.1. NORMAN SINGLE-GENE

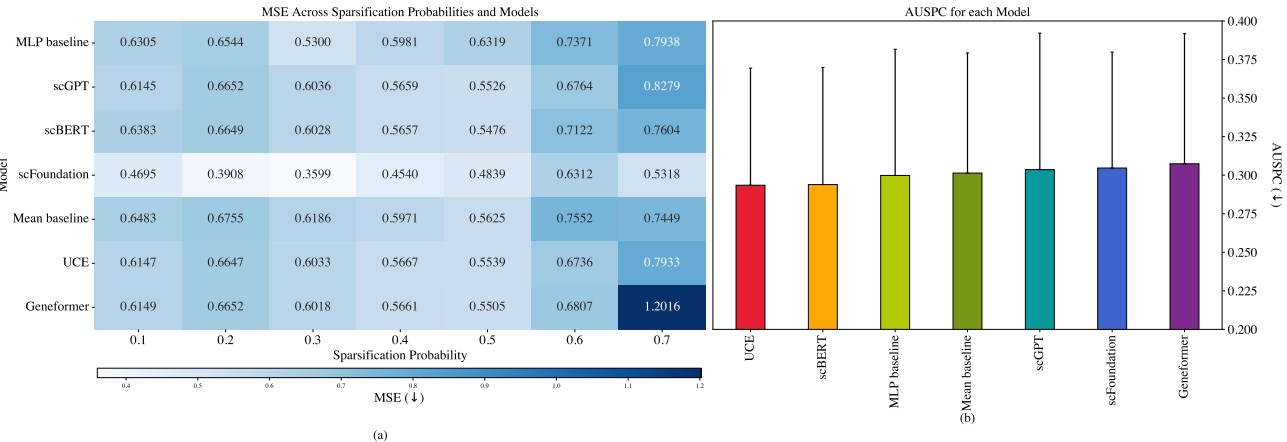

(a)

*Figure I15.* Predictions of single-gene perturbation effect for the Norman dataset evaluated across the top 20 differentially expressed genes for 8 train-test splits of increasing difficulty. (a) MSE for all prediction models. Experiments were carried out in triplicate for each model. The heatmap shows the mean MSE values (↓). (b) Average AUSPC (↓) across sparsification probabilities for each model with standard error bars.

### I.5.2. NORMAN DOUBLE-GENE

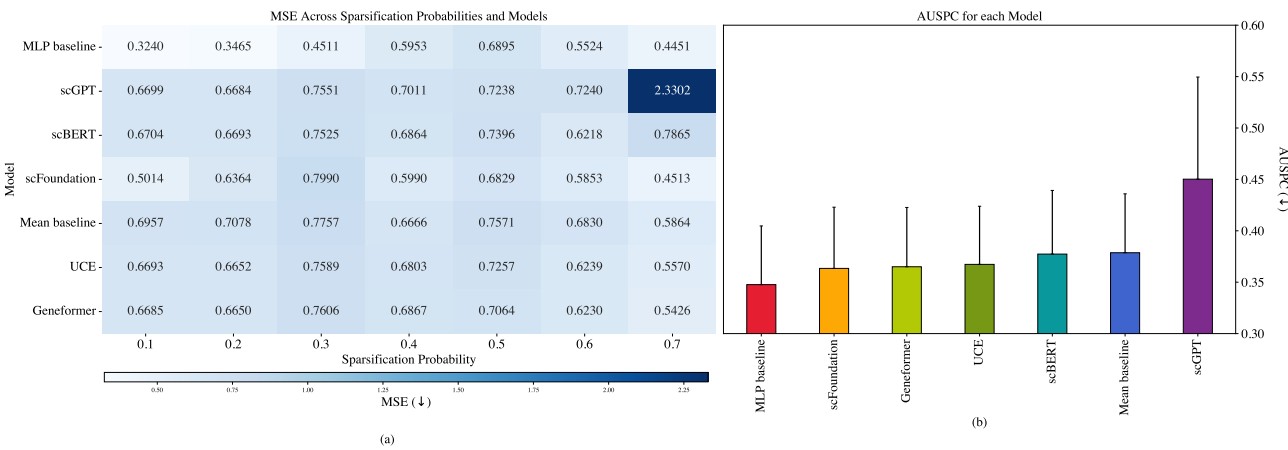

(a)

*Figure I16.* Predictions of double-gene perturbation effect for the Norman dataset evaluated across the top 20 differentially expressed genes for 8 train-test splits of increasing difficulty. (a) MSE for all prediction models. Experiments were carried out in triplicate for each model. The heatmap shows the mean MSE values (↓). (b) Average AUSPC (↓) across sparsification probabilities for each model with standard error bars.

## I.6. MSE for all Models compared to Mean Baseline across 20 DEGs

### I.6.1. NORMAN SINGLE-GENE

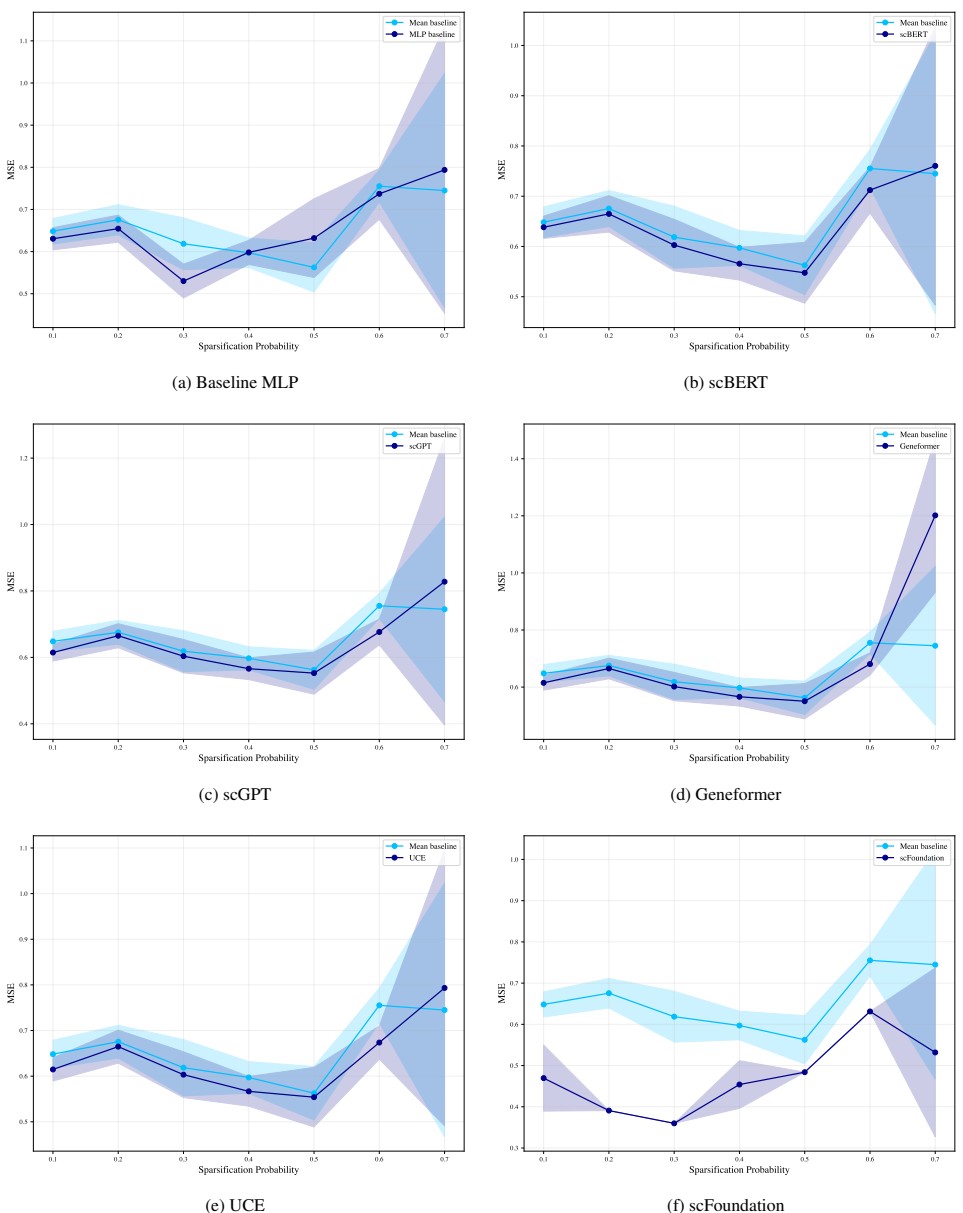

(a) Baseline MLP

(b) scBERT

(c) scGPT

(d) Geneformer

(e) UCE

(f) scFoundation

*Figure I17.* MSE as a function of the sparsification probability for the different models evaluated across the top 20 differentially expressed genes. This is a depiction of the curves that are used to calculate the $\Delta$AUSPC.

## I.6.2. NORMAN DOUBLE-GENE

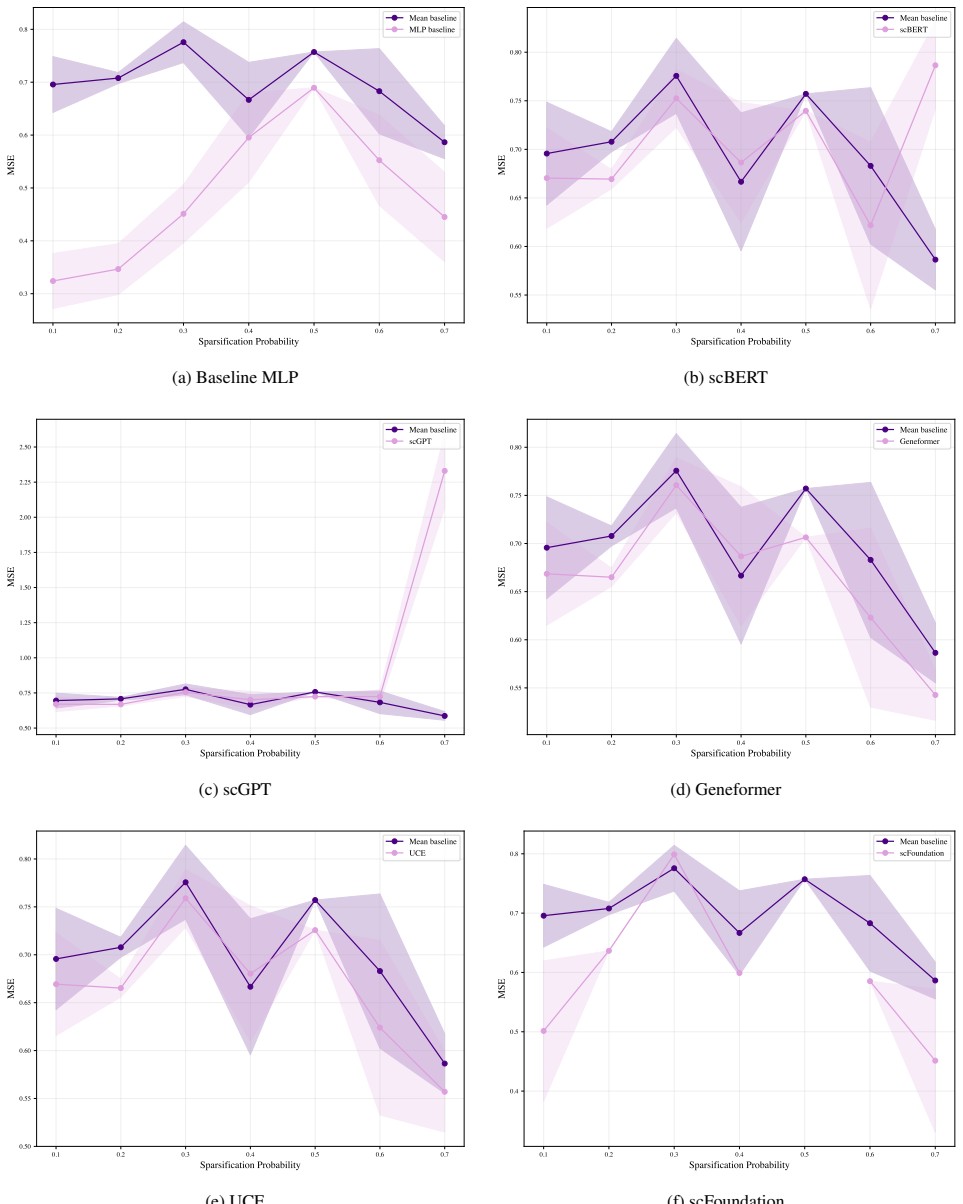

*Figure I18.* MSE as a function of the sparsification probability for the different models evaluated across the top 20 differentially expressed genes. This is a depiction of the curves that are used to calculate the ΔAUSPC.

## I.7. Mean Post-Perturbation Expression Profiles for *IKZF3* and *CEBPA*

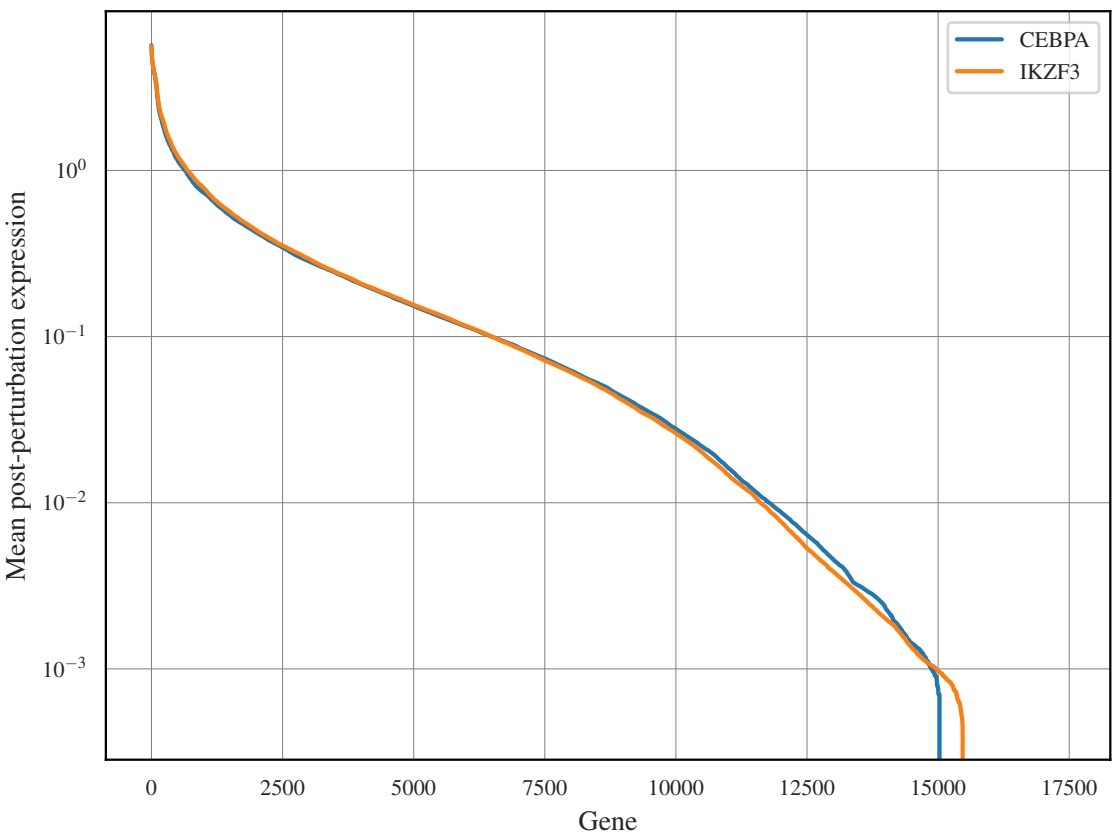

*Figure I19.* Post-perturbation mean expression profiles for *IKZF3* and *CEBPA*. The y-axis has been log-transformed for visual clarity.

