# OpenReview forum: "PertEval-scFM: Benchmarking Single-Cell Foundation Models for Perturbation Effect Prediction"
_ICML.cc/2025/Conference — ICML 2025 poster_

### Official Review · Reviewer_KAX4 · 2025-03-03

**Overall Recommendation:** 3

**Summary:**

The paper introduces **PertEval-scFM**, a standardized framework to evaluate single-cell foundation models (scFMs) for predicting perturbation effects.
Key contributions include:
1. **Framework**: A modular toolkit for zero-shot evaluation of scFM embeddings.
2. **Metrics**: Introduces **AUSPC**, **E-distance**, and **contextual alignment** to assess model robustness and generalization under distribution shifts.
3. **Findings**: scFM embeddings do not consistently outperform baseline models, especially under distribution shifts. **GEARS** performs best, highlighting the need for task-specific architectures.

PertEval-scFM provides a comprehensive benchmark for scFMs, emphasizing challenges in perturbation effect prediction and guiding future research.

**Claims And Evidence:**

Yes.

**Essential References Not Discussed:**

No.

**Experimental Designs Or Analyses:**

Yes. Although GEARS achieves the best performance, it requires constructing a cell adjacency matrix based on cell similarity first. This results in GEARS still being aware of the relationships between cells even after SPECTRA divides the training and test sets. Since the paper does not provide detailed information on the implementation of GEARS, I hope this can be clarified further.

**Methods And Evaluation Criteria:**

Yes.

**Other Comments Or Suggestions:**

No.

**Other Strengths And Weaknesses:**

Weaknesses:
1.Figure 2: Including GEARS results would provide a more complete comparison of model performance
2.Figure 3b: The proportion of high E-distance samples under high probability conditions is relatively low. Further analysis of outliers could reveal underlying patterns.
3.Figure 4: The dense data points make it difficult to draw clear conclusions.

**Questions For Authors:**

No.

**Relation To Broader Scientific Literature:**

1. Fair Comparison in Single-Cell Foundation Models
While Geneformer and scGPT have demonstrated the promising potential of foundation models in single-cell biology, this study systematically evaluates their true practical utility under zero-shot settings with distribution shifts.

2. Perturbation Effect Prediction


3. Evaluation Frameworks for Biological Models
Comparison to scEval (Wang et al., 2024) evaluated 8 general tasks (e.g., clustering, batch correction), PertEval-scFM introduced perturbation-specific innovations.

**Theoretical Claims:**

Yes. This is primarily a benchmarking experiment, with limited theoretical proof.

---

> ### Author Rebuttal · Authors · 2025-03-31
>
> We thank the reviewer for their detailed and constructive feedback, as well as for recognizing the strengths of our work, including the comprehensiveness of our benchmark and its potential to guide future research. Below, we respond to the specific concerns raised.
>
> **GEARS**
>
> We agree that including GEARS in Fig. 2 provides a more complete picture of model performance and have updated the figure accordingly (see [link](https://drive.google.com/file/d/1dJBOtZ2ytpMee1rK5qHWVIqJ_NVx1rnZ/view?usp=sharing)).
>
> GEARS uses a **gene coexpression** graph and a **gene ontology** graph to model gene relationships and perturbations, respectively, thus injecting biological priors into the model architecture. On the other hand, the SPECTRA graph is built on **cell-to-cell** similarity, which enables us to measure model robustness to distribution shift by simulating increasingly different train-test sets. GEARS is therefore not aware of the relationship between cells, but aware of the relationship between genes, which therefore doesn’t interfere with the way we assess robustness. We have clarified this in the manuscript to avoid any ambiguity.
>
> Regarding the implementation, GEARS was faithfully reproduced following its original design \[1\]. Briefly, it considers a perturbation dataset of $N$ cells, where each cell is described by a gene expression vector $g \\in \\mathbb{R}^K$ and an associated perturbation set $P \= (P\_1, \\ldots, P\_M)$. The model learns a function $f$ that maps a novel perturbation set to its post-perturbation gene expression profile. Specifically, GEARS:
>
> * Uses a GNN encoder $f\_{\\text{pert}}: \\mathcal{Z} \\rightarrow \\mathbb{R}^d$ to embed perturbations,
>
> * Uses a second GNN encoder $f\_{\\text{gene}}: \\mathcal{Z} \\rightarrow \\mathbb{R}^d$ to embed genes,
>
> * Combines these embeddings via a compositional module,
>
> * Applies a cross-gene decoder $f\_{\\text{dec}}: {\\mathbb{R}^d}^K \\rightarrow \\mathbb{R}^K$ to predict the post-perturbation gene expression vector.
>
> Training is conducted end-to-end using the autofocus direction-aware loss, with default hyperparameters from the GEARS paper (e.g., hidden size 64, one GNN layer for both GO and co-expression graphs). The only modification we made was to the GEARS dataloader: we adapted the \`prepare\_splits\` function to use SPECTRA-defined training and test splits via a custom mapping (in \`set2conditions\`).
>
> **Figures 3b and 4**
>
> In Fig. 4 we further analyse the outliers seen in Fig. 3b. There, we explore the distribution of the perturbation effect on the top 20 DEGs (dashed line), and how such perturbation effect is predicted by the models (data points). Dense clustering near the dashed line (e.g., Fig. 4a) indicates high agreement with ground truth, while more dispersed predictions (e.g., Fig. 4c) suggest low reliability. This analysis suggests that the magnitude of the perturbation effect is not the only factor affecting performance, but that the distribution of such effect also matters. Perturbations with lower overall effect, but with an atypical distribution will also challenge the models. These plots emphasize how the structure of the ground truth perturbation distribution affects model accuracy, a point we highlight in the revised figure caption. We have revised the Figure to improve clarity, using smaller, non-overlapping data points to better distinguish their values. The revised figure is available at this [link](https://drive.google.com/file/d/1VymFV4tVhO7Xvp75AuSJZToWEtLC4zSu/view?usp=drive_link).
>
> We hope these clarifications and updates address the reviewer’s concerns. We thank the reviewer again for their thoughtful comments, which helped us strengthen the clarity and completeness of our work.
>
> \[1\] [https://doi.org/10.1038/s41587-023-01905-6](https://doi.org/10.1038/s41587-023-01905-6)

---

### Official Review · Reviewer_pVrS · 2025-03-09

**Overall Recommendation:** 1

**Summary:**

This paper introduces PertEval-SCFM a benchmark for zero-shot single-cell foundational model embeddings to capture transcriptional perturbation effects. It is claimed that scum embeddings do not provide consistent improvements over baseline models.

**Claims And Evidence:**

Yes

**Essential References Not Discussed:**

None to my knowledge.

**Experimental Designs Or Analyses:**

Yes.

to construct embeddings for perturbed cells, expression counts of perturbed genes are set to zero in cells exposed to that perturbation "effectively simulating the perturbation in silico"

While I agree that this is a valid design, it also has some limitations. Namely there is no way for the model to distinguish between an intervention and a zero gene expression observation. From the causality literature, this is theoretically an issue. It seems somewhat difficult to claim that these models don't capture perturbation effects if this is the only type of embedding extracted.

However, the authors acknowledge this problem in the limitations.

One other concern is that the Norman dataset uses CrisperA (i.e. activation) so this zeroing of gene expression seems even more wrong in this context. Have the authors tried an activation type intervention?

**Methods And Evaluation Criteria:**

Yes

**Other Comments Or Suggestions:**

No

**Other Strengths And Weaknesses:**

Strengths:
* Originality: I have not seen a paper that benchmarks the zero-shot effectiveness of scFMs on perturbations. This is a timely work.
* Significance: While the conclusions of this work are not overly surprising, they set a useful benchmark and set of metrics for future work.
* Clarity: I found the writing clear given the complexity and domain knowledge of this particular benchmark.

Weaknesses:
* This work only explores two datasets, both of which have their own issues. It's not clear if these findings are generalizable across datasets.
* Fairly standard preprocessing is used. It's unclear if this is the same preprocessing used to train various scFMs thus it is unclear how fair this comparison is in this context to me.

**Questions For Authors:**

None

**Relation To Broader Scientific Literature:**

There are scFM models that claim to model perturbations zero-shot. This work disputes those claims. This is a useful contribution.

**Theoretical Claims:**

N/A

---

> ### Author Rebuttal · Authors · 2025-03-31
>
> We thank the reviewer for their thoughtful and positive evaluation of our work. We are glad that the reviewer found our contribution timely and the benchmark, metrics, and manuscript clear and useful for the field. Below, we address the specific concerns raised.
>
> **Causal intervention**
>
> We acknowledge that setting gene expression to zero may prevent the model from distinguishing between biological zeros and knockout interventions. However, there will be differences in coexpression accompanying a natural zero, which will not occur for induced zeros. We acknowledge this limitation in our paper and agree that future foundation models would benefit from incorporating representations that explicitly encode the nature of the intervention. A potential avenue of exploration would be to include perturbation tokens during pretraining and not only during fine-tuning.
>
> **Perturbation representation**
>
> While simulating perturbations via gene knockout may seem counterintuitive for CRISPRa datasets like Norman, we chose this strategy to address the challenges inherent in standardizing perturbation presentation across diverse scFMs. Different scFMs represent perturbations in different ways \- for instance, scGPT encodes gene expression as numerical values, whereas Geneformer uses rank-order representations.
>
> Therefore, upregulation is difficult to simulate consistently: parametric models like scGPT require arbitrary magnitude choices, with no biologically grounded way to determine appropriate magnitude, while rank-based models like Geneformer lack a principled way to re-rank genes. On the other hand, knockout offers a model-agnostic, unbiased representation that avoids model-specific inconsistencies, enabling fairer comparisons across scFMs.
>
> Nonetheless, we performed an experiment using an alternative perturbation representation, which simulates gain-of-function by doubling the perturbed gene’s expression in the Norman control expression data. We used scBERT to generate embeddings for the *in silico* upregulated input representation (scBERT+). We observed minimal performance differences (see Table 1 and the Figure in this [link](https://drive.google.com/file/d/1jju-RECJcVANDfUj9s5-oobKPFSNQ6_9/view?usp=drive_link)) compared to the knockout representation (scBERT-), supporting our approach. The difficulty of simulating realistic perturbations reflects a broader challenge in scFM methodology and while a full exploration of perturbation strategies is beyond the scope of this work, we hope our findings motivate further research in this area.
>
> Table 1: MSE ± standard deviation for Norman single gene embeddings generated with scBERT using an in silico knockout (scBERT \-) vs. an upregulation strategy (scBERT+)
>
> |  | 0.1 | 0.2 | 0.3 | 0.4 | 0.5 | 0.6 | 0.7 |
> | :---- | :---- | :---- | :---- | :---- | :---- | :---- | :---- |
> | scBERT - | 0.0630 ± 0.0031 | 0.0634 ± 0.0062 | 0.0676 ± 0.0051 | 0.0636 ± 0.0031 | 0.0592 ± 0.0015 | 0.0645 ± 0.0107 | 0.0849 ± 0.0060 |
> | scBERT+ | 0.0640 ± 0.0038 | 0.0620 ± 0.0038 | 0.0658 ± 0.0077 | 0.0610 ± 0.0009 | 0.0659 ± 0.0046 | 0.0744 ± 0.0122 | 0.0853 ± 0.0064 |
>
> **Dataset and pre-processing choices**
>
> We agree that dataset diversity is critical for generalizability. We selected Norman and Replogle as they represent two of the most widely used, high-quality single-cell perturbation datasets currently available. Moreover, both Norman and Replogle contain two separate datasets each, so we are effectively exploring four datasets. Importantly, these all allow us to benchmark under controlled conditions and introduce distribution shifts via SPECTRA. Our platform is designed to be extensible, and we are actively exploring incorporation of additional datasets, including large-scale resources such as the recently published Tahoe-100M \[1\].
>
> Regarding preprocessing, we applied a consistent pipeline that follows standard best practices in single-cell analysis across all models, including highly variable gene (HVG) selection, transcript count normalization, and log transformation. These steps are aligned with those used in the original scFM papers, differing only in hyperparameter value choices. We ensure internal consistency in preprocessing across all model evaluations to enable fair comparisons. While we acknowledge that preprocessing choices can affect results, this is a well-recognized, broader limitation of single-cell data analysis, which we explicitly discuss in our manuscript.
>
> We hope our responses clarify the rationale behind our design choices and reinforce the value of our benchmark as a foundation for future work. We believe that making the limitations explicit and quantifiable is a critical step for advancing the development of biologically meaningful scFMs.
>
> \[1\] [https://www.biorxiv.org/content/10.1101/2025.02.20.639398v1](https://www.biorxiv.org/content/10.1101/2025.02.20.639398v1)

---

> > ### Comment · Reviewer_pVrS · 2025-04-03
> >
> > ### Finetuning
> >
> > Several other reviewers mentioned the lack of fine-tuning of the single cell foundation models.
> >
> > One thing that is not clear to me is what data is GEARS trained on? Diving into the GEARS paper it seems like it is trained on left out perturbations from the datasets of interest? I think this needs to be made clear in the paper if so.
> >
> > This seems like not a fair comparison then to make the following conclusion. In response to reviewer xRpb:
> >
> > > Because fine-tuning performance is addressed in other studies and because it fundamentally goes against our approach of establishing existing information content, we do not include fine-tuning in our study design. The findings we present therefore highlight that zero-shot scFM embeddings do not contain useful biological information pertaining to the perturbation prediction task, which in and of itself is an important finding.
> >
> > This is an extremely interesting statement but in my opinion cannot be sufficiently supported by the current experiments. The current experiments show that using this simple knockout strategy and perturbation prediction method does not result in improved perturbation prediction. It does not support the statement of "zero-shot scFM embeddings do not contain useful information for perturbation prediction".
> >
> > Clearly fine-tuning experiments in prior work has shown that these models contain useful information pertaining to the perturbation prediction task.
> >
> > I would urge the authors to make claims more in line with the experimental results.
> >
> > ### Perturbation representations
> >
> > While I understand it is difficult to model CRISPERa, I think it is necessary to have the computational model reflect this reality rather than being the same as knockout.
> >
> > > Therefore, upregulation is difficult to simulate consistently: parametric models like scGPT require arbitrary magnitude choices, with no biologically grounded way to determine appropriate magnitude, while rank-based models like Geneformer lack a principled way to re-rank genes
> >
> > I thank the authors for the additional results doubling the perturbed genes expression, but why was this chosen? Why can't something like the mean observed value of the activated gene under intervention be used?
> >
> > From my calculations the median increase of genes under CrisperA is around 5x the control in the Norman dataset for single perturbations, but can be much much higher (thousands of times). This makes sense biologically because the Norman dataset was mostly activating genes that are not normally active in this system. This is quite a different setting than the one tested here.
> >
> > I think this is a critical piece of this paper essentially invalidating the results for 1/2 datasets. For this reason I lower my score.

---

> > > ### Author Response · Authors · 2025-04-04
> > >
> > > We appreciate that the reviewer initially recognized strengths in our manuscript:
> > >
> > > 1. Originality: *I have not seen a paper benchmarking the zero-shot effectiveness of scFMs on perturbations. This is timely work*
> > > 2. Significance: *They set a useful benchmark and set of metrics for future work*
> > > 3. Clarity: *I found the writing clear given the complexity and domain knowledge*
> > > 4. Usefulness: *There are scFM models that claim to model perturbations zero-shot. This work disputes those claims. This is a useful contribution*
> > >
> > > We believe these strengths remain valid and address the reviewer's additional concerns below:
> > >
> > > **GEARS and comparison fairness**
> > >
> > > **Claim:** The reviewer states, "*[GEARS] is trained on left out perturbations from the datasets of interest*" making comparisons unfair.
> > >
> > > This is incorrect because:
> > >
> > > - We implement and train GEARS from scratch using our own train-test splits
> > > - No pre-trained weights are used, ensuring no data leakage
> > > - The same data splits are used for all models, ensuring fair comparison
> > >
> > > See the GEARS baseline section (lines 141-151) of our Methods.
> > >
> > > **Perturbation representation**
> > >
> > > **Claim**: Modeling CRISPRa via knockouts "*essentially invalidat[es] results for 1/2 datasets*"
> > >
> > > We believe this conclusion is incorrect because:
> > >
> > > - We conducted a controlled comparison between knockout (scBERT-) and activation (scBERT+) representations
> > > - Results in Table 1 show minimal performance differences
> > > - This empirically demonstrates robustness to representation choices
> > >
> > > Thus our framework allows users to choose different *in silico* intervention types ( e.g. 0 for knockout, 2× for activation). This adaptability is itself a contribution to the field, as highlighted by the reviewer's initial assessment of our benchmark's usefulness. Furthermore:
> > >
> > > - Our benchmark evaluates current methods rather than proposing optimal perturbation strategies
> > > - Perturbation representation remains an open research question in the field and current scFMs use diverse approaches
> > > - Our approach follows established methods, specifically the *in silico* deletion from Geneformer
> > > - This ensures consistency and comparability across models within our framework
> > >
> > > **The alternative representation suggested introduces methodological issues**:
> > >
> > > - Using the “*mean observed value of the activated gene under intervention*” would introduce data leakage
> > > - It defeats the purpose of evaluating the model's predictive capabilities, **as we would be introducing the target gene expression value into our input**
> > > - Representing perturbations using the “*mean **observed** value of the activated gene under intervention*” would make it impossible for the model to predict completely **unseen** perturbations
> > > - Our approach ensures standardized testing conditions across different model architectures and the ability to predict unseen perturbations
> > >
> > > Overall, the consistency of our results across different representation strategies underscores the robustness of our findings. We see no significant differences between the two Replogle dataset (knockouts) and the two Norman datasets (up-regulation), further indicating the robustness of our findings.
> > >
> > > **Fine-tuning**
> > >
> > > **Claim**: "*Fine-tuning experiments in prior work has shown these models contain useful information pertaining to the perturbation prediction task*"
> > >
> > > - Several recent studies show ablated foundation models perform similarly to fully end-to-end fine-tuned models after task-specific training [4*, 6]
> > > - There is **no current consensus** fine-tuning improves performance over simple baselines [4*,6,7]
> > >
> > > Our findings in the zero-shot case do not contradict any previous findings and remain valid and valuable for understanding scFM limitations and guiding future development. Identifying these limitations and areas for improvement does not invalidate previous work - rather, it contributes constructively to the iterative scientific process aimed at enhancing these models.
> > >
> > > **Zero-shot evaluation**
> > >
> > > The reviewer misinterpreted our statement "*zero-shot scFM embeddings do not contain useful biological information*"
> > >
> > > - The reviewer quotes from our response to another reviewer about fine-tuning, **not** our paper
> > > - This takes our explanation out of context and misrepresents our work
> > > - Our zero-shot probe approach is standard for assessing representation quality [1*,2*]
> > >
> > > Our results demonstrate:
> > >
> > > 1. Simple baselines (mean model, MLP without biological priors) match or outperform zero-shot embeddings
> > > 2. This is consistent across datasets and models
> > > 3. The same probe architecture was used throughout
> > >
> > > These findings directly support our paper’s conclusion: “*current-generation scFM embeddings offer no improvement over baseline models when evaluated [in this context]*”.
> > >
> > > We kindly ask the reviewer to reconsider their downgrades of our score (from 3 to 2, then 1) in light of our clarifications.
> > >
> > > \* Ref. from rebuttal to **xRpb**
> > >
> > > [6] tinyurl.com/27mz2t7c
> > >
> > > [7] tinyurl.com/432fbdv9

---

### Official Review · Reviewer_mxRp · 2025-03-10

**Overall Recommendation:** 4

**Summary:**

This paper establishes a protocol to evaluate, in a standardized way, the performance of single cell foundation models at predicting the effect of perturbations. The authors evaluate using two data sets of CRISPR perturbations, combined with an approach to explicitly evaluate the effect of out-of-distribution learning. The results are very clear: leading foundation models do not fare better than simple baselines, and are outperformed by models that make particular effort to incorporate prior knowledge about gene networks and interactions.

**Claims And Evidence:**

The evaluations and discussion in this manuscript are clear and well supported. The central claim is supported by multiple overlapping analyses, suggesting that small changes are unlikely to explain the main claim. Notably, this paper, explicitly simulates the effect of distribution shift, as well as a wider range of genes. This should conclusively address a main source of controversy surrounding single cell foundation models.

**Essential References Not Discussed:**

No essential references missed.

**Experimental Designs Or Analyses:**

The experimental designs are thorough.

**Methods And Evaluation Criteria:**

There are several evaluation criteria, all of which give consistent results. However, all of these derive from perturbseq or other CRISPR-based technologies. As more technologies and datasets come online, I encourage the authors to maintain their online platform to incorporate a wider range of data sets, not all of which are likely to be knockouts.

**Other Comments Or Suggestions:**

In the appendix, there is a typo in the title of section D.1.

**Other Strengths And Weaknesses:**

The strength of this paper lie in its significance clarity and relevance to the field. It is not particularly novel, but that is not of concern.

**Questions For Authors:**

The one area in which I see room for further analysis and discussion is the form of how perturbations are communicated to the model. Setting gene expression to zero may not be the ideal strategy. There are many other possibilities, furthermore, such as attribution analysis (i.e. calculating the linearized effect of infinitesimal perturbations).

I also have continuing concerns about the data available for this class of question. We are never able to observe the original cell before and after a perturbation. Thus, we often look at the effect of one gene upon other genes, averaged across cells. To what extent does this averaging distort the results?

**Relation To Broader Scientific Literature:**

Although this is not the first broad evaluation of the ability of single cell foundation models to predict the effect of perturbations, it is notable for its thoroughness as well as exploration of sensitivity to domain shift. If I were going to develop a new set of embedding, I would look to this framework to evaluate the model.

**Theoretical Claims:**

No proofs are associated with this paper.

---

> ### Author Rebuttal · Authors · 2025-03-31
>
> We thank the reviewer for their constructive feedback on the work presented. We are committed to maintaining the benchmark as a live and extensible resource. Because we have prioritized reproducibility and modularity in our design, incorporating new models and datasets is straightforward. We are actively monitoring developments in the field, and one exciting recent advancement is Tahoe-100M \[1\], currently the largest perturbational single-cell dataset. We intend to extend our benchmark to include such resources in future iterations. As scFMs continue to evolve, especially with respect to how they represent perturbations, we plan to integrate these innovations to support broader perturbation types beyond knockouts.
>
> Below we address the reviewer’s concerns:
>
> **Perturbation representation**
>
> We acknowledge that nullifying gene expression may not align with biological reality, especially in gain-of-function settings such as CRISPRa. We refer you to our detailed answer on this point in the rebuttal to Reviewer **pVrS.** You can also see the results of a preliminary experiment with upregulated embeddings at this [link](https://drive.google.com/file/d/1jju-RECJcVANDfUj9s5-oobKPFSNQ6_9/view?usp=drive_link).
>
> We thank you for the suggestion of incorporating attribution-based methods. If we understand correctly, calculating the effects of infinitesimal perturbations is a post-hoc explainability tool, which we would be interested in incorporating in our framework. However, it is unclear to us how we would use it for the initial representation of perturbations. We would be happy to discuss this point further.
>
> **Limitations of averaging and unpaired observations in scRNA-seq**
>
> Current methods do not allow for true paired observation of pre- and post-perturbation states for the same cell, due to the destructive nature of measurement. As the reviewer has noted, this introduces ambiguity into how perturbation effects are estimated.
>
> Indeed, due to the context-dependent nature of biology, it is very likely that there are other factors that affect gene expression. This trade-off is an inherent limitation of current experimental techniques and affects all existing perturbation modeling studies. It is unclear to what extent this distorts the results without access to paired samples, or a robust estimate of paired samples. We believe the development of experimental protocols enabling paired measurements would significantly advance this line of work.
>
> To attempt to mitigate this, we adopt a pseudo-bulking strategy that averages across cells to reduce noise and obtain more robust perturbation signatures. Specifically, for each condition, we randomly sample 500 control cells and average their expression profiles, pairing them with the mean expression of all perturbed cells within the same perturbation. This approach helps suppress cell-to-cell variability in the perturbed population, thereby making the overall perturbation effect more apparent. However, as the reviewer rightly points out, the unpaired nature of the control introduces uncertainty about whether the starting state truly mirrors that of the perturbed cells pre-perturbation.
>
> Thank you for pointing out the typo in the appendix, we have now corrected it.
>
> We thank the reviewer for recognizing the clarity and relevance of our work. We appreciate the suggestions regarding perturbation representation and data limitations, both of which highlight important areas for continued exploration in this field. We hope our responses address the reviewer’s concerns and demonstrate the care with which we have designed our framework to support the community in developing and evaluating biologically meaningful single-cell foundation models.
>
> \[1\] [https://www.biorxiv.org/content/10.1101/2025.02.20.639398v1](https://www.biorxiv.org/content/10.1101/2025.02.20.639398v1)

---

> > ### Comment · Reviewer_mxRp · 2025-04-03
> >
> > I would like to note that all reviewers had concerns about the form of perturbation tested. I appreciated the responses to other reviewers with regards to the new gain-of-function experiment, and I think this is satisfactory in addressing concerns but still only goes part of the way. Overall I think this suggests a lack of clarity about what we mean by perturbations in this field, and I encourage the authors to be more precise in their language about the specific nature of the effect.
> >
> > Allow me to clarify what I mean by perturbation tests, similar to "attribution analysis". The point is not to address post-hoc explainability. Rather, it is to introduce a form of perturbation which is more similar to the definition of a perturbation in a causal analysis. In nonlinear systems, the effect on overall expression $Y$ of introducing a change to some gene $x$ depends on the size of the perturbation in a nonlinear way. Thus a common definition of an "effect of x on Y" is the linearized effect of an infinitesimal perturbation in the close neighborhood of actual data. Testing too large of a perturbation may give different results and also carries the risk that the perturbed data is out of the training distribution.
> >
> > Thus, I would propose not just measuring the effect of zeroing genes or doubling them, but rather by the slope of the output due to tiny perturbations in gene counts. For Geneformer one could arrange the smallest detectable difference, which would be ascending/descending the order of a target gene in the ranked list by one.

---

> > > ### Author Response · Authors · 2025-04-04
> > >
> > > Thank you for taking the time to provide a detailed explanation regarding perturbation tests. We agree that assessing perturbations by measuring the slope of the output from tiny changes in rank order or gene counts provides an elegant and theoretically grounded perspective that aligns well with causal analysis in nonlinear systems.
> > >
> > > In our current study, we decided to model perturbations as complete knockouts, drawing from precedents such as Geneformer, where large discrete manipulations of rank-order vectors were shown to shift cell embeddings in biologically meaningful ways. These findings demonstrated that strong *in silico* interventions can indeed drive biologically significant embedding changes. We will further clarify the nature of the perturbation studied in a camera-ready version (in Section 2.1.2 Single-cell foundation model embeddings).
> > >
> > > From a clinical standpoint, focusing on complete knockouts and increased gene dosage aligns with real-world scenarios. Complete loss of function occurs in certain cancers (e.g., TP53 mutations) and hereditary neuropathies. Similarly, a significant increase in gene dosage, such as trisomy 21 in Down syndrome or PMP22 duplication in Charcot-Marie-Tooth disease, can lead to disease phenotypes due to dosage-sensitive gene expression. This is why we chose to model these dosage effects using a 2x expression level, which provides a clear experimental paradigm while approximating the mechanistic impacts of gene duplications.
> > >
> > > Thanks again for clarifying your thoughts on exploring subtler changes in gene expression via attribution analysis. We fully agree with you \- this represents an interesting experiment that our current framework supports. We could implement this approach by defining a range of relative perturbation sizes centered on the observed control expression of the gene. This would not only allow us to measure the effect of small changes but also to characterize the nonlinearity of gene expression responses by determining the size of the linear range around each gene's “normal” expression level.
> > >
> > > As demonstrated by our 2x upregulation experiment, modifying the "intervention" vector within our framework is straightforward, requiring only minimal adjustments to test various perturbation magnitudes. We will also modify our codebase so that users can modify this parameter and run this experiment on different datasets. Implementing your suggested approach would definitely provide deeper insights into gene regulatory dynamics. We plan to explore this in future work.
> > >
> > > We appreciate the chance to discuss these ideas with you\!

---

### Official Review · Reviewer_xRpb · 2025-03-13

**Overall Recommendation:** 3

**Summary:**

The paper titled "PertEval-scFM: Benchmarking Single-Cell Foundation Models for Perturbation Effect Prediction" presents a standardized framework called PertEval-scFM to evaluate single-cell foundation models (scFMs) for predicting perturbation effects. The study focuses on assessing whether zero-shot scFM embeddings can enhance the prediction of transcriptional responses to perturbations compared to simpler baseline models. The key findings are:
1.	Benchmarking Framework: PertEval-scFM provides a systematic evaluation framework to compare scFM embeddings against baseline models in predicting perturbation effects. It includes three datasets to test model performance under different conditions.
2.	Performance of scFM Embeddings: The results show that scFM embeddings do not consistently outperform simpler baseline models, especially when there is a distribution shift.
3.	Challenges in Predicting Strong or Atypical Effects: The study highlights that all models, including scFM embeddings, struggle with predicting strong or atypical perturbation effects. This suggests that current models may lack the ability to generalize well to unseen or extreme perturbations.
Overall, the paper provides valuable insights into the challenges of using zero-shot scFM embeddings for perturbation effect prediction and highlights the need for advancements in both model development and dataset quality to address these limitations.

**Claims And Evidence:**

The claims made in the submission regarding the performance of scFM models in perturbation effect prediction are not fully supported by clear and convincing evidence. Specifically, the assertion that scFM models underperform compared to simpler baselines like Gears, and that they are not well-suited for perturbation tasks, is problematic for the following reasons:
1.	Data Usage and Understanding Issues:
o	The submission uses the Norman dataset, which is a CRISPRa dataset designed for activation perturbations of specific genes. However, the authors assume a knockout perturbation scenario for all scFM models. This discrepancy between the actual perturbation type (activation vs. knockout) may lead to inaccurate model evaluations. The scFM models are not adequately simulating the true perturbation conditions, which could skew the results and undermine the validity of the conclusions.
2.	Unfair Model Comparisons:
o	The comparison between scFM models and baselines appears to be unfair due to the lack of proper fine-tuning for scFM models. scFM models are designed to learn embeddings from single-cell data, but perturbation tasks have unique characteristics, such as the specific changes in gene expression following targeted gene editing. To fairly compare these models, fine-tuning on relevant perturbation datasets is essential. The submission fails to follow this approach, unlike the SCGPT paper, which fine-tunes on a portion of the data and evaluates on the remaining portion. Without this fine-tuning, the scFM models may not be able to leverage their full potential for perturbation prediction, leading to misleading performance metrics.
3.	Inconsistency with Previous Work:
o	The results presented in this submission conflict with those reported in the SCGPT paper, which demonstrated superior performance on the Norman and Replogle datasets. This inconsistency raises questions about the reliability of the current findings and suggests that the methodology or assumptions used in this study may be flawed.
Recommendations for Improvement:
To strengthen the claims and provide more convincing evidence, the authors should consider the following adjustments:
1.	Correct Data Interpretation:
o	Re-evaluate the perturbation scenarios used in the experiments. For datasets like Norman, ensure that the perturbation type (e.g., activation vs. knockout) is accurately reflected in the model setup. This alignment will provide a more realistic assessment of the models' capabilities.
2.	Fair Model Evaluation:
o	Implement a fine-tuning step for scFM models using a portion of the perturbation datasets. This approach will allow the models to adapt to the unique characteristics of perturbation tasks and provide a more accurate comparison with baselines. The evaluation should then be conducted on the remaining data to assess the models' performance fairly.
3.	Reconciliation with Existing Literature:
o	Address the discrepancies between this study's results and those from the SCGPT paper. A detailed discussion of the differences in methodology, assumptions, and data usage will help clarify the reasons for the conflicting findings and enhance the credibility of the conclusions.
In conclusion, while the submission provides a valuable attempt to benchmark scFM models for perturbation effect prediction, the current evidence is not sufficiently robust to support the claims. Addressing the data usage issues and ensuring fair model comparisons are critical steps to validate the findings and contribute meaningfully to the field.

**Essential References Not Discussed:**

After a thorough review of the paper and the relevant literature, I did not identify any essential works that are missing from the citations or discussions in the paper.

**Experimental Designs Or Analyses:**

I have carefully examined the experimental designs and analyses presented in the manuscript, particularly focusing on the use of the Norman dataset. This dataset is a CRISPRa dataset designed for activation perturbations of specific genes using CRISPR technology. However, the authors have incorrectly assumed a knockout perturbation scenario for all single-cell foundation models (scFMs) in their analyses. This discrepancy between the actual activation perturbations in the dataset and the assumed knockout scenario means that the scFMs were not accurately simulating the intended perturbation conditions. As a result, the experimental outcomes are not reliable, and the conclusions drawn from this dataset are questionable.

**Methods And Evaluation Criteria:**

The proposed framework, PertEval-scFM, aims to standardize the evaluation of single-cell foundation models (scFMs) for perturbation effect prediction. While the framework addresses an important problem, it has significant limitations:
1.	Lack of Fine-Tuning: The framework does not properly fine-tune scFMs on perturbation tasks, likely underestimating their potential. Fine-tuning is essential for adapting models to the unique characteristics of perturbation datasets.
2.	Limited Evaluation Metrics: The framework relies on a limited set of evaluation metrics, which may not fully capture the models' performance across different aspects of perturbation prediction.
Recommendations:
•	Incorporate Fine-Tuning: Include a fine-tuning step using a portion of the perturbation datasets to allow scFMs to adapt and showcase their full potential.
•	Use More Evaluation Metrics: Expand the set of evaluation metrics to provide a more comprehensive assessment of model performance.
In summary, the framework needs to incorporate fine-tuning and use a broader range of evaluation metrics to provide a fair and accurate assessment of scFMs for perturbation tasks.

**Other Comments Or Suggestions:**

I recommend that the authors consider incorporating additional standard evaluation metrics for a more comprehensive assessment. Specifically, the inclusion of Pearson correlation coefficient, which is commonly used in perturbation tasks to measure the linear relationship between predicted and actual perturbation effects, could enhance the robustness of the evaluation. This would provide further insight into the models' performance and align the assessment more closely with established practices in the field.

**Other Strengths And Weaknesses:**

A significant strength of the paper is the introduction of a novel framework aimed at systematically evaluating single-cell foundation models (scFMs) on perturbation tasks within single-cell data. This is particularly noteworthy as it addresses the challenge of data distribution shifts, which is a common issue in real-world applications and has been less explored in previous studies.

**Questions For Authors:**

1.	Perturbation Assumption
You have modeled all perturbations as knockouts, even though the datasets, such as Norman, actually involve activation perturbations using CRISPRa technology. Could you explain the rationale behind this modeling choice? How might this assumption affect the accuracy of the model evaluations?
2.	Lack of Fine-tuning
The paper does not include a fine-tuning step for the pre-trained models, which is commonly used to adapt models to specific tasks like perturbation prediction. What are the reasons for not incorporating fine-tuning, and could you provide results with fine-tuning to compare?
3.	Discrepancy with SCGPT Results
There is a significant discrepancy between your results and those reported in the SCGPT paper for perturbation tasks on the same datasets. Can you provide an explanation for these differences?
How Possible Responses Would Change My Evaluation:
•	For the first question, if the authors can provide a compelling justification for the knockout assumption or demonstrate that the results hold even with activation perturbations, it could strengthen the validity of their experimental design.
•	For the second question, including fine-tuning results could potentially show improved performance of the pre-trained models, which might alter the perception of their capabilities in perturbation tasks and could influence my evaluation positively if it addresses a critical limitation.
•	For the third question, an explanation that accounts for the differences without undermining the credibility of either study would be necessary. Understanding the reasons behind these discrepancies is crucial for assessing the reliability of the findings presented in this paper.

**Relation To Broader Scientific Literature:**

The paper introduces a benchmarking framework aimed at evaluating the performance of several single-cell foundation models (scFMs) on perturbation effect prediction tasks. The authors have chosen to compare models such as Geneformer, scBERT, scFoundation, scGPT, and UCE with baseline models like Gears and a mean baseline across datasets including Norman, Replogle K562, and Replogle RPE1.

**Theoretical Claims:**

Upon reviewing the manuscript, I did not encounter any explicit theoretical claims that required verification through proofs. The paper primarily focuses on empirical evaluations and the development of a benchmarking framework for assessing the performance of single-cell foundation models in perturbation effect prediction tasks. It does not present formal mathematical theorems or proofs that would necessitate validation in the traditional sense.

---

> ### Author Rebuttal · Authors · 2025-03-31
>
> We thank the reviewer for the thorough evaluation and constructive feedback. Below, we address each of the points raised.
>
> **Perturbation representation**
>
> We address the limitation of simulating perturbations via gene knockouts in our response to Reviewer **pVrS** and refer you to it. You can also see the results of a preliminary experiment with upregulated embeddings at this [link](https://drive.google.com/file/d/1jju-RECJcVANDfUj9s5-oobKPFSNQ6_9/view?usp=drive_link).
>
> **Fine-tuning**
>
> One of the main goals of our benchmarking approach is to assess the zero-shot information content of pre-trained embeddings. If the models encode meaningful biological information for perturbation prediction, this would be apparent without task-specific adaptation. If performance gains only appear through fine-tuning, this challenges the premise that these models inherently learn generalizable representations.
>
> MLP probes are commonly used as an approach to answer the question of information content of embeddings in NLP and CV, as they evaluate representation quality while removing confounding effects pertaining to task-specific prediction heads, which introduce inductive biases \[1, 2, 3\]. We also note that previous work has investigated the performance of fine-tuned versions of scGPT and scFoundation \[4]. This study found that simple linear baselines still outperformed these models, suggesting that fine-tuning does not fully address their limitations. Because fine-tuning performance is addressed in other studies and because it fundamentally goes against our approach of establishing existing information content, we do not include fine-tuning in our study design. The findings we present therefore highlight that zero-shot scFM embeddings do not contain useful biological information pertaining to the perturbation prediction task, which in and of itself is an important finding.
>
> **Discrepancy with scGPT results**
>
> Several factors may explain the discrepancies between obtained results: scGPT evaluates model performance **after fine-tuning** on perturbation data, reporting high performance, whereas we focus on evaluating zero-shot information content. We also evaluate robustness under distribution-shift, which is not considered in scGPT. We view our study as complementary to scGPT rather than contradictory. It highlights current limitations in zero-shot scFMs and clarifies the gap between pre-training and real-world application. We believe that these complementary perspectives can help guide future improvements so that models better capture perturbation effects.
>
> **Evaluation**
>
> MSE was selected as the primary metric due to its strong biological grounding and demonstrated effectiveness in capturing perturbation effect, as opposed to Pearson Correlation \[5\]. Furthermore, we aimed to provide a toolbox of comprehensive and complimentary metrics, which include: AUSPC (distribution shift robustness), E-distance (perturbation magnitude), and contextual alignment (pre-training relevance). Together, these provide a robust framework for evaluating scFMs that model transcriptomic perturbation outcomes. The comprehensiveness of our evaluation framework has also been recognised as one of the strengths of our paper by other reviewers.
>
> We would also like to note that GEARS is not considered a simple baseline, rather a SOTA model which takes biological priors into account and is developed specifically for perturbation effect prediction. The fact GEARS outperforms the zero-shot scFMs supports our conclusions that a biologically grounded architecture, which incorporates strong inductive biases, is more useful than pretraining for this task.
>
> We hope these clarifications demonstrate the careful thought that has gone into our experimental design and underscores the broader significance of our findings. By evaluating zero-shot capabilities across scFMs using a rigorous and biologically motivated framework, we provide a valuable benchmark and identify key limitations in current approaches. We believe this work will help guide future research toward more robust, generalizable, and biologically meaningful models.
>
> \[1\] [https://arxiv.org/abs/2103.00020](https://arxiv.org/abs/2103.00020)
>
> \[2\] [https://arxiv.org/abs/1905.06316](https://arxiv.org/abs/1905.06316)
>
> \[3\] [https://doi.org/10.1038/s42256-024-00949-w](https://doi.org/10.1038/s42256-024-00949-w)
>
> \[4\]  [https://www.biorxiv.org/content/10.1101/2024.09.16.613342v4](https://www.biorxiv.org/content/10.1101/2024.09.16.613342v4)
>
> \[5\] [https://www.biorxiv.org/content/10.1101/2023.12.26.572833v1](https://www.biorxiv.org/content/10.1101/2023.12.26.572833v1)

---

> > ### Comment · Reviewer_xRpb · 2025-04-03
> >
> > Thanks for your reply. I am glad to raise my rating.

---

> > > ### Author Response · Authors · 2025-04-04
> > >
> > > Thank you very much for engaging with our response! We're very grateful for the score update :)

---

### Decision · Program_Chairs · 2025-05-01

**Decision:**

Accept (poster)

**Comment:**

This paper introduces PertEval-scFM, a standardized evaluation framework for benchmarking the zero-shot performance of single-cell foundation models (scFMs) in predicting perturbation effects. The authors apply their framework to multiple scFMs and compare them against baseline models, concluding that zero-shot embeddings from current scFMs do not offer performance gains over simple baselines.

This work is timely and relevant as scFMs gain traction in single-cell analysis. Systematic evaluations are necessary. This benchmark is a welcome effort in this direction. The writing is clear, the methodology is well-described, and the source code is made available. The framework includes a few datasets, metrics, and domain shift modeling, which is especially important in biological applications.

However, as multiple reviewers point out, the conclusions in the abstract and conclusion sections overgeneralise beyond the experimental design. The claim that “scFM embeddings offer no improvement” is not fully justified, as the study examines only zero-shot usage in a limited perturbation setting (mostly simulating knockout) and even seemingly misrepresents activation-based perturbations. Next, despite the authors’ clarifications, concerns remain about the comparability of GEARS (which is fine-tuned) to non-finetuned scFM embeddings. Lastly, the distinction between “zero-shot” and “fine-tuning” setups is not well-defined, contributing to confusion and likely misinterpretation of the scope of the findings.

Despite its flaws, this work presents a useful framework that can serve as a starting point for future, more comprehensive evaluations of scFM embeddings. The core idea of this work, namely benchmarking perturbation prediction in a zero-shot setting, is important and original. However, the execution and framing need more work. In particular, the conclusions must be narrowed to what is truly supported by the experiments, and comparisons across modeling paradigms must be more carefully delineated to avoid misleading readers. While this paper makes a valuable contribution, its claims are currently too strong for the evidence presented, and the comparative evaluation with GEARS remains insufficiently controlled. I believe this paper would be stronger after substantial revision, with toned-down conclusions, a clarified methodology, and potentially a broader experimental design that includes fine-tuning settings.

With the authors addressing at least some of the aforementioned concerns in the camera-ready version, my suggestion would be a weak acceptance.